# Rényi Sharpness: A Novel Sharpness that Strongly Correlates with Generalization

**Qiaozhe Zhang , Jun Sun**[*]**, Yingzhuang Liu**
School of Electronic Information and Communications
Huazhong University of Science and Technology
`{qiaozhezhang, juns, liuyz}@hust.edu.cn`

## Abstract

Sharpness (of the loss minima) is widely believed to be a good indicator of generalization of neural networks. Unfortunately, the correlation between existing sharpness measures and the generalization is not that strong as expected, sometimes even contradiction occurs. To address this problem, a key observation in this paper is: what really matters for the generalization is the *average spread* (or unevenness) of the spectrum of loss Hessian $\mathbf{H}$. For this reason, the conventional sharpness measures, such as the trace sharpness $\mathrm{tr}(\mathbf{H})$, which cares about the *average value* of the spectrum, or the max-eigenvalue sharpness $\lambda_{\max}(\mathbf{H})$, which concerns the *maximum spread* of the spectrum, are not sufficient to well predict the generalization. To finely characterize the average spread of the Hessian spectrum, we leverage the notion of *Rényi entropy* in information theory, which is capable of capturing the unevenness of a probability vector and thus can be extended to describe the unevenness for a general non-negative vector (which is the case for the Hessian spectrum at the loss minima). In specific, in this paper we propose the *Rényi sharpness*, which is defined as the negative of the Rényi entropy of loss Hessian $\mathbf{H}$. Extensive experiments demonstrate that Rényi sharpness exhibit *strong* and *consistent* correlation with generalization in various scenarios. Moreover, on the theoretical side, two generalization bounds with respect to the Rényi sharpness are established, by exploiting the desirable reparametrization invariance property of Rényi sharpness. Finally, as an initial attempt to take advantage of the Rényi sharpness for regularization, Rényi Sharpness Aware Minimization (RSAM) algorithm is proposed where a variant of Rényi Sharpness is used as the regularizer. It turns out this RSAM is competitive with the state-of-the-art SAM algorithms, and far better than the conventional SAM algorithm based on the max-eigenvalue sharpness.

## 1 Introduction

Understanding why stochastic optimization methods, such as stochastic gradient descent (SGD) can achieve strong generalization performance even when the neural networks are overparameterized remains a fundamental yet open challenge in deep learning (Zhang et al., 2016; Gunasekar et al., 2017; Li et al., 2018; Soudry et al., 2018; Woodworth et al., 2020). Many empirical and theoretical studies have observed that the generalization of neural networks is closely related to the flatness of the loss landscape (Keskar et al., 2016; Neyshabur et al., 2017; Jiang et al., 2019; Petzka et al., 2019; Kaddour et al., 2022; Tsuzuku et al., 2020; Jang et al., 2022; Dziugaite & Roy, 2017; Jastrzębski et al., 2017; Wu et al., 2018; Blanc et al., 2020; Wei & Ma, 2019; Foret et al., 2020; Damian et al., 2021; Li et al., 2021; Ma & Ying, 2021; Ding et al., 2024; Nacson et al., 2022; Lyu et al., 2022; Wu & Su, 2023; Kwon et al., 2021; Zhou et al., 2024).

Intuitively speaking, local minima with flat (with low sharpness) neighborhood in the landscape are expected to incur small loss change (Hochreiter & Schmidhuber, 1994; Keskar et al., 2016).

---

[*]Jun Sun (juns@hust.edu.cn) is the corresponding author.
The source code is publicly available at this link.

The core question is therefore: how should we measure the flatness in a proper way? The flatness or *sharpness* is normally quantified by functionals of the loss Hessian $\mathbf{H}$—e.g., $\mathrm{tr}(\mathbf{H})$ and $\lambda_{\max}(\mathbf{H})$—or by loss increase with constrained weight perturbations, while the latter is normally closely related to the former. Despite the above intuition, recent empirical evidences indicate that existing sharpness measures usually correlate *weakly* with generalization (Andriushchenko et al., 2023), sometimes even contradicting phenomenon occurs (Dinh et al., 2017; Wen et al., 2023). To close the gap between the intuition and the reality, it is of crucial importance to develop a better sharpness measure.

To address this problem, a key observation of ours is: what really matters for characterizing the generalization lies in the *unevenness* or *average spread* of the spectrum of the Hessian. Intuitively speaking, an even spectrum (with almost identical eigenvalue) is very much desirable to ensure good generalization, since if there exists no particularly large eigen-direction, a small perturbation of data (which can be translated to weight perturbation) would just incur small loss change. More concretely, when characterizing the loss change resulting from the train-test data discrepancy, the unevenness or average spread of the the spectrum can reflect the influences from all categories of eigenvalues of loss Hessian (Sankar et al., 2021): 1) the *top eigenvalues*, which are very important for the loss change but are of quite small quantity; 2) the *middle eigenvalues*, which are less important for the loss change individually but are of very big quantity; 3) the *tail eigenvalues*, which are normally located near zero and thus play a minor role regarding the loss change. In contrast, the conventional sharpness measures, such as the trace or maximum eigenvalue of the loss Hessian, they care about only part of the eigenvalues. For example, the trace sharpness $\mathrm{tr}(\mathbf{H})$ actually cares about only middle eigenvalues, while the max-eigenvalue sharpness $\lambda_{max}(\mathbf{H})$ concerns only top eigenvalues. Therefore, both of them might experience significant information loss when predicting the generalization performance.

To finely characterize the unevenness or average spread of the Hessian spectrum, we propose a novel sharpness measure, Rényi sharpness by leveraging the notion of Rényi entropy (Rényi, 1961) in information theory, which can well describe the unevenness of a probability vector $\mathbf{p}$ by exploiting an appealing property, i.e. concavity in $\mathbf{p}$. Naturally, Rényi entropy can be employed to describe the unevenness of any general non-negative vector by normalization, i.e. by transforming the original vector to a virtual probability vector. Moreover, Rényi entropy enjoys extra advantages of flexibility (with one free parameter) compared against the classical Shannon entropy (c.f. Section 2). In addition, it is worth noting that to describe the average spread of a vector, the sample variance is an alternative which is easy to enter the mind. Unfortunately it is improper for characterizing the generalization, because the tail eigenvalues (near-zero) of the spectrum contribute a lot to the variance, while they play a very minor role for the generalization gap.

To rigorously establish the relationship between generalization and Rényi sharpness, we develop several generalization bounds in terms of Rényi sharpness, by taking advantage of the reparametrization invariance property of Rényi sharpness, and the technique of translating data discrepancy to the multiplicative weight perturbation. Moreover, to verify the correlation between the Rényi sharpness and generalization, we provide a fast algorithm, which is based on the Stochastic Lanczos Quadrature (SLQ) method (Yao et al., 2020), to estimate the Rényi sharpness. Finally, we introduce Rényi Sharpness-Aware Minimization (RSAM) for network training, which basically employs the Rényi sharpness as a regularizer.

In summary, our contributions are stated as follows:

- We introduce a novel notion of sharpness – *Rényi sharpness*, which is motivated by the observation that generalization highly depends on the *average spread* of the spectrum of the loss Hessian, which can be captured by the Rényi entropy, an important functional in information theory.

- We present two *generalization bounds* with respect to the Rényi sharpness, thus establishing the link between them in a rigorous way. In developing these generalization bounds, it is important to leverage the reparametrization invariance of the Rényi sharpness and the technique of translating data perturbation to (multiplicative) weight perturbation.

- We demonstrate that there exists *strong correlation* between Rényi sharpness and generalization. Meanwhile, a fast algorithm to estimate the Rényi sharpness, which leverages the Stochastic Lanczos Quadratur (SLQ) method, is proposed.

- A preliminary version of *Rényi Sharpness-Aware Minimization* (RSAM) is proposed, where a variant of Rényi Sharpness is employed as a regularizer during training. It turns out to be competitive with the state-of-the-art SAM algorithms and significantly outperform the conventional SAM method, such as that using max-eigenvalue sharpness.

## 1.1 RELATED WORKS

**Sharpness vs. Generalization:** The exploration of relationship between sharpness and generalization dates back to Hochreiter & Schmidhuber (1994), which proposes an algorithm to achieve high generalization capability by searching flat minima. Keskar et al. (2016) shows that the generalization performance of large batch SGD is correlated with the sharpness of the minima. Neyshabur et al. (2017) studies various generalization measures and highlights the promising correlation between sharpness and generalization. Jiang et al. (2019) performs a large-scale empirical study and finds that flatness-based measure is higher correlated with generalization than the concepts like weight norms, margin-, and optimization-based measures. Petzka et al. (2021) studies a relative flatness of a layer through a multiplicative perturbation setting and shows the correlation with generalization. However, many recent studies point out that sharpness does not correlate well with generalization. Dinh et al. (2017) focuses on deep networks with rectifier units and builds equivalent models whose sharpness can be significantly changed. Andriushchenko et al. (2023) find that sharpness may not have a strong correlation with generalization for a collection of modern architectures and settings. Wen et al. (2023) shows that flatness provably implies generalization but there exist non-generalizing flattest models. Kaur et al. (2023) shows that the maximum eigenvalue of the Hessian can not always predict generalization even for models obtained via standard training methods. A central reason why these works consider sharpness to be unreliable is that there exist sharp models with good generalization.

**Sharpness-Aware Minimization (SAM):** As early as 1994, Hochreiter & Schmidhuber (1994) sought to achieve stronger generalization by identifying flat minima, many recent researches find that sharpness is correlated with generalization. This investigation inspires multiple methods that optimize for more flat minima. These algorithms impose penalties based on different criteria, such as the trace in average case (Jia & Su, 2020) or the worst-case perturbation such as SAM (Foret et al., 2020) and its variations (Kwon et al., 2021; Zhuang et al., 2022; Du et al., 2022; Kim et al., 2022; Mi et al., 2022; Li & Giannakis, 2023; Li et al., 2024a). To enhance the generalization, Eigen-SAM is proposed (Luo et al., 2024) which periodically estimates the top eigenvalue of the Hessian matrix and incorporates its orthogonal component to the gradient into the perturbation, thereby achieving a more effective top eigenvalue regularization effect. To obtain parameter-invariant sharpness measures, a universal class of sharpness is proposed in Tahmasebi et al. (2024).

## 2 PROBLEM FORMULATION, KEY NOTIONS AND PROPERTIES

**Model**. Let $f(\boldsymbol{\theta}, \mathbf{x})$ be a model with $L$ layers, where $\boldsymbol{\theta} = \{\mathbf{W}_1, \mathbf{W}_2, \ldots, \mathbf{W}_{L-1}, \mathbf{W}_L\}$, and $\mathbf{W}_l$ is the weights of the $l$-th layer, the vectorization of $\boldsymbol{\theta}$ and $\mathbf{W}_l$ is $\theta$ and $\mathbf{w}_l = \text{vec}(\mathbf{W}_l)$ correspondingly. For a given training dataset $\mathcal{S} = \{\mathbf{x}_i, \mathbf{y}_i\}^n$, and a twice differentiable loss function $l(f(\boldsymbol{\theta}, \mathbf{x}), \mathbf{y})$, the empirical loss is given by $L(\mathcal{S}, \boldsymbol{\theta}) = \frac{1}{n} \sum_{i=1}^{n} l(f(\boldsymbol{\theta}, \mathbf{x}_i), \mathbf{y}_i)$. The training and testing dataset is sampled from the real data distribution $\mathcal{D}$, and the population loss is given by $L(\mathcal{D}, \boldsymbol{\theta}) = \mathbb{E}_{(\mathbf{x}, \mathbf{y}) \sim \mathcal{D}}[l(f(\boldsymbol{\theta}, \mathbf{x}), \mathbf{y})]$. The generalization gap is defined as the difference between the population loss $L(\mathcal{D}, \boldsymbol{\theta})$ and the empirical loss $L(\mathcal{S}, \boldsymbol{\theta})$.

Having observed only $\mathcal{S}$, the model utilizes $L(\mathcal{S}, \boldsymbol{\theta})$ as an estimation of $L(\mathcal{D}, \boldsymbol{\theta})$, and solves $\min_{\boldsymbol{\theta}} L(\mathcal{S}, \boldsymbol{\theta})$ using an optimization procedure such as SGD or Adam.

**Rényi Entropy**. Rényi entropy is a generalization of the classical Shannon entropy, which enjoys the advantage of increased flexibility by adding one parameter and reduced computational complexity. The Rényi entropy of a probability vector $\mathbf{p} = [p_1, p_2, \ldots, p_n]$ is defined as

$$H_\alpha(\mathbf{p}) = \frac{1}{1-\alpha} \log \sum_{i=1}^{n} p_i^\alpha \qquad (1)$$

for $0 < \alpha < \infty$ and $\alpha \neq 1$. The Shannon entropy can be seen as a special example when the order $\alpha \to 1$.

Two notable properties of Rényi entropy are as follows: 1) **Concavity over $\mathbf{p}$** : Rényi entropy is a concave function of the distribution $\mathbf{p}$. A direct implication of this property is that Rényi entropy takes its maximum when $\mathbf{p}$ is *uniformly* distributed. 2) **Monotonic decrease in $\alpha$** : When $\alpha$ increases, the penalty over the non-uniformity (or unevenness) gets more strict, thus more emphasis would be on the high probability mass, and vice versa.

The Rényi entropy can be generalized to the matrix setting. In specific, for a positive definite matrix $\mathbf{H}$, we can define its Rényi entropy as the normal Rényi entropy of its normalized eigenvalues, i.e.,

$$H_\alpha(\mathbf{H}) = \frac{1}{1-\alpha}\log\sum_{i=1}^{n}(\frac{\lambda_i(\mathbf{H})}{\text{Tr}(\mathbf{H})})^\alpha. \tag{2}$$

In theory, we typically analyze the Hessian at a (local) minima, and therefore assume the Hessian to be positive definite. In practice, however, due to imperfect convergence or numerical errors in the algorithm, some negative eigenvalues may appear. Since these eigenvalues usually have very small magnitudes, we commonly take their absolute values before performing subsequent computations.

**Definition 2.1 (Rényi Sharpness)** *For a neural network, consider an arbitrary layer within the model, denote the Hessian matrix of the loss function w.r.t. the layer's weight as $\mathbf{H}$. The Rényi sharpness is defined as the negative Rényi entropy of the normalized spectrum of $\mathbf{H}$, i.e., $-H_\alpha(\mathbf{H})$.*

Rényi Sharpness has a valuable property, i.e., the reparametrization invariance when the activation functions are homogeneous or nearly homogeneous. This property turns out to play an important role in developing the generalization bounds in terms of Rényi Sharpness. A formal statement regarding this property is as follows:

**Proposition 2.2 (Reparameterizaiton (Scaling) Invariance of Rényi Sharpness)** *Consider a $L$-layer feedforward neural network with positively homogeneous activation function $\sigma$ (i.e., $\sigma(c\mathbf{x}) = c\sigma(\mathbf{x})$ for all $c > 0$), and parameters $\{\mathbf{W}_1, \ldots, \mathbf{W}_L\}$. Let the network output be $f(\mathbf{x}) = \mathbf{W}_L \cdot \sigma(\mathbf{W}_{L-1}\cdots\sigma(\mathbf{W}_1 x))$, and let $\mathcal{L}(\boldsymbol{\theta})$ denote the loss function, where $\boldsymbol{\theta}$ denotes the weights of arbitrary layer, i.e., $\mathbf{W}_l$. Define the loss Hessian as $\mathbf{H}_{\boldsymbol{\theta}} = \nabla_{\boldsymbol{\theta}}^2 \mathcal{L}(\boldsymbol{\theta})$. Consider a layer-wise scaling transformation defined by $\tilde{\mathbf{W}}_l = c_l\mathbf{W}_l, \quad c_l > 0, \quad with \prod_{l=1}^{L} c_l = 1$. Let $\tilde{\boldsymbol{\theta}} = \tilde{\mathbf{W}}_l$ be the scaled parameters, and define $\mathbf{H}_{\tilde{\boldsymbol{\theta}}}$ as the corresponding Hessian. Then the spectrum-normalized Rényi entropy of $\mathbf{H}$ is invariant:*

$$H_\alpha(\mathbf{H}_{\tilde{\boldsymbol{\theta}}}) = H_\alpha(\mathbf{H}_{\boldsymbol{\theta}}), \quad \forall\alpha > 0,\ \alpha \neq 1. \tag{3}$$

The detailed description about reparameterization invariance and the proof of Proposition 2.2 is provided in Appendix E. This invariance is valid for the positive homogeneity of the activation function. In Transformer architectures (e.g., ViTs), although GELU is not strictly homogeneous, one has $\text{GELU}(\alpha x)/\alpha \approx \text{GELU}(x)$ (Andriushchenko et al., 2023), and thus the Rényi sharpness is approximately invariant in this setting. Note that when the order $\alpha \to 1$, the Rényi entropy reduces to the Shannon entropy, which is also invariant under the settings in Proposition 2.2. We also remark that this invariance only holds for the layerwise sharpness, the connection between global sharpness and the layerwise one can be found in Appendix F.

## 3 GENERALIZATIONS BOUNDS WITH RESPECT TO RÉNYI SHARPNESS

In this section, we will provide several generalization bounds in terms of Rényi sharpness, by taking advantage of the trick of translating the data discrepancy to multiplicative weight perturbation and the reparameterization invariance of Rényi sharpness.

First of all, we'll argue that the data perturbation can be translated to the multiplicative weight perturbation when characterizing the generalization.

**Proposition 3.1 (informally)** *For any $\rho > 0$, and a training set $\mathcal{S}$ draw from the distribution $\mathcal{D}$, with high probability,*

$$L(\mathcal{D}, \boldsymbol{\theta}) \leq \mathbb{E}_{\mathbf{A}}[L(\mathcal{S}(\mathbf{A}, \rho), \boldsymbol{\theta})] + C \tag{4}$$

*where $\mathcal{S}(\mathbf{A}, \rho) = \{(\mathbf{x} + \rho\mathbf{A}\mathbf{x}, \mathbf{y})|(\mathbf{x}, \mathbf{y}) \in \mathcal{S}\}$ and $\mathbf{A}$ is a orthogonal matrix sampled under Haar measure, i.e., uniform on $\mathcal{O}(d)$.*

The more detailed description and proof of Proposition 3.1 can be found in Appendix B. Intuitively, Theorem 3.1 uses $\mathcal{S}(\mathbf{A}, \rho)$ to approximate $\mathcal{D}$, treating the discrepancy between $\mathcal{D}$ and $\mathcal{S}$ as the perturbation to $\mathcal{S}$. This assumption is essentially akin to the data-separation assumption: data from different classes are spatially separated with no inter-class overlap. Under this premise, one can perturb a sample within its class (i.e., move along the within-class manifold) without affecting other classes. Note that $\mathcal{D}$ and $\mathcal{S}$ can also be feature distributions, thus we can also bound the population loss using the perturbation in the feature space.

The key idea of the perturbation translation is that a multiplicative perturbation in input (feature) space can be transferred into parameter space. A neural network can be written as a composite function $f = g(\mathbf{W}h(x))$, where $\mathbf{W}$ is the weight at a given layer, $h(x)$ is the function consisting of the layers in front of $\mathbf{W}$ all the way to the input, and $g$ is the function behind the $\mathbf{W}$ all the way to the output. Let $f = g(\mathbf{W}h(\mathbf{x}))$, if $h(\mathbf{x}) = \mathbf{x}$, then $\mathbf{W} = \mathbf{W}_1$, which is the weights of the first layer, and the perturbation to $h(\mathbf{x})$ happens in input space, other-wisely happens in feature space. Consequently,

$$g(\mathbf{W}(h(\mathbf{x}) + \rho \mathbf{A} h(\mathbf{x}))) = g(\mathbf{W}(\mathbf{I} + \rho \mathbf{A})h(\mathbf{x})) = g((\mathbf{W} + \rho \mathbf{W}\mathbf{A})h(\mathbf{x})) \qquad (5)$$

i.e., the perturbation to the $h(\mathbf{x})$ is fully transferred to the parameter $\mathbf{W}$. Thus, the generalization gap is closely related to the sharpness of a single layer, therefore we can examine the generalization by studying the sharpness of only a single layer.

Based on the above translation result and motivated by the work of (Jia & Su, 2020), we have the first generalization bound based on Rényi sharpness as follows (informally stated):

**Theorem 3.2 (informally)** *Let $\theta^* \in \mathbb{R}^n$ be the parameter of one layer and be an isolated local minimum of a bounded loss function $L(\cdot, \cdot) \in [0, 1]$, and define a posterior $\mathcal{Q}$ concentrated near $\theta^*$ via local loss deviations, i.e., $\mathcal{Q}$ has a density $q(\theta) \propto e^{-|L_0 - L(\theta)|}$, where $L(\theta)$ is the loss function and $L_0$ is the minima loss obtained by the optimization algorithm. Then, for any $\delta \in (0, 1]$ and $\alpha > 0, \alpha \neq 1$, with probability at least $1 - \delta$ over a training set $\mathcal{S}$ of size $N$, we have:*

$$\mathbb{E}_{\mathcal{Q}}[L(\mathcal{D}, \theta)] \leq \mathbb{E}_{\mathcal{Q}}[L(\mathcal{S}, \theta)] + 2\sqrt{\frac{2L_0 + C\, V^{2/n} \exp\left(-\frac{1}{n}\left[H_\alpha(\mathbf{H}) - A\right]\right) + \log \frac{2N}{\delta}}{N - 1}}, \qquad (6)$$

*where $V$ is the volume of the neighborhood $\mathcal{M}(\theta^*)$, and $A$, $C$ are positive constants, $\mathbf{H} = \nabla_\theta^2 L(\mathcal{S}, \theta^*)$ is the Hessian at $\theta^*$ and $H_\alpha(\mathbf{H})$ is the Rényi entropy of order $\alpha$ of the normalized eigenvalues of $\mathbf{H}$.*

To exhibit a more direct relationship between the population risk and the empirical risk, we provide another generalization bound as follows:

**Theorem 3.3 (informally)** *Given a loss function $L(\cdot, \cdot)$ and a layer-wise local minimum $\theta^* \in \mathbb{R}^n$. Let $\mathbf{H}$ denote the Hessian of the loss w.r.t. $\theta^*$. Take a prior uniform in a ball that contains the ellipsoid $E_{\mathbf{H}}(\rho) = \{\theta : (\theta - \theta^*)^\top \mathbf{H}(\theta - \theta^*) \leq \rho^2\}$, where $\rho$ is sufficiently small and satisfy $\rho > 0$. Take a posterior uniform in $E_{\mathbf{H}}(\rho)$. For any $\delta \in (0, 1]$ and $\alpha > 0, \alpha \neq 1$, we have with probability at least $1 - \delta$ over a training set $\mathcal{S}$ of size $N$, we have:*

$$L(\mathcal{D}, \theta^*) \leq L(\mathcal{S}, \theta^*) + \frac{n}{2(n+2)}\rho^2 + O(\varepsilon) + \sqrt{\frac{-\frac{1}{2}H_\alpha(\mathbf{H}) + \log \frac{2\sqrt{N}}{\delta} + C}{2(N - 1)}}. \qquad (7)$$

*where $C$ is a positive constant, $\mathbf{H} = \nabla_\theta^2 L(\mathcal{S}, \theta^*)$ is the Hessian at $\theta^*$ and $H_\alpha(\mathbf{H})$ is the Rényi entropy of order $\alpha$ of the normalized eigenvalues of $\mathbf{H}$.*

The detailed version and proof of Theorem 3.2 and Theorem 3.3 can be found in Appendices C and D, respectively. Both Theorem 3.2 and Theorem 3.3 indicate that the generalization is bounded by the Rényi entropy of the Hessian matrix of the loss with respect to the weights.

## 4 RÉNYI SHARPNESS: ORDER SELECTION & FUNCTIONAL ESTIMATION

In this section, we will discuss the choice of the order parameter $\alpha$ in Rényi sharpness. Furthermore, we will provide a fast algorithm for estimating the Rényi sharpness.

## 4.1 ORDER SELECTION IN RÉNYI SHARPNESS

The heavy-tailed spectrum of the Hessian matrix is a ubiquitous feature in deep networks. In this section, we compute the Hessian spectrum of each layer by PyHessian (Yao et al., 2020), and find that although all the spectra are heavy-tailed, the shapes of the spectrum can be divided into two categories, which correspond to different choices of $\alpha$.

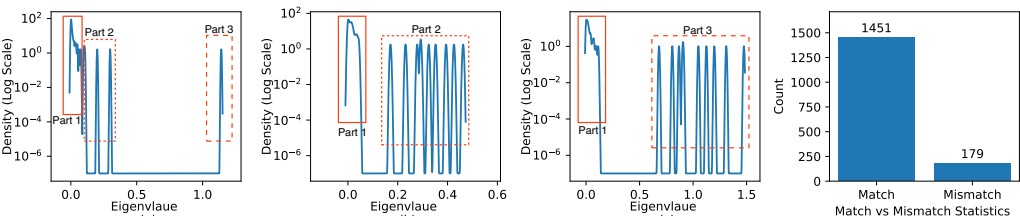

Figure 1: **Hessian spectra [a,b,c].** Two zero-dominant profiles are observed: (a) *multi-cluster* and (b,c) *uniform*. **Optimal $\alpha$ vs. Hessian spectral type [d].** Statistics summarizing whether the empirically optimal $\alpha$ matches the predicted choice under each Hessian spectral type.

We summarize the shape of the spectrum into the following two categories: 1) Zero-dominant, multi-cluster spectrum and 2) Zero-dominant, uniform spectrum. We selected representative plots from ResNet18-CIFAR10 to illustrate these two categories, as shown in Fig. 1. The zero-dominant, multi-cluster spectrum (Fig. 1 (a)) consists of a large number of near zeros (Part 1) and some large eigenvalues (Part 3), and between these two eigenvalues, there are some eigenvalues (Part2) that cannot be ignored but are significantly smaller than the large eigenvalues. The zero-dominant, uniform spectrum (Fig. 1 (b,c)), on the other hand, contains only a large number of near zeros and some large eigenvalues. The detailed spectrum of each layer across different tasks is pushed to Appendix K.1, and a similar spectrum can also be found in Sankar et al. (2021).

To capture the multi-cluster nature (Fig. 1 (a)), we note that eigenvalues near zero (Part 1) contribute less to sharpness and generalization. Therefore, it is important to choose a suitable $\alpha$ that embodies the differences among the dominant (Part 3) eigenvalues and those small but non-negligible eigenvalues (Part 2). When $\alpha > 1$, the measure disproportionately amplifies large eigenvalues while ignoring smaller ones. To better capture the spectrum's subtle variations, especially on Part 2, it is preferable to use an order $\alpha \in (0, 1)$, which balances sensitivity across both large and small eigenvalues. In practice, we observe that setting $\alpha = 0.5$ typically yields the most stable and significant correlation between Rényi sharpness and generalization.

In the case of uniform spectrum (Fig. 1 (b,c)), one part of Part 2 and Part 3 vanish, leaving only a few dominant ones. Therefore, it becomes crucial to capture the differences among these dominant eigenvalues. When $\alpha \in (0, 1)$, the order tends to suppress these differences, which is undesirable in this context. Thus, choosing $\alpha \geq 1$ is more appropriate, as it captures the contribution of every eigenvalue and highlights their differences. However, as $\alpha$ approaches 1, practical numerical computation becomes unstable. Balancing theory and practice, $\alpha > 1$ will be better, and we find that $\alpha = 1.5$ performs well and exhibits a strong and robust correlation.

Overall, the key to choosing $\alpha$ is whether the eigenvalues that influence generalization form clusters whose inter-cluster separation exceeds the clusters' enlargement. If there is a single cluster, selecting $\alpha > 1$ suffices to examine inter-eigenvalue differences. When clusters are widely separated, we should choose $\alpha < 1$ to avoid over-emphasizing the larger eigenvalues when $\alpha > 1$. In practice, $\alpha = 0.5$ and $\alpha = 1.5$ tend to provide robust and consistent results across different datasets and models. The summary statistics of the average correlations for different values of $\alpha$ can be found in the Appendix K.4.

We conducted a statistical analysis of the experiments in Section 5, examining whether the value of $\alpha$ that yields the highest correlation between the layer-wise Rényi sharpness and generalization is consistent with our prior analysis. We then recorded the number of successful and unsuccessful matches in 60 models, with a total of 1630 cases: 1451 matches and 179 mismatches, as shown in Fig. 1 (d). Overall, the empirical findings agree well with our preceding intuitive analysis.

## 4.2 Estimation of Rényi Sharpness

To estimate the Rényi entropy of the Hessian matrix, it would be of prohibitive complexity if we directly calculate the spectrum of the Hessian matrix, due to the huge size of the matrix. To circumvent this difficulty, we will reformulate the Rényi entropy as a functional of the trace of matrix functions and then leverage the stochastic trace estimator (also known as the Hutchinson method) and stochastic Lanczos quadrature method to greatly reduce the complexity.

Firstly, the Rényi entropy is reformulated as follows:

$$H_\alpha(\mathbf{H}) = \frac{1}{1-\alpha}\log\sum_{i=1}^{n}(\frac{\lambda_i}{\text{Tr}(\mathbf{H})})^\alpha = \frac{1}{1-\alpha}\log\frac{\sum_{i=1}^{n}\lambda_i^\alpha}{\text{Tr}(\mathbf{H})^\alpha} = \frac{1}{1-\alpha}\log\frac{\text{Tr}(\mathbf{H}^\alpha)}{\text{Tr}(\mathbf{H})^\alpha}. \tag{8}$$

Thus the estimation task boils down to calculating the trace of matrix functions $\text{Tr}(\mathbf{H})$ and $\text{Tr}(\mathbf{H}^\alpha)$.

To estimate the trace of matrix functions $f(\mathbf{H})$, the stochastic trace estimator can be leveraged to greatly reduce the complexity:

$$\text{Tr}(f(\mathbf{H})) = \text{Tr}(f(\mathbf{H})\mathbf{I}) = \text{Tr}(f(\mathbf{H})\mathbb{E}[\mathbf{v}\mathbf{v}^\top]) = \mathbb{E}[\text{Tr}(f(\mathbf{H})\mathbf{v}\mathbf{v}^\top)] = \mathbb{E}[\mathbf{v}^\top f(\mathbf{H})\mathbf{v}], \tag{9}$$

where $f$ is analytic inside a closed interval function, $\mathbf{I}$ is the identity matrix, and $\mathbf{v}$ is sampled from a Rademacher distribution.

To economically calculate the expectation of the quadratic form $\mathbf{v}^\top f(\mathbf{H})\mathbf{v}$, the Gaussian quadrature rule can be employed to transform the expectation to an integral. Further, the integral can be computed with the nodes and the weights of the quadrature rule given by the Lanczos algorithm, (Golub & Strakoš, 1994; Golub & Meurant, 2009; Bai & Golub, 1996; Bai et al., 1996; Golub & Van Loan, 2013; Ubaru et al., 2017) which basically generates an orthonormal basis for the Krylov subspace such that the matrix can be reduced to tri-diagonal one, hence greatly lower the computational burden. Combined all the above, it constitutes the framework of the stochastic Lanczos quadrature (SLQ) algorithm (Ubaru et al., 2017), which is exactly the basis of Algorithm 1.

The details for the estimation of Rényi entropy are shown in **Algorithm** 1.

---

**Algorithm 1** Rényi Entropy Estimation via Stochastic Lanczos Quadrature

---

**Input:** Positive definite matrix $\mathbf{H}$ of size $n \times n$, Lanczos iterations $m$, computation iterations $l$, order $\alpha > 0$ and $\alpha \neq 1$.
**Output:** Estimation of $H_\alpha(\mathbf{H})$.
**for** $k = 1, ..., l$ **do**
    Draw two random vector $\mathbf{v}_1$ and $\mathbf{g}_k$ of size $n \times 1$ from $\mathcal{N}(0,1)$ and normalize it, $\mathbf{w}_1^{'} = \mathbf{H}\mathbf{v}_1$,
    $\alpha_1 = \mathbf{w}_1^{'\top}\mathbf{v}_1$, $\mathbf{w}_1 = \mathbf{w}_1^{'} - \alpha_1\mathbf{v_1}$;
    **for** $i = 2, ..., m$ **do**
        1). $\beta_i = \|\mathbf{w}_{j-1}\|$;
        2). stop if $\beta_i = 0$ else $\mathbf{v}_i = \mathbf{w}_{i-1}/\beta_j$
        3). $\mathbf{w}_i^{'} = \mathbf{H}\mathbf{v}_i$, $\alpha_i = \mathbf{w}_i^{'\top}\mathbf{v}_i$, $\mathbf{w}_i = \mathbf{w}_i^{'} - \alpha_i\mathbf{v}_i - \beta_j\mathbf{v}_{i-1}$;
    **end for**
    4). $\mathbf{T}_k(i,i) = \alpha_i$, $i = 1, \ldots, m$, $\mathbf{T}_k(i,i+1) = \mathbf{T}_k(i+1,i) = \beta_i$, $i = 1, \ldots, m-1$.
    5). $A_k = \mathbf{e}_1^\top\mathbf{T}_k^\alpha\mathbf{e}_1$, $B_k = \mathbf{g}_k^\top\mathbf{H}\mathbf{g}_k$;
**end for**
**Return:** $H_\alpha(\mathbf{H}) = \frac{1}{1-\alpha}\log\frac{\sum_{k=1}^{l}A_k}{(\sum_{k=1}^{l}B_k)^\alpha}$

---

## 5 Correlation between Rényi Sharpness and Generalization

In this section, we estimate the Rényi entropy via Algorithm 1, and validate that Rényi entropy is strongly correlated with generalization.

### 5.1 Task

We evaluate the correlation between Rényi sharpness and generalization on: ResNet18/34 (He et al., 2016), and Simple Vision Transformer architecture from the `vit-pytorch` library on CIFAR10

(Krizhevsky & Hinton, 2009), ResNet18/34 on CIFAR100, and ResNet18 on TinyImageNet (Le & Yang, 2015). We vary the learning rate, optimization algorithm, and the weight decay strength to generate different local minima, and then estimate the layer-wise and global Rényi sharpness. More details can be found in Appendix J. We compare with the classical Hessian-based flatness measures using the trace of the loss-Hessian, the Fisher-Rao norm(Liang et al., 2019), the PAC-Bayes flatness measure that performed best in the extensive study of Jiang et al. (2019), the Frobenius norm of the weights, and the sharpness defined in SAM (Foret et al., 2020) and ASAM (Kwon et al., 2021). Notably, the sharpness defined in ASAM (Kwon et al., 2021) has been empirically shown by Andriushchenko et al. (2023), on larger-scale datasets and models, to have little or no correlation with generalization performance. The definition and detailed implementation of those measures can be found in Appendix I, and the hyperparameter $\rho$ in SAM and ASAM is searched over $\{10^{-6}, 3\times10^{-6}, 10^{-5}, 3\times10^{-5}, 10^{-4}, 3\times10^{-4}, 10^{-3}, 3\times10^{-3}, 10^{-2}, 3\times10^{-2}, 10^{-1}, 0.3, 1\}$.

To detect correlation, we follow the previous works by Dziugaite et al. (2020); Jiang et al. (2019); Kwon et al. (2021); Andriushchenko et al. (2023) and use the Kendall rank correlation coefficient:

$$\tau(\mathbf{x}, \mathbf{y}) = \frac{2}{N(N-1)} \sum_{i<j} \text{sign}(x_i - x_j)\text{sign}(y_i - y_j) \qquad (10)$$

where $\mathbf{x}, \mathbf{y} \in \mathbb{R}^N$ are vectors of generalization gap and sharpness values for $N$ different models. We follow the approach of Andriushchenko et al. (2023) by comparing sharpness and generalization within the same model architecture. This contrasts with prior works such as Dziugaite et al. (2020) and Jiang et al. (2019), which focus on comparisons across models with varying width or depth. We always evaluate sharpness on the same training points taken without any data augmentations, while the data augmentation tools are allowed in training.

## 5.2 CORRELATION BETWEEN RÉNYI SHARPNESS AND GENERALIZATION

After training with a range of hyperparameters, we estimate Rényi sharpness and compute the Kendall rank correlation between Rényi entropy and the generalization gap (defined as the difference between training and test loss). We vary $\alpha$ and plot the sharpness that attains the highest correlation coefficient. Fig. 2 reports these correlations on CIFAR-10 with ResNet-18. The "layer 1" through "all layer" subplots correspond to Rényi sharpness; the remaining subplots show alternative metrics. As evident in Fig. 2, Rényi sharpness aligns closely with generalization performance and outperforms the other measures in capturing the generalization gap.

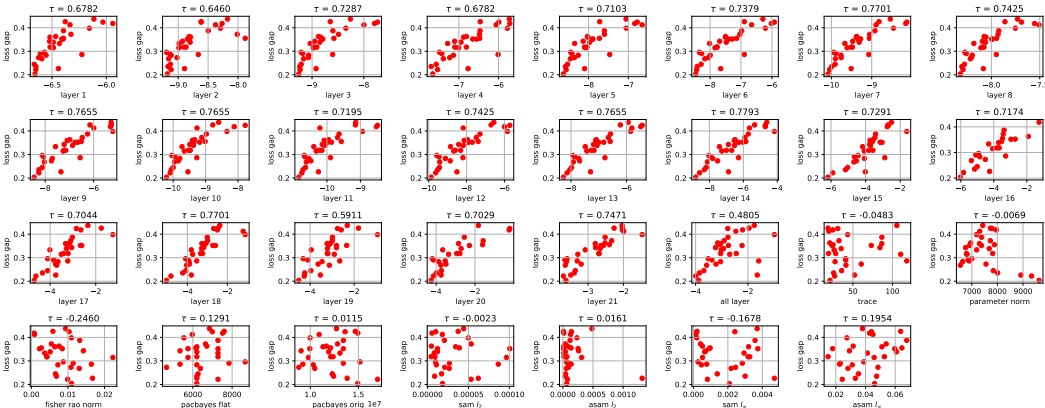

Figure 2: ResNet18 on CIFAR10, The layer 1 to all layer subplots correspond to the Rényi sharpness measure. Rényi sharpness is strongly correlated with generalization than the other measures.

Owing to page limits, we present the remaining tasks in a compact format that aggregates all statistics into a single panel (Fig. 3). As shown in Fig. 3, Rényi sharpness is strongly correlated with generalization. Full per-task figures in the style of Fig. 2 are provided in the Appendix K.2.

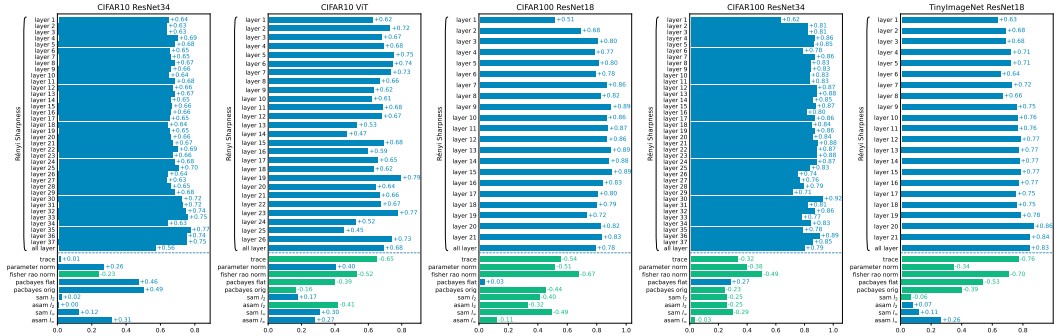

Figure 3: Kendall correlations on various tasks. Signed coefficients are mapped to 0–1 (blue = positive, green = negative). Rényi sharpness shows the strongest correlation with generalization than other sharpness measures.

## 6 REGULARIZATION BY RÉNYI SHARPNESS

In this section, we propose to use Rényi sharpness as a regularizer during training, i.e. the Rényi Sharpness Aware Minimization algorithm. To reduce the complexity, in practice we will employ an approximation of the Rényi sharpness.

### 6.1 RÉNYI REGULARIZATION AND RÉNYI SHARPNESS AWARE MINIMIZATION (RSAM)

If the original form Rényi sharpness was used for regularizer, it would require multiple cycles of gradient descent, thus increasing the computational complexity by dozens of times, as compared with the traditional training method. To reduce the computational burden, we will resort to the approximations of Rényi sharpness. In specific, following the work by Khan et al. (2018); Kim et al. (2022), we will employ the gradient magnitude as an approximation of the Hessian matrix:

$$\mathbf{H} \approx \mathbf{GM} = \left[ \mathrm{Diag}(\frac{1}{N} \sum_{i=1}^{N} \nabla_{\boldsymbol{\theta}} l(\boldsymbol{\theta}, \mathbf{x}_i, \mathbf{y}_i)) \right]^2 \tag{11}$$

Consequently, the Rényi sharpness can be approximated by

$$-H_\alpha(\mathbf{H}) \approx -H_\alpha(\mathbf{GM}) = -\frac{1}{1-\alpha} \log \frac{\sum_j |g_j|^{2\alpha}}{(\sum_j g_j^2)^{\alpha}} \tag{12}$$

where $\mathbf{g}$ is the gradient vector computed by the optimization algorithms, and $g_j = \frac{1}{N} \sum_{i=1}^{N} \nabla_{\theta_j} l(\boldsymbol{\theta}, \mathbf{x}_i, \mathbf{y}_i)$ is the element in $\mathbf{g}$. Thus we can use $-\mathrm{sign}(1-\alpha) \frac{\sum_j |g_j|^{2\alpha}}{(\sum_j g_j^2)^{\alpha}}$ as the **Rényi regularizer**. To avoid the memory usage and compute cost caused by explicitly computing the gradient with computational graph preserved (e.g., `create_graph=True` in PyTorch), we consider minimizing the following objective instead:

$$L(\boldsymbol{\theta} + \boldsymbol{\epsilon}) = L(\boldsymbol{\theta} - \rho \cdot \mathrm{sign}(1-\alpha) \cdot \frac{\sum_j |g_j|^{2\alpha}}{(\sum_j g_j^2)^{\alpha+1}} \mathbf{g}^\top) \tag{13}$$

Eq. 13 can be expanded as follows:

$$L(\boldsymbol{\theta} + \boldsymbol{\epsilon}) \approx L(\boldsymbol{\theta}) - \rho \cdot \mathrm{sign}(1-\alpha) \cdot \frac{\sum_j |g_j|^{2\alpha}}{(\sum_j g_j^2)^{\alpha+1}} \mathbf{g}^\top \mathbf{g} = L(\boldsymbol{\theta}) - \rho \cdot \mathrm{sign}(1-\alpha) \cdot \frac{\sum_j |g_j|^{2\alpha}}{(\sum_j g_j^2)^{\alpha}} \tag{14}$$

Thus, optimizing Eq. 13 is approximately optimizing the original loss with Rényi regularizer, namely, Rényi sharpness-aware minimization (RSAM, see Algorithm 2). We observe that penalizing a single layer (e.g., the final layer) typically requires extending training for more epochs to achieve strong generalization, unless multiple layers are optimized concurrently. Given the combinatorial cost of tuning layer-specific regularization strengths, we adopt a single global Rényi regularizer applied across all layers. Appendix F establishes that optimizing this global objective implies optimizing the layer-wise objectives as well.

Moreover, it is observed that incorporating the approximate Hessian matrix and penalizing Rényi sharpness at the early stages of training introduces substantial instability. To mitigate this effect,

we first train with plain SGD and adapt the warm-up length based on validation accuracy. For easy tasks, five epochs suffice to attain high accuracy, so the SGD warm-up is capped at five epochs. For harder tasks such as TinyImageNet, we defer switching to RSAM until the validation Top-1 exceeds 30%, which typically occurs around epoch 20. The discussion and comparison with other related SAM variants can be found in the Appendix H.

## 6.2 COMPARISON BETWEEN RSAM AND OTHER SAM ALGORITHMS

We now apply our sharpness measure as a regularizer to train neural networks. We consider the image classification tasks involving the CIFAR10/100 and TinyImageNet datasets. Various convolutional neural networks such as ResNet, and WideResNet (Zagoruyko & Komodakis, 2016) are used for CIFAR10/100 experiments. We also evaluated performance by fine-tuning a ViT-B-16 model pre-trained on ImageNet for CIFAR-10 and CIFAR-100. We used the checkpoint provided by Py-Torch's official repository. For comparison, we consider the sharpness-aware minimization (SAM) method, the adaptive SAM (ASAM) method, an extension of SAM to involve the scale-invariance, the Eigen-SAM (Luo et al., 2024) method, which regularizes the top Hessian eigenvalue, the Fisher SAM (Kim et al., 2022) method which minimize sharpness under the Riemannian metric, and the Sparse SAM Mi et al. (2022) which mask the sharpness to speed up SAM algorithm. More details are provided in Appendix J.3.2.

Table 1: Test accuracies (avg. ± standard error) for SGD/SAM/ASAM/Eigen-SAM/FSAM/RSAM.

| Dataset | Model | SGD(%) | SAM(%) | ASAM(%) | Eigen-SAM(%) | FSAM(%) | SSAM(%) | OURS(%) |
|---|---|---|---|---|---|---|---|---|
| **CIFAR10** | ResNet20 | $92.68^{\pm0.25}$ | $93.44^{\pm0.07}$ | $93.62^{\pm0.16}$ | $93.24^{\pm0.20}$ | $93.54^{\pm0.12}$ | $93.44^{\pm0.14}$ | $\mathbf{93.69^{\pm0.12}}$ |
| | ResNet56 | $94.24^{\pm0.23}$ | $94.96^{\pm0.19}$ | $95.12^{\pm0.08}$ | $94.96^{\pm0.10}$ | $95.17^{\pm0.05}$ | $95.15^{\pm0.12}$ | $\mathbf{95.26^{\pm0.12}}$ |
| | WideResNet-28-10 | $96.36^{\pm0.08}$ | $96.95^{\pm0.05}$ | $96.79^{\pm0.10}$ | $96.78^{\pm0.06}$ | $96.96^{\pm0.06}$ | $96.96^{\pm0.04}$ | $\mathbf{97.13^{\pm0.06}}$ |
| **CIFAR100** | ResNet20 | $69.12^{\pm0.17}$ | $70.53^{\pm0.30}$ | $70.73^{\pm0.14}$ | $70.51^{\pm0.20}$ | $70.57^{\pm0.32}$ | $70.14^{\pm0.16}$ | $\mathbf{70.91^{\pm0.25}}$ |
| | ResNet56 | $72.60^{\pm0.34}$ | $74.86^{\pm0.23}$ | $75.20^{\pm0.29}$ | $74.80^{\pm0.32}$ | $74.91^{\pm0.21}$ | $75.42^{\pm0.18}$ | $\mathbf{75.71^{\pm0.18}}$ |
| | WideResNet-28-10 | $81.47^{\pm0.18}$ | $83.55^{\pm0.14}$ | $83.56^{\pm0.11}$ | $82.81^{\pm0.08}$ | $83.48^{\pm0.14}$ | $83.47^{\pm0.09}$ | $\mathbf{83.67^{\pm0.09}}$ |
| **TinyImageNet** | ResNet50 | $59.62^{\pm1.51}$ | $60.70^{\pm0.70}$ | $62.56^{\pm0.25}$ | - | $61.21^{\pm0.64}$ | - | $\mathbf{63.33^{\pm0.27}}$ |

Table 2: Test accuracy for fine-tuning ViT-B-16 pretrained on ImageNet-1K on CIFAR-10 and CIFAR-100.

| Dataset | Model | SGD(%) | SAM(%) | ASAM(%) | FSAM(%) | OURS(%) |
|---|---|---|---|---|---|---|
| **CIFAR10** | ViT-B-16 | $98.06^{\pm0.09}$ | $98.50^{\pm0.05}$ | $98.39^{\pm0.05}$ | $98.42^{\pm0.11}$ | $\mathbf{98.59^{\pm0.03}}$ |
| **CIFAR100** | ViT-B-16 | $88.27^{\pm0.15}$ | $89.38^{\pm0.04}$ | $88.78^{\pm0.33}$ | $89.41^{\pm0.11}$ | $\mathbf{89.58^{\pm0.07}}$ |

We provide the averages and standard errors of the test accuracies obtained from five runs of each method in Table 1 and Table 2. As can be seen from the table, one can confirm that the generalization performance of SGD is significantly improved with our regularizer. Furthermore, our method outperforms the SAM, ASAM, and Eigen-SAM methods. Although our method outperforms ASAM overall, the margin is modest on certain tasks. We hypothesize this gap arises because we currently employ an approximate surrogate of the Rényi sharpness, introduced for computational efficiency. We expect further improvements if the exact Rényi sharpness can be used as the regularizer (or if a tighter estimator becomes feasible), and we leave this as a promising direction for future work. Since we first warm up with plain SGD before switching to RSAM, we did not adjust RSAM's epoch budget to equalize total compute across methods; instead, we fixed the total number of epochs. Consequently, given a fixed compute budget, RSAM would be allowed to run more epochs and thus expected to improve further the performance.

## 7 CONCLUSION

In this work, we propose a novel measure of sharpness – Rényi sharpness, which is defined as the negative Rényi entropy of the loss Hessian. By leveraging the reparameterization invariance of Rényi sharpness and the fact that data perturbations can be absorbed into the weight perturbations, we develop several generalization bounds based on the Rényi sharpness. Extensive experiments demonstrate a strong correlation between the Rényi sharpness and generalization. Furthermore, we propose the Rényi Sharpness-Aware Minimization (RSAM) algorithm, which penalizes Rényi sharpness during training. Experimental results demonstrate that RSAM outperforms all existing sharpness-aware minimization methods, across multiple tasks.

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

## A   ORGANIZATION OF APPENDIX

The appendix is organized as follows:

- Sec. A: an overview of the organization of the appendix.
- Sec. B: detailed proof of the PAC Bayesian generalization bound under multiplicative perturbation (Theorem 3.1).
- Sec. C: detailed proof of the PAC Bayesian generalization bound for Rényi entropy motivated by (Jia & Su, 2020) (Theorem 3.2).
- Sec. D: detailed proof of the PAC Bayesian generalization bound for Rényi entropy (Theorem 3.3).
- Sec. E: detailed proof of the reparameterization invaricance of Rényi entropy (Proposition 2.2).
- Sec. F: detailed proof of optimizing global Rényi regularization implies optimizing layer-wise Rényi regularization.
- Sec. G: a proof of arbitrary trace rescaling under fixed normalized spectrum.
- Sec. H: detailed discussion and comparison with Rényi sharpness-aware minimization and some related sharpness-aware minimization variants.
- Sec. I: detailed description and definition of the baseline sharpness measures.
- Sec. J: detailed descriptions of the datasets, models, hyper-parameter choices used in our experiments, including correlation experiments and the sharpness-aware minimization experiments.
- Sec. K: This section presents the Hessian spectrum which determine the Rényi order choice and the correlation coefficient under different Rényi order $\alpha$. The correlation comparison between the Rényi sharpness and other sharpness measures across multiple tasks is also included.
- Sec. L: limitations of our assumptions and theoretical results.
- Sec. M: broader impacts statement of this research.

# B  PAC BAYESIAN GENERALIZATION BOUND UNDER MULTIPLICATIVE PERTURBATION

Below, we state a generalization bound based on multiplicative perturbation.

**Theorem B.1** *For any $\rho > 0$, and a training set $\mathcal{S}$ draw from the distribution $\mathcal{D}$, we assumed that $L(\mathcal{D}, \boldsymbol{\theta}) \leq L(\mathcal{D}, \boldsymbol{\theta} + \boldsymbol{\delta})$, where $\boldsymbol{\delta}$ is the pertubation to the weights, $\mathcal{S}(\mathbf{A}, \rho) = \{(\mathbf{x} + \rho\mathbf{A}\mathbf{x}, \mathbf{y}) | (\mathbf{x}, \mathbf{y}) \in \mathcal{S}\}$ and $\mathbf{A}$ is a orthogonal matrix sampled under Haar measure, i.e., uniform on $\mathcal{O}(d)$. With probability $1 - \epsilon$,*

$$L(\mathcal{D}, \boldsymbol{\theta}) \leq \mathbb{E}_{\mathbf{A}}[L(\mathcal{S}(\mathbf{A}, \rho), \boldsymbol{\theta})] + C\sqrt{\frac{\log\frac{1}{\epsilon}}{2n}}$$

The condition $L(\mathcal{D}, \boldsymbol{\theta}) \leq L(\mathcal{D}, \boldsymbol{\theta} + \boldsymbol{\delta})$ means that adding perturbation to weights should not decrease the test error. This is expected to hold in practice for the final solution but does not necessarily hold for any $\boldsymbol{\theta}$.

$Proof$. Based on the Hoeffding's inequaliy, which is stated as follows:

**Theorem B.2 (Hoeffding's inequaliy)** *Let $U_1, \ldots, U_n$ beindependent random variables taking values in an interval $[a, b]$. Then, for any $t \in \mathbb{R}$,*

$$\mathbb{E}\left[e^{t\sum_{i=1}^{n}[\mathbb{E}U_i - U_i]}\right] \leq e^{\frac{nt^2(b-a)^2}{8}} \tag{15}$$

Let $U_i = \mathbb{E}_{\mathbf{A}}\left[l(f(\boldsymbol{\theta}, \mathbf{x}_i + \rho\mathbf{A}\mathbf{x}_i), y_i)\right]$, thus $\mathbb{E}U_i = \mathbb{E}_{\mathbf{A}}[L(\mathcal{D}(\mathbf{A}, \rho))]$ and $\frac{1}{n}\sum_{i=1}^{n} U_i = \mathbb{E}_{\mathbf{A}}[L(\mathcal{S}(\mathbf{A}, \rho))]$, where $\mathcal{D}(\mathbf{A}, \rho) = \{(\mathbf{x} + \rho\mathbf{A}\mathbf{x}, \mathbf{y}) | (\mathbf{x}, \mathbf{y}) \in \mathcal{D}\}$, $\mathcal{S}(\mathbf{A}, \rho) = \{(\mathbf{x} + \rho\mathbf{A}\mathbf{x}, \mathbf{y}) | (\mathbf{x}, \mathbf{y}) \in \mathcal{S}\}$ and $\mathbf{A}$ is a orthogonal matrix sampled under Haar measure, i.e., uniform on $\mathcal{O}(d)$. Consequently, we have

$$\mathbb{E}_{\mathcal{S}}\left[e^{tn\left[\mathbb{E}_{\mathbf{A}}[L(\mathcal{D}(\mathbf{A}, \rho))] - \mathbb{E}_{\mathbf{A}}[L(\mathcal{S}(\mathbf{A}, \rho))]\right]}\right] \leq e^{\frac{nt^2C^2}{8}} \tag{16}$$

For any $s$,

$$\mathbb{P}_{\mathcal{S}}\left(\mathbb{E}_{\mathbf{A}}[L(\mathcal{D}(\mathbf{A}, \rho))] - \mathbb{E}_{\mathbf{A}}[L(\mathcal{S}(\mathbf{A}, \rho))] > s\right) \tag{17}$$

$$= \mathbb{P}_{\mathcal{S}}\left(e^{nt\left[\mathbb{E}_{\mathbf{A}}[L(\mathcal{D}(\mathbf{A}, \rho))] - \mathbb{E}_{\mathbf{A}}[L(\mathcal{S}(\mathbf{A}, \rho))]\right]} > e^{nts}\right) \tag{18}$$

$$\leq \frac{e^{nt\left[\mathbb{E}_{\mathbf{A}}[L(\mathcal{D}(\mathbf{A}, \rho))] - \mathbb{E}_{\mathbf{A}}[L(\mathcal{S}(\mathbf{A}, \rho))]\right]}}{e^{nts}} \qquad \text{Markov's inequality} \tag{19}$$

$$\leq e^{\frac{nt^2C^2}{8} - nts} \tag{20}$$

Consequently,

$$\mathbb{P}_{\mathcal{S}}\left(\mathbb{E}_{\mathbf{A}}[L(\mathcal{D}(\mathbf{A}, \rho))] > \mathbb{E}_{\mathbf{A}}[L(\mathcal{S}(\mathbf{A}, \rho))] + s\right) \leq e^{\frac{nt^2C^2}{8} - nts} \tag{21}$$

when $t = 4s/C^2$, $nt^2C^2/8 - nts$ is minimized, thus,

$$\mathbb{P}_{\mathcal{S}}\left(\mathbb{E}_{\mathbf{A}}[L(\mathcal{D}(\mathbf{A}, \rho))] > \mathbb{E}_{\mathbf{A}}[L(\mathcal{S}(\mathbf{A}, \rho))] + s\right) \leq e^{\frac{-2ns^2}{C^2}} \tag{22}$$

let $\epsilon = e^{\frac{-2ns^2}{C^2}}$, we have

$$\mathbb{P}_{\mathcal{S}}\left(\mathbb{E}_{\mathbf{A}}[L(\mathcal{D}(\mathbf{A}, \rho))] > \mathbb{E}_{\mathbf{A}}[L(\mathcal{S}(\mathbf{A}, \rho))] + C\sqrt{\frac{\log\frac{1}{\epsilon}}{2n}}\right) \leq \epsilon \tag{23}$$

consequently,

$$\mathbb{P}_{\mathcal{S}}\left(\mathbb{E}_{\mathbf{A}}[L(\mathcal{D}(\mathbf{A},\rho))] \leq \mathbb{E}_{\mathbf{A}}[L(\mathcal{S}(\mathbf{A},\rho))] + C\sqrt{\frac{\log\frac{1}{\epsilon}}{2n}}\right) > 1 - \epsilon \qquad (24)$$

For any multiplicative perturbation, the perturbation in the input space can be fully transformed into weight space, which means $\mathbb{E}_{\mathbf{A}}[L(\mathcal{D}(\mathbf{A},\rho))] = L(\mathcal{D},\boldsymbol{\theta}+\boldsymbol{\delta})$, where $\boldsymbol{\delta}$ obeys some unknown distribution. Consider the assumption that $L(\mathcal{D},\boldsymbol{\theta}) \leq L(\mathcal{D},\boldsymbol{\theta}+\boldsymbol{\delta})$, we have

$$\mathbb{P}_{\mathcal{S}}\left(L(\mathcal{D},\boldsymbol{\theta}) \leq \mathbb{E}_{\mathbf{A}}[L(\mathcal{S}(\mathbf{A},\rho))] + C\sqrt{\frac{\log\frac{1}{\epsilon}}{2n}}\right) \geq 1 - \epsilon \qquad (25)$$

**Discussion:** The idea about multiplicative perturbation under haar measure is also reported in Petzka et al. (2021), whose sharpness is define by the Hessian matrix of the loss function w.r.t a full connect layer's weights, but their follow-up results need to split the Hessian matrix into multiple blocks and compute the corresponding traces individually, which proposes a huge computation burden when dealing with a big layer, thus they only compute the sharpness of last layer in small model. Contrary to deriving a bound via multiplicative perturbations like Petzka et al. (2021), this section aims to show that the dependency between the real and empirical data distributions can be transformed to a weight perturbation of an individual layer, enabling the application of Theorem 3.2 and 3.3 to study the corresponding layer-wise spectrum. Unlike the global spectrum, the layer-wise spectrum is more likely to be invariant under reparameterization. In Section 4, we prove the invariance of the Rényi entropy in Theorem 3.2 and 3.3. Since the invariance conditions for the normalized global spectrum are much more restrictive, Theorem 3.2 and 3.3 only apply to the layer-wise Rényi entropy. Nevertheless, in Section 5 we empirically observe that the Rényi entropy of the global spectrum is still correlated with generalization. We attribute this phenomenon to the fact that the global spectrum is composed of the layer-wise spectra; hence, when the layer-wise spectra exhibit strong correlations, the global spectrum also demonstrates significant correlations.

**Corollary B.3** *For any $\rho > 0$, and a training set $\mathcal{S}$ draw from the distribution $\mathcal{D}$, we assumed that $L(\mathcal{D},\boldsymbol{\theta}) \leq L(\mathcal{D},\boldsymbol{\theta}+\boldsymbol{\delta})$, where $\boldsymbol{\theta}$ is the pertubation to the weights, $\mathcal{S}(\mathbf{A},\rho) = \{(\mathbf{x}+\rho\mathbf{A}\mathbf{x},\mathbf{y})|(\mathbf{x},\mathbf{y}) \in \mathcal{S}\}$ and $\mathbf{A}$ is a orthogonal matrix sampled under Haar measure, i.e., uniform on $\mathcal{O}(d)$. With probability $1-\epsilon$, we have*

$$L(\mathcal{D},\boldsymbol{\theta}) \leq \mathbb{E}_{\mathbf{A}}[L(\mathcal{S}(\mathbf{A},\rho),\boldsymbol{\theta})] + C\sqrt{\frac{\log\frac{1}{\epsilon}}{2n}}$$

## C  PAC BAYESIAN GENERALIZATION BOUND FOR RÉNYI ENTROPY MOTIVATED BY (JIA & SU, 2020)

In this section, we will propose a generalization bound based on the Rényi entropy of the Hessian spectrum of the loss function with respect to the weights.

**Proposition C.1** *Given a training set $\mathcal{S}$ with size $N$ draw from the data distribution $\mathcal{D}$ and a loss function $L(\cdot, \cdot) \in [0, 1]$, a layer-wise local minimum $\theta^* \in \mathbb{R}^n$ is isolated and unique in its neighborhood $\mathcal{M}(\theta^*)$ whose volume $V$ is sufficiently small, pick a uniform prior $\mathcal{P}$ over $\theta \in \mathcal{M}(\theta^*)$ and pick the posterior $\mathcal{Q}$ of density $q(\theta) \propto e^{-|L_0 - L(\mathcal{S}, \theta)|}$ with $L_0 = L(\mathcal{S}, \theta^*)$. For any $\delta \in (0, 1]$ and $\alpha > 0, \alpha \neq 1$, we have with probability at least $1 - \delta$ that:*

$$\mathbb{E}_{\mathcal{Q}}[L(\mathcal{D}, \theta)] \leq \mathbb{E}_{\mathcal{Q}}[L(\mathcal{S}, \theta)] + 2\sqrt{\frac{2L_0 + 2\mathcal{A} + \log\frac{2N}{\delta}}{N - 1}} \tag{26}$$

*where $\mathcal{A} = \frac{1}{4\pi e} n V^{\frac{2}{n}} \pi^{\frac{1}{n}} \exp\{\frac{-H_\alpha(\mathbf{H}) + A}{n}\}$, and $A > 0$ is the constant item. $\mathbf{H}$ is the Hessian matrix of loss function w.r.t. $\theta^*$.*

*Proof.* Using PAC-Bayesian generalization bound proved by (Jia & Su, 2020):

**Theorem C.2** *Given a training set $\mathcal{S}$ with size $N$ draw from the data distribution $\mathcal{D}$ and a loss function $L(\cdot, \cdot) \in [0, 1]$, a local minimum $\theta^* \in \mathbb{R}^n$ is isolated and unique in its neighborhood $\mathcal{M}(\theta^*)$ whose volume $V$ is sufficiently small, pick a uniform prior $\mathcal{P}$ over $\theta \in \mathcal{M}(\theta^*)$ and pick the posterior $\mathcal{Q}$ of density $q(\theta) \propto e^{-|L_0 - L(\mathcal{S}, \theta)|}$ with $L_0 = L(\mathcal{S}, \theta^*)$. For any $\delta \in (0, 1]$, we have with probability at least $1 - \delta$ that:*

$$\mathbb{E}_{\mathcal{Q}}[L(\mathcal{D}, \theta)] \leq \mathbb{E}_{\mathcal{Q}}[L(\mathcal{S}, \theta)] + 2\sqrt{\frac{2L_0 + 2\mathcal{A} + \log\frac{2N}{\delta}}{N - 1}} \tag{27}$$

*where $\mathcal{A} = \frac{1}{4\pi e} n V^{\frac{2}{n}} \pi^{\frac{1}{n}} \exp\{\frac{\log|\mathbf{H}|}{n}\}$, and $\mathbf{H}$ is the Hessian matrix of loss function w.r.t. $\theta^*$.*

Next, we will utilize the Rényi entropy to bound the $\log|\mathbf{H}|$ term.

$$\log|\mathbf{H}| = \sum_{i=1}^{n} \log\lambda_i \tag{28}$$

$$= \sum_{i=1}^{n} \log(\text{Tr}(\mathbf{H})\frac{\lambda_i}{\text{Tr}(\mathbf{H})}) \tag{29}$$

$$= n\log\text{Tr}(\mathbf{H}) + \sum_{i=1}^{n} \log\frac{\lambda_i}{\text{Tr}(\mathbf{H})} \tag{30}$$

let $p_i = \frac{\lambda_i}{\text{Tr}(\mathbf{H})}$, we have for $\alpha > 1$

$$\sum_{i=1}^{n} \log p_i \leq \sum_{i=1}^{n} p_i \log p_i \tag{31}$$

$$= -H_1(\mathbf{p}) \tag{32}$$

$$\leq -H_\alpha(\mathbf{p}) \qquad \text{monotonicity of Rényi entropy} \tag{33}$$

consequently,

$$\sum_{i=1}^{n} \log\frac{\lambda_i}{\text{Tr}(\mathbf{H})} \leq -H_\alpha(\mathbf{H}) \tag{34}$$

Thus for $\alpha > 1, 1 - \alpha < 0$, larger entropy means a smaller $\sum_{i=1}^{n} \log\frac{\lambda_i}{\text{Tr}(\mathbf{H})}$.

When $0 < \alpha < 1$, considering Jensen's inequality, we have

$$\frac{1}{n}\sum_{i=1}^{n} p_i^\alpha \leq \left(\frac{1}{n}\sum_{i=1}^{n} p_i\right)^\alpha = \left(\frac{1}{n}\right)^\alpha, \tag{35}$$

Thus,

$$\sum_{i=1}^{n} p_i^{\alpha} \leq n^{1-\alpha}. \tag{36}$$

Using the AM-GM inequality, we will get

$$\Big(\prod_{i=1}^{n} p_i\Big)^{1/n} \leq \frac{1}{n}\sum_{i=1}^{n} p_i = \frac{1}{n} \tag{37}$$

consequently,

$$\prod_{i=1}^{n} p_i \leq n^{-n}. \tag{38}$$

Combining equation 36 and equation 38, we have

$$\Big(\prod_{i=1}^{n} p_i\Big)\Big(\sum_{i=1}^{n} p_i^{\alpha}\Big)^{1/(1-\alpha)} \leq n^{-n}\big(n^{1-\alpha}\big)^{1/(1-\alpha)} = n^{1-n} \leq 1. \tag{39}$$

Thus we have

$$\sum_{i=1}^{n} \log p_i + \frac{1}{1-\alpha}\log\Big(\sum_{i=1}^{n} p_i^{\alpha}\Big) \leq 0 \iff \sum_{i=1}^{n} \log p_i \leq -H_{\alpha}(p). \tag{40}$$

consequently,

$$\sum_{i=1}^{n} \log\frac{\lambda_i}{\mathrm{Tr}(\mathbf{H})} \leq -H_{\alpha}(\mathbf{H}) \tag{41}$$

Combine Eq.41, Eq.34, we have for all $\alpha > 0, \alpha \neq 1$,

$$\sum_{i=1}^{n} \log\frac{\lambda_i}{\mathrm{Tr}(\mathbf{H})} \leq -H_{\alpha}(\mathbf{H}) \tag{42}$$

Now we apply Eq.42 to Eq.30 and Eq.27:

$$\mathbb{E}_{\mathcal{Q}}[L(\mathcal{D},\theta)] \leq \mathbb{E}_{\mathcal{Q}}[L(\mathcal{S},\theta)] + 2\sqrt{\frac{2L_0 + 2\mathcal{A} + \log\frac{2N}{\delta}}{N-1}} \tag{43}$$

where $\mathcal{A} = \frac{1}{4\pi e}nV^{\frac{2}{n}}\pi^{\frac{1}{n}}\exp\{\frac{n\log\mathrm{Tr}(\mathbf{H})-H_{\alpha}(\mathbf{H})}{n}\}$, and $\mathbf{H}$ is the Hessian matrix of loss function w.r.t. $\theta^*$.

We decompose the bound as

$$\mathrm{Gen}(f_{\theta}) \leq g(A(\theta) + B(\theta) + C), \qquad A(\theta) = \mathrm{Tr}(\mathbf{H}_{\theta}), \tag{44}$$

where $A(\theta)$ is parameterization-dependent while $B(\theta)$ is reparameterization-invariant and $C$ is the constant. Let $[\theta] = \{S\theta : S \in \mathcal{G}\}$ denote the reparameterization equivalence class that leaves the predictor $f_{\theta}$ unchanged (e.g., reparameterization induced by homogeneous activation function). Since $A(\theta)$ is not invariant and can be arbitrarily altered within $[\theta]$, thus it is not an identifiable property of $f_{\theta}$.

To remove this ambiguity, we define a canonical projection $\Pi : [\theta] \to [\theta]$ that selects, for every $\theta$, a representative $\theta^{\star} = \Pi(\theta) \in [\theta]$ satisfying

$$A(\theta^{\star}) = A_0, \tag{45}$$

where $A_0$ is a constant independent of the underlying function $f$. Because $B$ is invariant under reparameterization, we have $B(\theta^{\star}) = B(\theta) =: B(f)$. Therefore, for every function $f$,

$$\mathrm{Gen}(f) = \mathrm{Gen}(f_{\theta^{\star}}) \leq g(A(\theta^{\star}) + B(\theta^{\star})) = g(A_0 + B(f)). \tag{46}$$

Hence, *up to an additive constant $A_0$ determined by the canonical projection*, generalization is governed by the reparameterization-invariant term $B$. Accordingly, we absorb the trace term into the constant $A$, and obtain $\mathcal{A} = \frac{1}{4\pi e}nV^{\frac{2}{n}}\pi^{\frac{1}{n}}\exp\{\frac{-H_{\alpha}(\mathbf{H})+A}{n}\}$. The reparameterization invariance of the Rényi entropy is proved in Appendix E.

**Corollary C.3** *Given a training set $\mathcal{S}$ with size $N$ draw from the data distribution $\mathcal{D}$ and a loss function $L(\cdot, \cdot) \in [0, 1]$, a layer-wise local minimum $\theta^* \in \mathbb{R}^n$ is isolated and unique in its neighborhood $\mathcal{M}(\theta^*)$ whose volume $V$ is sufficiently small, pick a uniform prior $\mathcal{P}$ over $\theta \in \mathcal{M}(\theta^*)$ and pick the posterior $\mathcal{Q}$ of density $q(\theta) \propto e^{-|L_0 - L(\mathcal{S}, \theta)|}$ with $L_0 = L(\mathcal{S}, \theta^*)$. For any $\delta \in (0, 1]$ and $\alpha > 0, \alpha \neq 1$, we have with probability at least $1 - \delta$ that:*

$$\mathbb{E}_{\mathcal{Q}}[L(\mathcal{D}, \theta)] \leq \mathbb{E}_{\mathcal{Q}}[L(\mathcal{S}, \theta)] + 2\sqrt{\frac{2L_0 + 2\mathcal{A} + \log \frac{2N}{\delta}}{N - 1}} \tag{47}$$

*where $\mathcal{A} = \frac{1}{4\pi e} n V^{\frac{2}{n}} \pi^{\frac{1}{n}} \exp\{\frac{-H_\alpha(\mathbf{H}) + A}{n}\}$, and $A > 0$ is the constant item. $\mathbf{H}$ is the Hessian matrix of loss function w.r.t. $\theta^*$.*

# D    PAC BAYESIAN GENERALIZATION BOUND FOR RÉNYI ENTROPY

**Theorem D.1** *Given a training set $\mathcal{S}$ with $N$ samples draw from the data distribution $\mathcal{D}$ and a loss function $L(\cdot,\cdot)$, a layer-wise local minimum $\theta^* \in \mathbb{R}^n$. We assumed that $L(\mathcal{D},\theta^*) \leq L(\mathcal{D},\theta^* + \epsilon)$, where $\epsilon$ is the pertubation to the weights. Consider a prior uniform in a ball which contains the ellipsoid that satisfy $\{\,\theta : (\theta - \theta^*)^\top \mathbf{H}(\theta - \theta^*) \leq \rho^2\,\}$. Take the posterior uniform on this ellipsoid. For any $\delta \in (0,1]$ and $\alpha > 0, \alpha \neq 1$, we have with probability at least $1 - \delta$ that:*

$$L(\mathcal{D},\theta^*) \; \leq \; L(\mathcal{S},\theta^*) + \tfrac{n}{2(n+2)}\rho^2 + O(\varepsilon) + \sqrt{\frac{-\frac{1}{2}H_\alpha(\mathbf{H}) + \log\frac{2\sqrt{N}}{\delta} + C}{2(N-1)}}. \tag{48}$$

Where $A > 0$ is the constant term. The condition $L(\mathcal{D},\theta^*) \leq L(\mathcal{D},\theta^* + \epsilon)$ means that adding perturbation to weights should not decrease the test error. This is expected to hold in practice for the final solution but does not necessarily hold for any $\theta$.

*Proof.*

We recall the standard PAC-Bayes bound (e.g. McAllester (2003)): for any prior $P$ independent of the data, with probability at least $1 - \delta$ over the draw of the sample $S$ of size $N$, for any posterior $Q$ we have

$$\mathbb{E}_{\theta\sim Q}[L(\theta)] \; \leq \; \mathbb{E}_{\theta\sim Q}[\hat{L}_S(\theta)] \; + \; \sqrt{\frac{D_{\mathrm{KL}}(Q\|P) + \log\frac{2\sqrt{N}}{\delta}}{2(N-1)}}. \tag{49}$$

Suppose $\theta^*$ is a local minimum and in a sufficiently small neighborhood we have the quadratic approximation

$$\hat{L}_S(\theta) \; = \; \hat{L}_0 + \tfrac{1}{2}(\theta - \theta^*)^\top \mathbf{H}(\theta - \theta^*) + R_3(\theta), \qquad |R_3(\theta)| \leq \varepsilon, \tag{50}$$

with Hessian $\mathbf{H} \succ 0$. We now consider two different posterior distributions $Q$, both paired with a uniform prior $P$.

Fix $\rho > 0$ independent of $\mathbf{H}$. Define the ellipsoid

$$E_{\mathbf{H}}(\rho) = \{\,\theta : (\theta - \theta^*)^\top \mathbf{H}(\theta - \theta^*) \leq \rho^2\,\}.$$

We take $Q = \mathrm{Unif}(E_{\mathbf{H}}(\rho))$ and the prior $P = \mathrm{Unif}(B_R)$, the uniform distribution over a large Euclidean ball $B_R$ containing all such ellipsoids.

**Step 1. Empirical risk under $Q$.**    With the change of variables $y = \mathbf{H}^{1/2}(\theta - \theta^*)$, $Q$ becomes uniform on the ball $B_n(\rho)$. Then

$$\mathbb{E}_{\theta\sim Q}[(\theta - \theta^*)^\top \mathbf{H}(\theta - \theta^*)] = \mathbb{E}\|y\|^2 = \int_0^\rho r^2 f_R(r)\,dr = \int_0^\rho r^2 \cdot \frac{n\,r^{n-1}}{\rho^n}\,dr = \frac{n}{n+2}\rho^2.$$

Thus

$$\mathbb{E}_{\theta\sim Q}[\hat{L}_S(\theta)] = \hat{L}_0 + \tfrac{1}{2}\frac{n}{n+2}\rho^2 + O(\varepsilon),$$

which is a constant independent of $\mathbf{H}$.

**Step 2. KL divergence.**    The KL between uniform distributions is a log-volume ratio:

$$D_{\mathrm{KL}}(Q\|P) = \log\frac{\mathrm{Vol}(B_R)}{\mathrm{Vol}(E_{\mathbf{H}}(\rho))}.$$

The ellipsoid volume is

$$\mathrm{Vol}(E_{\mathbf{H}}(\rho)) = \mathrm{Vol}(B_n(1))\,\rho^n\,(\det \mathbf{H})^{-1/2}.$$

Hence

$$D_{\mathrm{KL}}(Q\|P) = \underbrace{\log \mathrm{Vol}(B_R) - \log \mathrm{Vol}(B_n(1)) - n\log\rho}_{\text{constant}} + \tfrac{1}{2}\log\det \mathbf{H}.$$

**Step 3. Bound.** Plugging into equation 49 gives

$$\mathbb{E}_{\theta \sim Q}[L(\theta)] \ \leq \ \hat{L}_0 + \tfrac{n}{2(n+2)}\rho^2 + O(\varepsilon) + \sqrt{\frac{\tfrac{1}{2}\log\det\mathbf{H} + \log\frac{2\sqrt{N}}{\delta} + \text{constant}}{2(N-1)}}.$$

Thus the only dependence on $\mathbf{H}$ is through $\tfrac{1}{2}\log\det H$.

The PAC-Bayes upper bound under quadratic approximation has the form

$$\mathbb{E}_{\theta \sim Q}[L(\theta)] \ \leq \ \text{constant} \ + \ f\!\left(\tfrac{1}{2}\log\det H\right)$$

where $f(\cdot)$ is the complexity term of the chosen PAC-Bayes bound. Thus the only dependence on the curvature $H$ comes from $\log\det H$; all trace-type terms are absorbed into constants. Take Taylor expansion at $\theta^*$, we assume that $L(\mathcal{D}, \boldsymbol{\theta}) \leq L(\mathcal{D}, \boldsymbol{\theta} + \boldsymbol{\delta})$, which means adding perturbation to weights should not decrease the test error, thus we have

$$L(\theta) \ \leq \ \hat{L}_S(\theta) + \text{constant} \ + \ f\!\left(\tfrac{1}{2}\log\det H\right)$$

Recall Eq.30, Eq. 42, and that Rényi entropy is reparameterization invariant, follow the poof in Appendix C, we have

$$\boxed{L(\theta) \ \leq \ \hat{L}_S(\theta) + \text{constant 1} \ + \ f(\text{constant 2} - H_\alpha(\mathbf{H}))}$$

**Corollary D.2** *Given a training set $\mathcal{S}$ with $N$ samples draw from the data distribution $\mathcal{D}$ and a loss function $L(\cdot, \cdot)$, a layer-wise local minimum $\theta^* \in \mathbb{R}^n$. We assumed that $L(\mathcal{D}, \theta^*) \leq L(\mathcal{D}, \theta^* + \epsilon)$, where $\epsilon$ is the pertubation to the weights. Consider a prior uniform in a ball which contains the ellipsoid that satisfy $\{\theta : (\theta - \theta^*)^\top \mathbf{H}(\theta - \theta^*) \leq \rho^2\}$. Take the posterior uniform on this ellipsoid. For any $\delta \in (0,1]$ and $\alpha > 0, \alpha \neq 1$, we have with probability at least $1 - \delta$ that:*

$$L(\mathcal{D}, \theta^*) \ \leq \ L(\mathcal{S}, \theta^*) + \tfrac{n}{2(n+2)}\rho^2 + O(\varepsilon) + \sqrt{\frac{-\tfrac{1}{2}H_\alpha(\mathbf{H}) + \log\frac{2\sqrt{N}}{\delta} + C}{2(N-1)}}. \tag{51}$$

## E    REPARAMETERIZATION (SCALING) INVARIANCE OF RÉNYI ENTROPY

Neural networks that use activation functions like ReLU or leaky ReLU exhibit **reparametrization-invariant properties**. Specifically, when scaling each layer's weights by a positive constant, the overall function computed by the network remains unchanged as long as the *product of all scaling factors equals one*.

For example, consider a network defined as

$$f(\mathbf{x}; \{\mathbf{W}_1, \ldots, \mathbf{W}_L\}) = \mathbf{W}_L \cdot \text{ReLU}(\mathbf{W}_{L-1} \cdots \text{ReLU}(\mathbf{W}_1 x)),$$

where $\mathbf{W}_l \in \mathbb{R}^{d_l \times d_{l-1}}$. If each weight matrix $\mathbf{W}_l$ is scaled by a positive constant $s_l > 0$, and the scaling factors satisfy $\prod_{l=1}^{L} s_l = 1$, then the output of the network remains unchanged for any input $\mathbf{x}$. The sharpness defined by Rényi entropy is invariant under this scaling trick:

**Proposition E.1** *Consider a $L$-layer feedforward neural network with positively homogeneous activation function $\sigma$ (i.e., $\sigma(c\mathbf{x}) = c\sigma(\mathbf{x})$ for all $c > 0$), and parameters $\{\mathbf{W}_1, \ldots, \mathbf{W}_L\}$. Let the network output be $f(\mathbf{x}) = \mathbf{W}_L \cdot \sigma(\mathbf{W}_{L-1} \cdots \sigma(\mathbf{W}_1 x))$, and let $\mathcal{L}(\boldsymbol{\theta})$ denote the loss function, where $\boldsymbol{\theta}$ denotes the weights of arbitrary layer, i.e., $\mathbf{W}_l$. Define the loss Hessian as $\mathbf{H}_{\boldsymbol{\theta}} = \nabla_{\boldsymbol{\theta}}^2 \mathcal{L}(\boldsymbol{\theta})$. Consider a layer-wise scaling transformation defined by $\tilde{\mathbf{W}}_l = c_l \mathbf{W}_l$, $c_l > 0$, with $\prod_{l=1}^{L} c_l = 1$. Let $\tilde{\boldsymbol{\theta}} = \tilde{\mathbf{W}}_l$ be the scaled parameters, and define $\mathbf{H}_{\tilde{\boldsymbol{\theta}}}$ as the corresponding Hessian. Then the spectrum-normalized Rényi entropy of $\mathbf{H}$ is invariant:*

$$H_\alpha(\mathbf{H}_{\tilde{\boldsymbol{\theta}}}) = H_\alpha(\mathbf{H}_{\boldsymbol{\theta}}), \quad \forall \alpha > 0, \ \alpha \neq 1. \tag{52}$$

*Proof.*

The network function $f(x)$ remains unchanged under the layer-wise scaling due to the positive homogeneity of the activation since $\prod c_l = 1$. Consequently, the loss $\mathcal{L}(\boldsymbol{\theta})$ is invariant:

$$\mathcal{L}(\tilde{\boldsymbol{\theta}}) = \mathcal{L}(\boldsymbol{\theta}). \tag{53}$$

Thus, the spectrum of $\mathbf{H}(\tilde{\theta})$ will undergo a scaling transformation:

$$\mathbf{H}_{\tilde{\boldsymbol{\theta}}} = c_l^2 \cdot \mathbf{H}_{\boldsymbol{\theta}}, \tag{54}$$

This implies that the eigenvalues $\{\tilde{\lambda}_i\}$ of $\mathbf{H}_{\tilde{\boldsymbol{\theta}}}$ satisfy:

$$\tilde{\lambda}_i = \frac{1}{c_l^2} \lambda_i \tag{55}$$

Then the normalized spectrum satisfies:

$$\tilde{p}_i = \frac{\tilde{\lambda}_i}{\sum_j \tilde{\lambda}_j} = \frac{\frac{1}{c_l^2} \lambda_i}{\frac{1}{c_l^2} \sum_j \lambda_j} = \frac{\lambda_i}{\sum_j \lambda_j} = p_i, \tag{56}$$

so the Rényi entropy remains unchanged:

$$H_\alpha(\mathbf{H}_{\tilde{\boldsymbol{\theta}}}) = \frac{1}{1-\alpha} \log\left(\sum_i \tilde{p}_i^\alpha\right) = \frac{1}{1-\alpha} \log\left(\sum_i p_i^\alpha\right) = H_\alpha(\mathbf{H}_{\boldsymbol{\theta}}). \tag{57}$$

**Corollary E.2** *Consider a $L$-layer feedforward neural network with positively homogeneous activation function $\sigma$ (i.e., $\sigma(c\mathbf{x}) = c\sigma(\mathbf{x})$ for all $c > 0$), and parameters $\{\mathbf{W}_1, \ldots, \mathbf{W}_L\}$. Let the network output be $f(\mathbf{x}) = \mathbf{W}_L \cdot \sigma(\mathbf{W}_{L-1} \cdots \sigma(\mathbf{W}_1 x))$, and let $\mathcal{L}(\boldsymbol{\theta})$ denote the loss function, where $\boldsymbol{\theta}$ denotes the weights of arbitrary layer, i.e., $\mathbf{W}_l$. Define the loss Hessian as $\mathbf{H}_{\boldsymbol{\theta}} = \nabla_{\boldsymbol{\theta}}^2 \mathcal{L}(\boldsymbol{\theta})$. Consider a layer-wise scaling transformation defined by $\tilde{\mathbf{W}}_l = c_l \mathbf{W}_l$, $c_l > 0$, with $\prod_{l=1}^{L} c_l = 1$. Let $\tilde{\boldsymbol{\theta}} = \tilde{\mathbf{W}}_l$ be the scaled parameters, and define $\mathbf{H}_{\tilde{\boldsymbol{\theta}}}$ as the corresponding Hessian. Then the spectrum-normalized Rényi entropy of $\mathbf{H}$ is invariant:*

$$H_\alpha(\mathbf{H}_{\tilde{\boldsymbol{\theta}}}) = H_\alpha(\mathbf{H}_{\boldsymbol{\theta}}), \quad \forall \alpha > 0, \ \alpha \neq 1. \tag{58}$$

**Discussion**   The reparameterization invariance is indeed a scale invariance, as the Rényi entropy of the Hessian matrix is not invariant under non-linear reparameterization. We regard reparameterization invariance as a necessary, but not sufficient, requirement for studying correlations with generalization. For a given minimum, there typically exists a large family of functionally equivalent parameterizations (obtained via reparameterization), and optimization may converge to any element of this family. To obtain a stable and comparable metric, it is therefore natural to seek quantities that are invariant within this equivalence class, which motivates the necessity of reparameterization invariance.

However, reparameterization invariance by itself does not guarantee a strong correlation with generalization. There are many possible invariant candidates, and they differ substantially in how sensitively they capture spectral structure. As a result, their empirical association with generalization can vary, even though they all satisfy the same invariance requirement.

# F    CONNECTION BETWEEN GLOBAL AND LOCAL RÉNYI SHARPNESS REGULARIZATION

**Proposition F.1** *Minimizing the global negative Rényi entropy with order $\alpha > 1$ is equivalent, in the block-diagonal case, to making each layer's spectrum uniform* and *balancing trace per dimension across layers. This configuration simultaneously minimizes the layerwise negative Rényi entropy for all orders $\alpha > 0$, including $\alpha < 1$. With small cross-layer couplings, the same conclusion holds up to a perturbation of order $\|\mathbf{E}\|_F / T$, where $T$ is the trace of the global Hessian matrix, and $\mathbf{E}$ is the difference between the Hessian matrix and the diagonal Hessian matrix. Considering that layer-wise trace can be adjusted without performance degradation, thus balancing trace per dimension across layers doesn't change the loss. Consequently, optimizing the global negative Rényi entropy is indeed optimizing the layer-wise negative Rényi entropy, i.e. layer-wise Rényi sharpness.*

*Proof.*

**Setup.**    Let $\mathbf{H} \in \mathbb{R}^{d \times d}$ be the (symmetric) Hessian at a candidate minimizer; we first treat $H \succeq 0$ and discuss standard relaxations in Remark F.7. Denote the eigenvalues by

$$\lambda_1(\mathbf{H}) \geq \cdots \geq \lambda_d(\mathbf{H}) \geq 0, \qquad T := \mathrm{Tr}(\mathbf{H}) > 0.$$

Define the *normalized spectrum* $p_i(\mathbf{H}) := \lambda_i(\mathbf{H})/T$ so that $\sum_{i=1}^d p_i(\mathbf{H}) = 1$. For $\alpha > 1$ define

$$\widetilde{\mathcal{R}}_\alpha(\mathbf{H}) := \sum_{i=1}^d \big(p_i(\mathbf{H})\big)^\alpha, \qquad -H_\alpha(\mathbf{H}) := \frac{1}{\alpha - 1} \log \widetilde{\mathcal{R}}_\alpha(\mathbf{H}). \tag{59}$$

Since $x \mapsto \log x$ is strictly increasing, minimizing $-H_\alpha(\mathbf{H})$ is equivalent to minimizing $\widetilde{\mathcal{R}}_\alpha(\mathbf{H})$ for any fixed $\alpha \neq 1$ (monotone transform).

Assume the network parameters are partitioned into $L$ layers with dimensions $d_1, \ldots, d_L$ (so $\sum_\ell d_\ell = d$). Let $\mathbf{H}_{\ell\ell} \in \mathbb{R}^{d_\ell \times d_\ell}$ be the principal block associated with layer $\ell$, with eigenvalues $\lambda_1(\mathbf{H}_{\ell\ell}) \geq \cdots \geq \lambda_{d_\ell}(\mathbf{H}_{\ell\ell}) \geq 0$ and trace $T_\ell := \mathrm{Tr}(\mathbf{H})_{\ell\ell} > 0$. Write

$$w_\ell := \frac{T_\ell}{T} \in (0, 1), \qquad \sum_{\ell=1}^L w_\ell = 1, \qquad \sigma_\alpha(\mathbf{H}_{\ell\ell}) := \sum_{i=1}^{d_\ell} \Big(\frac{\lambda_i(\mathbf{H}_{\ell\ell})}{T_\ell}\Big)^\alpha.$$

EXACT FACTORIZATION UNDER BLOCK-DIAGONALITY

**Lemma F.2 (Exact decomposition)** *If $\mathbf{H}$ is block diagonal with blocks $\mathbf{H}_{11}, \ldots, \mathbf{H}_{LL}$, then for any $\alpha > 0$,*

$$\widetilde{\mathcal{R}}_\alpha(\mathbf{H}) = \sum_{\ell=1}^L w_\ell^\alpha \, \sigma_\alpha(\mathbf{H}_{\ell\ell}). \tag{60}$$

*proof.* The spectrum of a block-diagonal matrix is the disjoint union of the spectra of its blocks. Since $p_i(\mathbf{H}) = \lambda_i(\mathbf{H})/T$ and $T = \sum_\ell T_\ell$, we compute

$$\sum_{i=1}^d \Big(\frac{\lambda_i(\mathbf{H})}{T}\Big)^\alpha = \sum_{\ell=1}^L \sum_{i=1}^{d_\ell} \Big(\frac{\lambda_i(\mathbf{H}_{\ell\ell})}{T}\Big)^\alpha = \sum_{\ell=1}^L \Big(\frac{T_\ell}{T}\Big)^\alpha \sum_{i=1}^{d_\ell} \Big(\frac{\lambda_i(\mathbf{H}_{\ell\ell})}{T_\ell}\Big)^\alpha.$$

**Lemma F.3 (Power-sum bounds within a layer)** *Fix $\ell$ and set $x_i := \lambda_i(\mathbf{H}_{\ell\ell})/T_\ell$ so that $x_i \geq 0$ and $\sum_{i=1}^{d_\ell} x_i = 1$. Then:*

1. *If $\alpha > 1$ (convex power), $\sigma_\alpha(\mathbf{H}_{\ell\ell}) = \sum_i x_i^\alpha \geq d_\ell^{1-\alpha}$, with equality iff $x_i \equiv 1/d_\ell$ (uniform spectrum inside the block).*

2. *If $0 < \beta < 1$ (concave power), $\sum_i x_i^\beta \leq d_\ell^{1-\beta}$, with equality iff $x_i \equiv 1/d_\ell$.*

*Both follow from Jensen's inequality (or Karamata's inequality) under the linear constraint $\sum_i x_i = 1$.*

**Theorem F.4 (Global optimum under block-diagonality for $\alpha > 1$)** *Assume* $\mathbf{H} =$ blk_diag$(\mathbf{H}_{11}, \ldots, \mathbf{H}_{LL})$ *and $\alpha > 1$. Then*

$$\widetilde{\mathcal{R}}_\alpha(\mathbf{H}) = \sum_{\ell=1}^{L} w_\ell^\alpha \, \sigma_\alpha(\mathbf{H}_{\ell\ell}) \;\geq\; \sum_{\ell=1}^{L} w_\ell^\alpha \, d_\ell^{1-\alpha} \;\geq\; d^{1-\alpha}, \tag{61}$$

*and the following are equivalent:*

1. *$\widetilde{\mathcal{R}}_\alpha(\mathbf{H})$ attains its global minimum $d^{1-\alpha}$.*

2. *(Layerwise uniformity) For each $\ell$, the normalized spectrum inside $\mathbf{H}_{\ell\ell}$ is uniform: $\lambda_i(\mathbf{H}_{\ell\ell})/T_\ell \equiv 1/d_\ell$.*

3. *(Trace-per-dimension balancing) The layer traces satisfy $w_\ell = \frac{d_\ell}{d}$, i.e. $\frac{T_\ell}{d_\ell}$ is constant across layers (equal average curvature per parameter).*

*proof.* The first inequality in equation 61 follows from Lemma F.3(1) applied to each $\sigma_\alpha(\mathbf{H}_{\ell\ell})$. Hence

$$\widetilde{\mathcal{R}}_\alpha(\mathbf{H}) \;\geq\; \sum_{\ell=1}^{L} a_\ell \, w_\ell^\alpha, \qquad a_\ell := d_\ell^{1-\alpha} > 0.$$

For fixed positive coefficients $a_\ell$ and $\alpha > 1$, the function $f(\boldsymbol{w}) := \sum_\ell a_\ell w_\ell^\alpha$ is strictly convex on the simplex $\{\boldsymbol{w} \geq 0, \; \sum_\ell w_\ell = 1\}$ and has a unique minimizer characterized by the KKT conditions:

$$\alpha a_\ell w_\ell^{\alpha-1} = \lambda \quad \Rightarrow \quad w_\ell \;\propto\; a_\ell^{-1/(\alpha-1)} = (d_\ell^{1-\alpha})^{-1/(\alpha-1)} = d_\ell.$$

Normalizing gives $w_\ell = d_\ell/d$. Substituting this and the layerwise lower bounds $\sigma_\alpha(\mathbf{H}_{\ell\ell}) \geq d_\ell^{1-\alpha}$ into equation 60 yields

$$\widetilde{\mathcal{R}}_\alpha(\mathbf{H}) \;\geq\; \sum_{\ell=1}^{L} \Big(\frac{d_\ell}{d}\Big)^\alpha d_\ell^{1-\alpha} = \frac{1}{d^\alpha} \sum_{\ell=1}^{L} d_\ell = d^{1-\alpha}.$$

Equality throughout holds iff (i) each $\sigma_\alpha(\mathbf{H}_{\ell\ell})$ attains its lower bound, i.e. the layer spectra are uniform, and (ii) $w_\ell = d_\ell/d$. This proves both necessity and sufficiency and the equivalences claimed. ∎

**Corollary F.5 (Simultaneous layerwise optimality for all orders $\beta > 0$, $\beta \neq 1$)** *Under the conditions of Theorem F.4, if the global minimum is attained (equivalently: each block has uniform normalized spectrum and $w_\ell = d_\ell/d$), then for every order $\beta > 0$,*

$$\text{the quantity} \quad -H_\beta(\mathbf{H}_{\ell\ell}) = \frac{1}{\beta-1} \log \sum_{i=1}^{d_\ell} \Big(\frac{\lambda_i(\mathbf{H}_{\ell\ell})}{T_\ell}\Big)^\beta \quad \text{is minimized (for all $\ell$).}$$

*In particular, the same configuration minimizes the layerwise negative Rényi entropy for $\beta > 1$ and for $0 < \beta < 1$.*

*proof.* For $\beta > 1$, Lemma F.3(1) shows that the uniform layer spectrum uniquely minimizes $\sum_i x_i^\beta$ subject to $\sum_i x_i = 1$; since the logarithm and the factor $(\beta-1)^{-1} > 0$ are monotone, it also minimizes $-H_\beta$. For $0 < \beta < 1$, Lemma F.3(2) shows that the uniform layer spectrum uniquely *maximizes* $\sum_i x_i^\beta$; because $(\beta-1)^{-1} < 0$, this again minimizes $-H_\beta$. The claim holds for each layer $\ell$. ∎

STABILITY UNDER CROSS-LAYER COUPLINGS

Real Hessians may not be exactly block diagonal. Write

$$\mathbf{B} := \text{blk\_diag}(\mathbf{H}_{11}, \ldots, \mathbf{H}_{LL}), \qquad \mathbf{E} := \mathbf{H} - \mathbf{B}.$$

Note that $\text{Tr}(\mathbf{E}) = 0$ (off-diagonal blocks contribute zero trace), hence $\text{Tr}(\mathbf{H}) = \text{Tr}(\mathbf{B}) = T$.

**Proposition F.6 (Perturbation bound for $\alpha > 1$)** *Let $\alpha > 1$ and set $\Lambda_* := \max\{\lambda_{\max}(\mathbf{H}), \lambda_{\max}(\mathbf{B})\}$. Then*

$$\left|\widetilde{\mathcal{R}}_\alpha(\mathbf{H}) - \widetilde{\mathcal{R}}_\alpha(\mathbf{B})\right| \leq \alpha\left(\frac{\Lambda_*}{T}\right)^{\alpha-1} \frac{\sqrt{d}\,\|\mathbf{E}\|_F}{T}. \tag{62}$$

*Consequently, if $\|\mathbf{E}\|_F/T$ is small, minimizing $\widetilde{\mathcal{R}}_\alpha(\mathbf{H})$ is optimization-equivalent up to $O(\|\mathbf{E}\|_F/T)$ to minimizing $\widetilde{\mathcal{R}}_\alpha(\mathbf{B})$, which by Theorem F.4 drives each layer toward its uniform spectrum (and hence decreases all layerwise $-H_\beta$, $\beta > 0$, simultaneously).*

$proof$. Let $\{\lambda_i\}$ and $\{\mu_i\}$ be the eigenvalues of $\mathbf{H}$ and $\mathbf{B}$ sorted in nonincreasing order. By the Hoffman–Wielandt inequality, $\sum_{i=1}^d (\lambda_i - \mu_i)^2 \leq \|\mathbf{E}\|_F^2$. For $\alpha > 1$, the function $\phi(x) = x^\alpha$ has derivative bounded on $[0, \Lambda_*]$ by $\alpha\Lambda_*^{\alpha-1}$. Hence by the mean value theorem and Cauchy–Schwarz,

$$\left|\sum_i \lambda_i^\alpha - \sum_i \mu_i^\alpha\right| \leq \alpha\Lambda_*^{\alpha-1} \sum_i |\lambda_i - \mu_i| \leq \alpha\Lambda_*^{\alpha-1}\sqrt{d}\,\|\mathbf{E}\|_F.$$

Since $\mathrm{Tr}(\mathbf{H}) = \mathrm{Tr}(\mathbf{B}) = T$, dividing both sides by $T^\alpha$ yields equation 62.

REMARK (ORDER-ROBUSTNESS FOR $0 < \alpha < 1$).

Recall the decomposition $\widetilde{\mathcal{R}}_\alpha(\mathbf{H}) = \sum_{\ell=1}^L w_\ell^\alpha\, \sigma_\alpha(\mathbf{H}_{\ell\ell})$. Passing from $\alpha > 1$ to $0 < \alpha < 1$ only changes the *curvature* of $\widetilde{\mathcal{R}}_\alpha(\mathbf{H})$ and $\sigma_\alpha(\mathbf{H}_{\ell\ell})$ (from convex to concave) and flips the outer optimization direction (since $\frac{1}{1-\alpha}$ changes sign), but it does *not* change the location of the optimizer.

Consequently, in the block-diagonal setting, minimizing the *global* negative Rényi entropy $-H_\alpha(\mathbf{H})$ for any order $\alpha > 0$, $\alpha \neq 1$ is equivalent to making each layer's spectrum uniform and balancing trace per dimension across layers; this configuration simultaneously minimizes the *layerwise* negative Rényi entropy for all $\beta > 0$ (including $\beta < 1$). With small cross-layer couplings $\mathbf{H} = \mathrm{blk\_diag}(\mathbf{H}_{11}, \ldots, \mathbf{H}_{LL}) + \mathbf{E}$, the same conclusion holds up to a perturbation of order $O(\|\mathbf{E}\|_F/\mathrm{Tr}(\mathbf{H}))$ by continuity of $H_\alpha$ in total variation.

**Remark F.7 (PSD reduction and alternatives)** *If $H$ is indefinite, one may work with $|H|$ (absolute value via spectral decomposition), with a Gauss–Newton/Fisher approximation, or with a shifted PSD proxy (e.g. $\mathbf{H} + \gamma\mathbf{I}$ with $\gamma > 0$), apply the above results verbatim to the PSD object, and then track the dependence on the chosen proxy. The normalized formulation equation 59 is unchanged as long as the trace $T > 0$.*

## G   Arbitrary Trace Rescaling under Fixed Normalized Spectrum

In this appendix, we study how the Hessian trace behaves under linear reparameterizations, and in particular under those that preserve the *spectral shape* (normalized eigenvalue distribution) of the Hessian. We show that, for each individual model, such reparameterizations give a continuous family of possible scalings of the Hessian trace. For a finite collection of models, this leads to an explicit infinite feasibility condition under which all Hessian traces can be aligned to a common value while preserving spectral shape.

Let $w \in \mathbb{R}^n$ denote the parameter vector of a given (layer of a) model, and let $H \in \mathbb{R}^{n \times n}$ be the corresponding Hessian at a local minimum. Throughout this subsection, we assume that $H$ is symmetric positive definite.

We consider linear reparameterizations of the form

$$w = A\theta, \qquad A \in \mathbb{R}^{n \times n} \text{ invertible}, \tag{63}$$

and define the reparameterized loss by $L(\theta) := L(A\theta)$. By the chain rule, the Hessian in $\theta$–coordinates is

$$H_\theta := \nabla_\theta^2 L(\theta) = A^\top H A. \tag{64}$$

The corresponding parameter vector is

$$\theta = A^{-1} w. \tag{65}$$

We are particularly interested in reparameterizations that preserve the *spectral shape* of the Hessian, i.e. that only rescale all eigenvalues by a common positive factor.

**Lemma G.1 (Spectral-shape–preserving reparameterizations)** *Let $H \succ 0$. For any scalar $d > 0$ and any orthogonal matrix $Q \in \mathbb{R}^{n \times n}$ ($Q^\top Q = I$), define*

$$A(d, Q) := H^{-1/2} (dQ) H^{1/2}. \tag{66}$$

*Then the corresponding reparameterized Hessian $H_\theta = A(d, Q)^\top H A(d, Q)$ satisfies*

$$H_\theta = d^2 H. \tag{67}$$

*In particular, the eigenvalues of $H_\theta$ are $\{d^2 \lambda_i(H)\}_i$, so the normalized spectrum $\{\lambda_i(H_\theta)/\text{Tr}(H_\theta)\}_i$ coincides with that of $H$.*

$Proof$. A direct computation yields

$$\begin{aligned}
H_\theta &= A(d, Q)^\top H A(d, Q) \\
&= \left( H^{1/2} Q^\top d H^{-1/2} \right) H \left( H^{-1/2} dQ H^{1/2} \right) \\
&= d^2 \, H^{1/2} Q^\top H^{-1/2} H H^{-1/2} Q H^{1/2} \\
&= d^2 \, H^{1/2} Q^\top Q H^{1/2} = d^2 H.
\end{aligned}$$

Thus all eigenvalues are scaled by $d^2$, and the normalized eigenvalue distribution is unchanged.

We next study the effect of equation 66 on the parameter norm. Let

$$u := H^{1/2} w, \qquad r := \|u\|_2 > 0. \tag{68}$$

For $A = A(d, Q)$ as in Lemma G.1, we have

$$\begin{aligned}
\theta = A^{-1} w &= \left( H^{-1/2} (dQ) H^{1/2} \right)^{-1} w = \frac{1}{d} H^{-1/2} Q^\top H^{1/2} w \\
&= \frac{1}{d} H^{-1/2} Q^\top u.
\end{aligned} \tag{69}$$

Let $\text{Tr}(H)$ denote the original trace. Under the reparameterization with factor $d^2$ we have

$$\text{Tr}(H_\theta) = d^2 \text{Tr}(H). \tag{70}$$

As $d \in \mathbb{R}$, we can adjust the trace of the Hessian matrix to an arbitrarily prescribed value while keeping the normalized eigenvalue spectrum completely unchanged.

Since our sharpness measure is defined in terms of the normalized spectrum (e.g. via the Rényi entropy of $\{\lambda_i(H)/\mathrm{Tr}(H)\}_i$), the global scale of the trace is factored out by design. Combining this observation with the reparameterization freedom described above, we conclude that scale-dependent quantities such as the raw trace $\mathrm{Tr}(H)$ do not carry reparameterization-robust geometric information. What remains intrinsic is precisely the *shape* of the Hessian spectrum, which we quantify via its Rényi entropy.

## H  ON THE DISCUSSION OF SAM VARIANTS

In this section, we discuss several sharpness-aware minimization variants and compare them with Rényi sharpness-aware minimization (RSAM). We focus on closely related methods, including the original SAM (Foret et al., 2020), Sparse SAM (Mi et al., 2022), Eigen SAM (Luo et al., 2024), Tilted SAM (Li et al., 2024b), Frobenius SAM (Tahmasebi et al., 2024), and Fisher SAM (Kim et al., 2022).

Vanilla SAM has been shown to implicitly minimize the largest eigenvalue of the training loss Hessian (Wen et al., 2023), and Sparse SAM, which accelerates SAM by explicitly masking part of the updates, essentially targets the same quantity. Eigen SAM directly penalizes the largest eigenvalue in its minimization step. Tilted SAM samples noise in multiple directions to perturb the weights and penalizes the sum of the exponentiated perturbed losses over these noise samples. Intuitively, the exponential transform amplifies the sharpest directions of the loss landscape, so it imposes a stronger penalty along these directions. From this perspective, Tilted SAM can be viewed as effectively penalizing the largest (or relatively large) eigenvalues of the Hessian. Frobenius SAM penalizes the Frobenius norm of the Hessian matrix; if we normalize this norm by the squared trace, the resulting quantity becomes essentially a monotone function of the order-2 Rényi entropy. Fisher SAM minimizes the same type of robust objective as SAM, but with the neighborhood defined by a Riemannian metric induced by the Fisher information; this is equivalent to penalizing the largest eigenvalue of the Hessian with respect to the Fisher metric.

Overall, these methods regularize some spectral function of the Hessian eigenvalues. Whether one penalizes the largest eigenvalue or minimizes the Frobenius norm of the Hessian, the implicit goal is to encourage the eigenvalues to move closer to each other; for example, reducing the largest eigenvalue typically decreases the overall spread of the spectrum.

In contrast, Rényi sharpness explicitly focuses on the dispersion of the normalized eigenvalues. Modern deep models usually enjoy certain reparameterization invariances, so we can rescale the overall magnitude of the Hessian without changing the model's behavior. Consequently, if the regularizer depends only on the unnormalized eigenvalues (such as the spectral norm or a generic spectral function), then shrinking the global scale of the Hessian will always reduce the regularization term, even when the model performance and the relative shape of the spectrum remain unchanged. Therefore, minimizing such penalties alone does not guarantee that the eigenvalues become more uniformly distributed.

# I   SHARPNESS MEASURES

In this section, we give a detailed introduction to the sharpness measure we use. The content of this section refers to Jiang et al. (2019) and the original works corresponding to these measures.

## I.1   NORM BASED MEASURES

Several generalization bounds have been proved for neural networks using margin and norm notions. In this section, we go over several such measures. For fully connected networks, Bartlett & Mendelson (2002) have shown a bound based on product of $\ell_{1,\infty}$ norm of the layer weights times a $2^d$ factor where $\ell_{1,\infty}$ is the maximum over hidden units of the $\ell_2$ norm of the incoming weights to the hidden unit. Neyshabur et al. (2015) proved a bound based on product of Frobenius norms of the layer weights times a $2^d$ factor and Golowich et al. (2017) was able to improve the factor to $\sqrt{d}$. Bartlett et al. (2017) proved a bound based on product of spectral norm of the layer weights times sum over layers of ratio of Frobenius norm to spectral norm of the layer weights and Neyshabur et al. (2018a) showed a similar bound can be achieved in a simpler way using PAC-bayesian framework.

**Spectral Norm**   Unfortunately, none of the above founds are directly applicable to convolutional networks. Pitas et al. (2017) built on Neyshabur et al. (2018a) and extended the bound on the spectral norm to convolutional networks. The bound is very similar to the one for fully connected networks by Bartlett et al. (2017). We next restate their generalization bound for convolutional networks including the constants.

**Theorem I.1 (Pitas et al. (2017))** *Let $B$ an upper bound on the $\ell_2$ norm of any point in the input domain. For any $B, \gamma, \delta > 0$, the following bound holds with probability $1 - \delta$ over the training set:*

$$L \leq \hat{L}_\gamma + \sqrt{\frac{\left(84B \sum_{i=1}^d k_i \sqrt{c_i} + \sqrt{\ln(4n^2 d)}\right)^2 \prod_{i=1}^d \|\mathbf{W}_i\|_2^2 \sum_{j=1}^d \frac{\|\mathbf{W}_j - \mathbf{W}_j^0\|_F^2}{\|\mathbf{W}_j\|_2^2} + \ln(\frac{m}{\delta})}{\gamma^2 m}} \quad (71)$$

**Parameter Norm**   Given recent evidence on the importance of distance to initialization (Dziugaite & Roy, 2017; Nagarajan & Kolter, 2019; Neyshabur et al., 2018b), we calculate the following measures:

$$\mu_{\text{frobenius-distance}}(f_\mathbf{w}) = \sum_{i=1}^d \|\mathbf{W}_i - \mathbf{W}_i^0\|_F^2 \quad (72)$$

In the case when the reference matrix $\mathbf{W}_i^0 = \mathbf{0}$ for all weights, Eq (72) the Frobenius norm of the parameters, which also corresponds to the distance from the origin:

$$\mu_{\text{param-norm}}(f_\mathbf{w}) = \sum_{i=1}^d \|\mathbf{W}_i\|_F^2 \quad (73)$$

**Fisher-Rao Norm**   Fisher-Rao metric was introduced in Liang et al. (2017) as a complexity measure for neural networks. Liang et al. (2017) showed that Fisher-Rao norm is a lower bound on the path-norm and it correlates in some cases. We define a measure based on the Fisher-Rao metric of the network:

$$\mu_{\text{Fisher-Rao}}(f_\mathbf{w}) = \frac{(d+1)^2}{m} \sum_{i=1}^m \langle \mathbf{w} \nabla_\mathbf{w} \ell(f_\mathbf{w}(\mathbf{X}_i)), y_i \rangle^2 \quad (74)$$

where $\ell$ is the cross-entropy loss.

**Trace**   Trace measure is defined as the trace of the Hessian matrix of the loss function on the training dataset with respect to the weights, i.e., $\text{Tr}(\mathbf{H})$, where $\mathbf{H} = \nabla_\mathbf{w}^2 L(\mathcal{S}, \mathbf{w})$.

## I.2 FLATNESS-BASED MEASURES

PAC-Bayesian framework (McAllester, 1999) allows us to study flatness of a solution and connect it to generalization. Given a prior $P$ is is chosen before observing the training set and a posterior $Q$ which is a distribution on the solutions of the learning algorithm (and hence depends on the training set), we can bound the expected generalization error of solutions generated from $Q$ with high probability based on the $D_{\mathrm{KL}}$ divergence of $P$ and $Q$. The next theorem states a simplified version of PAC-Bayesian bounds.

**Theorem I.2** *For any $\delta > 0$, distribution $D$, prior $P$, with probability $1 - \delta$ over the training set, for any posterior $Q$ the following bound holds:*

$$\mathbb{E}_{\mathbf{v}\sim Q}\left[L(f_{\mathbf{v}})\right] \leq \mathbb{E}_{\mathbf{w}\sim Q}\left[\hat{L}(f_{\mathbf{v}})\right] + \sqrt{\frac{D_{\mathrm{KL}}(Q||P) + \log\left(\frac{m}{\delta}\right)}{2(m-1)}} \tag{75}$$

If $P$ and $Q$ are Gaussian distributions with $P = \mathcal{N}(\mu_P, \Sigma_P)$ amd $Q = \mathcal{N}(\mu_Q, \Sigma_Q)$, then the $D_{\mathrm{KL}}$-term can be written as follows:

$$D_{\mathrm{KL}}(\mathcal{N}(\mu_Q, \Sigma_Q)||\mathcal{N}(\mu_P, \Sigma_P)) = \frac{1}{2}\left[\mathrm{tr}\left(\Sigma_P^{-1}\Sigma_Q\right) + (\mu_Q - \mu_P)^\top \Sigma_P^{-1}(\mu_Q - \mu_P) - k + \ln(\frac{\det\Sigma_P}{\det\Sigma_Q})\right].$$

Setting $Q = \mathcal{N}(\mathbf{w}, \sigma^2 I)$ and $P = \mathcal{N}(\mathbf{w}^0, \sigma^2 I)$ similar to Neyshabur et al. (2017), the $D_{\mathrm{KL}}$ term will be simply $\frac{\|\mathbf{w}-\mathbf{w}^0\|_2^2}{2\sigma^2}$. However, since $\sigma$ belongs to prior, if we search to find a value for $\sigma$, we need to adjust the bound to reflect that. Since we search over less than 20000 predefined values of $\sigma$ in our experiments, we can use the union bound which changes the logarithmic term to $\log(20000m/\delta)$ and we get the following bound:

$$\mathbb{E}_{\mathbf{u}\sim\mathcal{N}(u,\sigma^2 I)}\left[L(f_{\mathbf{w}+\mathbf{u}})\right] \leq \mathbb{E}_{\mathbf{u}\sim\mathcal{N}(u,\sigma^2 I)}\left[\hat{L}(f_{\mathbf{w}+\mathbf{u}})\right] + \sqrt{\frac{\frac{\|\mathbf{w}-\mathbf{w}^0\|_2^2}{4\sigma^2} + \log(\frac{m}{\sigma}) + 10}{m-1}} \tag{76}$$

Based on the above bound, we define the following measures using the origin as reference tensors:

$$\mu_{\text{pac-bayes-orig}}(f_{\mathbf{w}}) = \frac{\|\mathbf{w}\|_2^2}{4\sigma^2} + \log(\frac{m}{\delta}) + 10 \tag{77}$$

where $\sigma$ is chosen to be the largest number such that $\mathbb{E}_{\mathbf{u}\sim\mathcal{N}(u,\sigma^2 I)}\left[\hat{L}(f_{\mathbf{w}+\mathbf{u}})\right] \leq 0.1$.

To understand the importance of the flatness parameters $\sigma$, we also define the following measure:

$$\mu_{\text{pac-bayes-flatness}}(f_{\mathbf{w}}) = \frac{1}{\sigma^2} \tag{78}$$

where $\sigma$ is computed as explained above.

## I.3 SHARPNESS-BASED MEASURES

**SAM** Foret et al. (2020) proposed a generalization bound under weight perturbation:

**Theorem I.3** *For any $\rho > 0$ and any distribution $D$, with probability $1 - \delta$ over the choice of the training set $\mathcal{S} \sim D$,*

$$L_D(\boldsymbol{w}) \leq \max_{\|\boldsymbol{\epsilon}\|_2 \leq \rho} L_{\mathcal{S}}(\boldsymbol{w} + \boldsymbol{\epsilon}) + \sqrt{\frac{k\log\left(1 + \frac{\|\boldsymbol{w}\|_2^2}{\rho^2}\left(1 + \sqrt{\frac{\log(n)}{k}}\right)^2\right) + 4\log\frac{n}{\delta} + \tilde{O}(1)}{n-1}} \tag{79}$$

*where $n = |\mathcal{S}|$, $k$ is the number of parameters and we assumed $L_D(\boldsymbol{w}) \leq \mathbb{E}_{\epsilon_i\sim\mathcal{N}(0,\rho)}[L_D(\boldsymbol{w} + \boldsymbol{\epsilon})]$.*

Thus, the sharpness of SAM is defined as

$$\max_{\|\boldsymbol{\epsilon}\|_p \leq \rho} L_S(\boldsymbol{w} + \boldsymbol{\epsilon}) - L_S(\boldsymbol{w}) \tag{80}$$

if we minimize $\max_{||\epsilon||_p \leq \rho} L_S(w + \epsilon)$, the solution via a first-order approximation will be

$$\epsilon(\mathbf{w}) = \rho \frac{\text{sign}(\mathbf{g}) \odot |\mathbf{g}|^{q-1}}{\|\mathbf{g}\|_q^{q-1}}, \qquad \mathbf{g} = \nabla L_S(\mathbf{w}), \quad \frac{1}{p} + \frac{1}{q} = 1 \tag{81}$$

Especially, if $p = 2$

$$\epsilon(w) = \rho \frac{g}{\|g\|_2}, \qquad g = \nabla L_S(w). \tag{82}$$

and if $p = \infty$

$$\epsilon(w) = \rho \, \text{sign}(g), \qquad g = \nabla L_S(w). \tag{83}$$

**ASAM**  Kwon et al. (2021) proposed a new adaptive sharpness which is reparameterization invariant with a normalization operator:

**Definition I.4 (Normalization operator)**  *Let $\{T_{\mathbf{w}}, \mathbf{w} \in \mathbb{R}^k\}$ be a family of invertible linear operators on $\mathbb{R}^k$. Given a weight $\mathbf{w}$, if $T_{A\mathbf{w}}^{-1} A = T_{\mathbf{w}}^{-1}$ for any invertible scaling operator $A$ on $\mathbb{R}^k$ which does not change the loss function, we say $T_{\mathbf{w}}^{-1}$ is a normalization operator of $\mathbf{w}$.*

Using the normalization operator, we define adaptive sharpness as follows.

**Definition I.5 (Adaptive sharpness)**  *If $T_{\mathbf{w}}^{-1}$ is the normalization operator of $\mathbf{w}$ in Definition I.4, adaptive sharpness of $\mathbf{w}$ is defined by*

$$\max_{\|T_{\mathbf{w}}^{-1}\epsilon\|_p \leq \rho} L_S(\mathbf{w} + \epsilon) - L_S(\mathbf{w}) \tag{84}$$

*where $1 \leq p \leq \infty$.*

They also demonstrated a generalization bound for adaptive sharpness:

**Theorem I.6**  *Let $T_{\mathbf{w}}^{-1}$ be the normalization operator on $\mathbb{R}^k$. If $L_D(\mathbf{w}) \leq E_{\epsilon_i \sim \mathcal{N}(0,\sigma^2)}[L_D(\mathbf{w}+\epsilon)]$ for some $\sigma > 0$, then with probability $1 - \delta$,*

$$L_D(\mathbf{w}) \leq \max_{\|T_{\mathbf{w}}^{-1}\epsilon\|_2 \leq \rho} L_S(\mathbf{w} + \epsilon) + h\left(\frac{\|\mathbf{w}\|_2^2}{\eta^2 \rho^2}\right) \tag{85}$$

*where $h : \mathbb{R}^+ \to \mathbb{R}^+$ is a strictly increasing function, $n = |S|$ and $\rho = \sqrt{k}\sigma(1 + \sqrt{\log n/k})/\eta$.*

For a minimax problem

$$\min_{\mathbf{w}} \max_{\|T_{\mathbf{w}}^{-1}\epsilon\|_p \leq \rho} L_S(\mathbf{w} + \epsilon) + \frac{\lambda}{2}\|\mathbf{w}\|_2^2. \tag{86}$$

The solution under a first-order approximation for adaptive sharpness is

$$\epsilon = \rho T_{\mathbf{w}} \, \text{sign}(\nabla L_S(\mathbf{w})) \frac{|T_{\mathbf{w}} \nabla L_S(\mathbf{w})|^{q-1}}{\|T_{\mathbf{w}} \nabla L_S(\mathbf{w})\|_q^{q-1}} \tag{87}$$

Especially, if $p = 2$,

$$\epsilon = \rho \frac{T_{\mathbf{w}}^2 \nabla L_S(\mathbf{w})}{\|T_{\mathbf{w}} \nabla L_S(\mathbf{w})\|_2} \tag{88}$$

and if $p = \infty$,

$$\epsilon = \rho T_{\mathbf{w}} \, \text{sign}(\nabla L_S(\mathbf{w})). \tag{89}$$

## I.4   Implementation

The measures, including Fisher-Rao norm (Eq. 74), Parameter norm (Eq. 73), Trace of the Hessian matrix, Pac-Bayes from origin (Eq.77), and the Pac-Bayes flatness (Eq. 78) are computed through the repository by Dziugaite et al. (2020).

The measures, including $L_2$ sharpness (Eq. 82), $L_\infty$ sharpness (Eq. 83), $L_2$ adaptive sharpness (Eq. 88), and $L_\infty$ sharpness (Eq. 83) compute the corresponding sharpness directly from the solution of the minimax problem.

For the SAM(ASAM) sharpness, we conducted a grid search over $\rho \in \{10^{-6}, 3 \times 10^{-6}, 10^{-5}, 3 \times 10^{-5}, 10^{-4}, 3 \times 10^{-4}, 10^{-3}, 3 \times 10^{-3}, 10^{-2}, 3 \times 10^{-2}, 10^{-1}, 0.3, 1\}$ and select the $\rho$ with highest correlation coefficient for each task.

## J EXPERIMENTAL DETAILS

In this section, we describe the datasets, models, hyperparameter choices, and eigenspectrum adjustment used in our experiments. All of our experiments are run using PyTorch on Nvidia GTX1080ti, RTX3090s, RTX4090s, and RTX5090s.

### J.1 DATASET

**CIFAR-10.** CIFAR-10 consists of 60,000 color images, with each image belonging to one of ten different classes with size $32 \times 32$. The classes include common objects such as airplanes, automobiles, birds, cats, deer, dogs, frogs, horses, ships, and trucks. The CIFAR-10 dataset is divided into two subsets: a training set and a test set. The training set contains 50,000 images, while the test set contains 10,000 images (Krizhevsky & Hinton, 2009). For data processing, we follow the standard augmentation: normalize channel-wise, randomly horizontally flip, and random cropping.

**CIFAR-100.** The CIFAR-100 dataset consists of 60,000 color images, with each image belonging to one of 100 different fine-grained classes (Krizhevsky & Hinton, 2009). These classes are organized into 20 superclasses, each containing 5 fine-grained classes. Similar to CIFAR-10, the CIFAR-100 dataset is split into a training set and a test set. The training set contains 50,000 images, and the test set contains 10,000 images. Each image is of size 32x32 pixels and is labeled with its corresponding fine-grained class. Augmentation includes normalize channel-wise, randomly horizontally flip, and random cropping.

**TinyImageNet.** TinyImageNet comprises 100,000 images distributed across 200 classes, with each class consisting of 500 images (Le & Yang, 2015). These images have been resized to 64 × 64 pixels and are in full color. Each class encompasses 500 training images, 50 validation images, and 50 test images. Data augmentation techniques encompass normalization, random rotation, and random flipping. The dataset includes distinct train, validation, and test sets for experimentation.

### J.2 MODEL

In all experiments, the neural networks are initialized by the default initialization provided by Pytorch.

**ResNet18, ResNet20, ResNet34 and ResNet50 (He et al., 2016).** We use the standard ResNet architecture for TinyImageNet and tune it for the CIFAR dataset on the correlation validation tasks. The detailed network architecture parameters are shown in Table 3 and Table 4. ResNet18, ResNet20, ResNet34, and ResNet56 are trained on CIFAR-100 . The standard ResNet18 is trained on TinyImageNet for efficient computing and tuned ResNet18 is trained on TinyImageNet for sharpness-aware minimization.

**WideResNet (Zagoruyko & Komodakis, 2016).** The Wide ResNet implementation uses the `wrn28_10` model from the *horuma* (Hataya, 2018) library. Architecture details can be found in Table 4.

**Vision Transformer.** We use the SimpleViT architecture from the `vit-pytorch` library, which is a modification of the standard ViT (Dosovitskiy et al., 2020) with a fixed positional embedding and global average pooling instead of the CLS embedding.

### J.3 TRAINING HYPER-PARAMETERS SETUP

#### J.3.1 CORRELATION EXPERIMENTS

We train models for 200 epochs, and cosine learning rate decay is adopted after a linear warm-up for the first 10 epochs. For the task on CIFAR10/CIFAR100, we vary the initial learning rate {0.001, 0.03, 0.1}, batch size {128, 384, 1280}, and weight decay {0.00001, 0.00005, 0.0001, 0.0003, 0.0005} for SGD with momentum and the initial learning rate {0.00001, 0.0003, 0.001}, batch size {128, 384, 1280}, and weight decay {0.00001, 0.00005, 0.0001, 0.0003, 0.0005} for Adam. For

Table 3: ResNet architecture used in correlation experiments.

| Layer | ResNet18$_{\text{CIFAR}}$ | ResNet34 | ResNet18$_{\text{TinyImageNet}}$ |
|---|---|---|---|
| Conv 1 | 3×3, 64
padding 1
stride 1 | 3×3, 64
padding 1
stride 1 | 7×7, 64
padding 3
stride 2
Max Pool, ks 3, str 2, pad 1 |
| Layer stack 1 | $\begin{bmatrix} 3{\times}3,\ 64 \\ 3{\times}3,\ 64 \end{bmatrix}{\times}2$ | $\begin{bmatrix} 3{\times}3,\ 64 \\ 3{\times}3,\ 64 \end{bmatrix}{\times}3$ | $\begin{bmatrix} 3{\times}3,\ 64 \\ 3{\times}3,\ 64 \end{bmatrix}{\times}2$ |
| Layer stack 2 | $\begin{bmatrix} 3{\times}3,\ 128 \\ 3{\times}3,\ 128 \end{bmatrix}{\times}2$ | $\begin{bmatrix} 3{\times}3,\ 128 \\ 3{\times}3,\ 128 \end{bmatrix}{\times}4$ | $\begin{bmatrix} 3{\times}3,\ 128 \\ 3{\times}3,\ 128 \end{bmatrix}{\times}2$ |
| Layer stack 3 | $\begin{bmatrix} 3{\times}3,\ 256 \\ 3{\times}3,\ 256 \end{bmatrix}{\times}2$ | $\begin{bmatrix} 3{\times}3,\ 256 \\ 3{\times}3,\ 256 \end{bmatrix}{\times}6$ | $\begin{bmatrix} 3{\times}3,\ 256 \\ 3{\times}3,\ 256 \end{bmatrix}{\times}2$ |
| Layer stack 4 | $\begin{bmatrix} 3{\times}3,\ 512 \\ 3{\times}3,\ 512 \end{bmatrix}{\times}2$ | $\begin{bmatrix} 3{\times}3,\ 512 \\ 3{\times}3,\ 512 \end{bmatrix}{\times}3$ | $\begin{bmatrix} 3{\times}3,\ 512 \\ 3{\times}3,\ 512 \end{bmatrix}{\times}2$ |
| FC | Adaptive Avg Pool, output size $(1,1)$
$512 \times$ N_CLASSES | $512 \times$ N_CLASSES | $512 \times$ N_CLASSES |

Table 4: ResNet architecture used in sharpness-aware minimization experiments.

| Layer | ResNet-20 | ResNet-56 | ResNet-50 | WideResNet-28-10 |
|---|---|---|---|---|
| Conv 1 | 3×3, 16
padding 1
stride 1 | 3×3, 16
padding 1
stride 1 | 3×3, 64
padding 1
stride 1 | 3×3, 16
padding 1
stride 1 |
| Layer stack 1 | $\begin{bmatrix} 3{\times}3,\ 16 \\ 3{\times}3,\ 16 \end{bmatrix}{\times}3$ | $\begin{bmatrix} 3{\times}3,\ 16 \\ 3{\times}3,\ 16 \end{bmatrix}{\times}9$ | $\begin{bmatrix} 1{\times}1,\ 64 \\ 3{\times}3,\ 64 \\ 1{\times}1,\ 256 \end{bmatrix}{\times}3$ | $\begin{bmatrix} 3{\times}3,\ 160 \\ 3{\times}3,\ 160 \end{bmatrix}{\times}4$ |
| Layer stack 2 | $\begin{bmatrix} 3{\times}3,\ 32 \\ 3{\times}3,\ 32 \end{bmatrix}{\times}3$ | $\begin{bmatrix} 3{\times}3,\ 32 \\ 3{\times}3,\ 32 \end{bmatrix}{\times}9$ | $\begin{bmatrix} 1{\times}1,\ 128 \\ 3{\times}3,\ 128 \\ 1{\times}1,\ 512 \end{bmatrix}{\times}4$ | $\begin{bmatrix} 3{\times}3,\ 320 \\ 3{\times}3,\ 320 \end{bmatrix}{\times}4$ |
| Layer stack 3 | $\begin{bmatrix} 3{\times}3,\ 64 \\ 3{\times}3,\ 64 \end{bmatrix}{\times}3$ | $\begin{bmatrix} 3{\times}3,\ 64 \\ 3{\times}3,\ 64 \end{bmatrix}{\times}9$ | $\begin{bmatrix} 1{\times}1,\ 256 \\ 3{\times}3,\ 256 \\ 1{\times}1,\ 1024 \end{bmatrix}{\times}6$ | $\begin{bmatrix} 3{\times}3,\ 640 \\ 3{\times}3,\ 640 \end{bmatrix}{\times}4$ |
| Layer stack 4 | - | | $\begin{bmatrix} 1{\times}1,\ 512 \\ 3{\times}3,\ 512 \\ 1{\times}1,\ 2048 \end{bmatrix}{\times}3$ | - |
| FC | Avg Pool, kernel size 8
$64 \times$ N_CLASSES | Avg Pool, kernel size 8
$64 \times$ N_CLASSES | Adaptive Avg Pool, output size $(1,1)$
$2048 \times$ N_CLASSES | Avg Pool, kernel size 8
$640 \times$ N_CLASSES |

the task on TinyImageNet, we vary the initial learning rate {0.001, 0.03, 0.1}, batch size {128, 384, 1280}, and weight decay {0.000003, 0.00001, 0.00003, 0.00005, 0.0001, 0.0003} for SGD with momentum and the initial learning rate {0.00001, 0.0003, 0.001}, batch size {128, 384, 1280}, and weight decay {0.000003, 0.00001, 0.00003, 0.00005, 0.0001, 0.0003} for Adam.

Different from Jiang et al. (2019), we pick the data augmentation in the training scheme, which is a common setting in modern deep learning, but we still compute the sharpness measure without data augmentation, as from a theoretical perspective, data augmentation is also challenging to analyze since the training samples generated from the procedure are no longer identical and independently distributed.

To investigate the relationship between sharpness and generalization under common training strategies, we pick the stopping criterion based on the number of iterations or the number of epochs. To avoid differences in optimization speed across hyperparameter settings, we follow the linear scaling rule recommended by Goyal et al. (2017) and scale the learning rate and batch size in tandem, which yields comparable convergence after the same number of epochs.

### J.3.2 SHARPNESS-AWARE MINIMIZATION EXPERIMENTS

Firstly, we will introduce the Rényi Sharpness-Aware Minimization algorithm as follows:

---

**Algorithm 2** Rényi Sharpness-Aware Minimization (RSAM) Algorithm

---

**Input:** Loss function $\ell$, training dataset $S := \bigcup_{i=1}^{n}\{(\mathbf{x}_i, \mathbf{y}_i)\}$, mini-batch size $b$, radius $\rho$, Rényi order $\alpha$, plain SGD epoch $e_1$, RSAM epoch $e_2$, weight decay coefficient $\lambda$, scheduled learning rate $\beta$, initial weight $\mathbf{w}_0$.
**Output:** Trained weight $\mathbf{w}$. Initialize weight $\mathbf{w} \leftarrow \mathbf{w}_0$
**for** $i = 1, ..., e_1$ **do**
   1). Sample a mini-batch $B$ of size $b$ from $S$
   2). $\mathbf{w} \leftarrow \mathbf{w} - \beta\big(\nabla L_B(\mathbf{w}) + \lambda\mathbf{w}\big)$
**end for**
**for** $j = 1, ..., e_2$ **do**
   4). Sample a mini-batch $B$ of size $b$ from $S$
   5). $\boldsymbol{\epsilon} \leftarrow \rho \cdot \text{sign}(1 - \alpha) \cdot \frac{\sum_j |\nabla L_B(\mathbf{w})_j|^{2\alpha}}{(\sum_j \nabla L_B(\mathbf{w})_j^2)^{\alpha+1}} \nabla L_B(\mathbf{w})^{\top}$
   6). $\mathbf{w} \leftarrow \mathbf{w} - \beta\big(\nabla L_B(\mathbf{w} + \boldsymbol{\epsilon}) + \lambda\mathbf{w}\big)$
**end for**
**Return:** $\mathbf{w}$

---

We first train the neural network with vanilla SGD for $e_1$ epochs, without applying the Rényi regularizer. The intuition is that the gradient-magnitude approximation underlying our method becomes more accurate only after the model has achieved a reasonable training loss/accuracy, so penalizing the Rényi term at the very beginning of training is unnecessary and may even be harmful. Once the model reaches this warm-up stage, we activate the Rényi regularizer. For each mini-batch $B$, we compute the loss $L_B(\mathbf{w})$ and its gradient $\nabla L_B(\mathbf{w})$. We then construct the perturbation $\boldsymbol{\epsilon}$ according to Eq. 13 and form the perturbed parameters $\mathbf{w} + \boldsymbol{\epsilon}$. Next, we evaluate the gradient at the perturbed point, $\nabla L_B(\mathbf{w} + \boldsymbol{\epsilon})$, and perform a gradient-descent step on the original parameters $\mathbf{w}$ using this gradient. This procedure is structurally identical to SAM (Foret et al., 2020) and ASAM (Kwon et al., 2021); the only difference lies in how the perturbation $\boldsymbol{\epsilon}$ is computed, which in our case is defined by the Rényi sharpness objective in Eq. 13.

We set $\rho$ for SAM and Eigen-SAM as 0.05 for CIFAR10 and 0.1 for CIFAR100, and $\rho$ for ASAM as 0.5 for CIFAR10 and 1.0 for CIFAR100. $\eta$ for ASAM is set to 0.01. $\rho$ and $\alpha$ for RSAM is described in Table. 5 and Table. 6. The mini-batch size is set to 128. The number of epochs is set to 200 for SGD, SAM, ASAM, Eigen-SAM, and RSAM. Although prior work recommends training SGD for 400 epochs to assess improvements under a matched compute budget, RSAM introduces the regularizer only after a warm-up period, so compute parity no longer holds. Moreover, those studies have already shown performance superior to 400-epoch SGD. Consequently, our experiments are not strictly designed under equal-compute conditions. Momentum and weight decay coefficient are set to 0.9 and 0.0005, respectively. Cosine learning rate decay is s adopted with an initial learning rate of 0.1. Also, random cropping, padding by four pixels, normalization and random horizontal flip are applied for data augmentation. As label smoothing is not adopted in Eigen-SAM, all experiments are conducted without label smoothing.

For the evaluations at a larger scale, we compare the performance of SGD, SAM, ASAM, Eigen-SAM, and RSAM on TinyImageNet. We apply $\rho = 0.05$ for SAM and Eigen-SAM and $\rho = 1.0$ for ASAM. $\rho$ for RSAM is set to . The number of training epochs are all set to 100. We use a mini-batch size of 128, an initial learning rate of 0.2, and SGD optimizer with weight decay coefficient of 0.0001. Other hyperparameters are the same as those of CIFAR-10/100 tests.

All the hyper-parameters are summarized in Table 5, Table 6, and Table 7.

### J.4 RÉNYI ENTROPY COMPUTATION SETUP

The Rényi entropy is computed on the subset of the training dataset. For the CIFAR10 and CIFAR100 datasets, we randomly sample 2000 samples to compute Rényi entropy (1000 for ViT on CIFAR10), and for the TinyImageNet dataset, we randomly sample 1000 samples. Batch size is set to 128. $l = 100$ and $m = 15$ are set for the Rényi entropy estimation algorithm. The Rényi order is

Table 5: Hyper-parameters of Sharpness-aware Minimization on CIFAR10

| Algorithm | Model | Momen-tum | LR | SGD Epochs | SAM Epochs | Batch Size | Weight Decay | $\rho$ | $\eta$ | $\alpha$ |
|---|---|---|---|---|---|---|---|---|---|---|
| **SGD** | ResNet20 | 0.9 | 0.1 | 200 | 0 | 128 | 0.0005 | 0 | 0 | 0 |
| | ResNet56 | 0.9 | 0.1 | 200 | 0 | 128 | 0.0005 | 0 | 0 | 0 |
| | WideResNet-28-10 | 0.9 | 0.1 | 200 | 0 | 128 | 0.0005 | 0 | 0 | 0 |
| **SAM** | ResNet20 | 0.9 | 0.1 | 0 | 200 | 128 | 0.0005 | 0.05 | 0 | 0 |
| | ResNet56 | 0.9 | 0.1 | 0 | 200 | 128 | 0.0005 | 0.05 | 0 | 0 |
| | WideResNet-28-10 | 0.9 | 0.1 | 0 | 200 | 128 | 0.0005 | 0.05 | 0 | 0 |
| **ASAM** | ResNet20 | 0.9 | 0.1 | 0 | 200 | 128 | 0.0005 | 0.5 | 0.01 | 0 |
| | ResNet56 | 0.9 | 0.1 | 0 | 200 | 128 | 0.0005 | 0.5 | 0.01 | 0 |
| | WideResNet-28-10 | 0.9 | 0.1 | 0 | 200 | 128 | 0.0005 | 0.5 | 0.01 | 0 |
| **Eigen-SAM** | ResNet20 | 0.9 | 0.1 | 0 | 200 | 128 | 0.0005 | 0.05 | 0 | 0.2 |
| | ResNet56 | 0.9 | 0.1 | 0 | 200 | 128 | 0.0005 | 0.05 | 0 | 0.2 |
| | WideResNet-28-10 | 0.9 | 0.1 | 0 | 200 | 128 | 0.0005 | 0.05 | 0 | 0.2 |
| **FSAM** | ResNet20 | 0.9 | 0.1 | 0 | 200 | 128 | 0.0005 | 0.1 | 1.0 | 0 |
| | ResNet56 | 0.9 | 0.1 | 0 | 200 | 128 | 0.0005 | 0.1 | 1.0 | 0 |
| | WideResNet-28-10 | 0.9 | 0.1 | 0 | 200 | 128 | 0.0005 | 0.1 | 1.0 | 0 |
| **SSAM** | ResNet20 | 0.9 | 0.1 | 0 | 200 | 128 | 0.0005 | 0.2 | 0.0 | 0 |
| | ResNet56 | 0.9 | 0.1 | 0 | 200 | 128 | 0.0005 | 0.2 | 0.0 | 0 |
| | WideResNet-28-10 | 0.9 | 0.1 | 0 | 200 | 128 | 0.0005 | 0.1 | 0.0 | 0 |
| **RSAM** | ResNet20 | 0.9 | 0.1 | 5 | 195 | 128 | 0.0005 | 0.65 | 0 | 1.2 |
| | ResNet56 | 0.9 | 0.1 | 5 | 195 | 128 | 0.0005 | 0.8 | 0 | 1.2 |
| | WideResNet-28-10 | 0.9 | 0.1 | 5 | 195 | 128 | 0.0005 | 0.3 | 0 | 1.05 |

Table 6: Hyper-parameters of Sharpness-aware Minimization on CIFAR100

| Algorithm | Model | Momen-tum | LR | SGD Epochs | SAM Epochs | Batch Size | Weight Decay | $\rho$ | $\eta$ | $\alpha$ |
|---|---|---|---|---|---|---|---|---|---|---|
| **SGD** | ResNet20 | 0.9 | 0.1 | 200 | 0 | 128 | 0.0005 | 0 | 0 | 0 |
| | ResNet56 | 0.9 | 0.1 | 200 | 0 | 128 | 0.0005 | 0 | 0 | 0 |
| | WideResNet-28-10 | 0.9 | 0.1 | 200 | 0 | 128 | 0.0005 | 0 | 0 | 0 |
| **SAM** | ResNet20 | 0.9 | 0.1 | 0 | 200 | 128 | 0.0005 | 0.1 | 0 | 0 |
| | ResNet56 | 0.9 | 0.1 | 0 | 200 | 128 | 0.0005 | 0.1 | 0 | 0 |
| | WideResNet-28-10 | 0.9 | 0.1 | 0 | 200 | 128 | 0.0005 | 0.1 | 0 | 0 |
| **ASAM** | ResNet20 | 0.9 | 0.1 | 0 | 200 | 128 | 0.0005 | 1.0 | 0.01 | 0 |
| | ResNet56 | 0.9 | 0.1 | 0 | 200 | 128 | 0.0005 | 1.0 | 0.01 | 0 |
| | WideResNet-28-10 | 0.9 | 0.1 | 0 | 200 | 128 | 0.0005 | 1.0 | 0.01 | 0 |
| **Eigen-SAM** | ResNet20 | 0.9 | 0.1 | 0 | 200 | 128 | 0.0005 | 0.1 | 0 | 0.2 |
| | ResNet56 | 0.9 | 0.1 | 0 | 200 | 128 | 0.0005 | 0.1 | 0 | 0.2 |
| | WideResNet-28-10 | 0.9 | 0.1 | 0 | 200 | 128 | 0.0005 | 0.1 | 0 | 0.2 |
| **FSAM** | ResNet20 | 0.9 | 0.1 | 0 | 200 | 128 | 0.0005 | 0.1 | 1.0 | 0 |
| | ResNet56 | 0.9 | 0.1 | 0 | 200 | 128 | 0.0005 | 0.1 | 1.0 | 0 |
| | WideResNet-28-10 | 0.9 | 0.1 | 0 | 200 | 128 | 0.0005 | 0.1 | 1.0 | 0 |
| **SSAM** | ResNet20 | 0.9 | 0.1 | 0 | 200 | 128 | 0.0005 | 0.5 | 0.0 | 0 |
| | ResNet56 | 0.9 | 0.1 | 0 | 200 | 128 | 0.0005 | 0.5 | 0.0 | 0 |
| | WideResNet-28-10 | 0.9 | 0.1 | 0 | 200 | 128 | 0.0005 | 0.2 | 0.0 | 0 |
| **RSAM** | ResNet20 | 0.9 | 0.1 | 5 | 195 | 128 | 0.0005 | 0.76 | 0 | 1.1 |
| | ResNet56 | 0.9 | 0.1 | 5 | 195 | 128 | 0.0005 | 0.9 | 0 | 1.1 |
| | WideResNet-28-10 | 0.9 | 0.1 | 5 | 195 | 128 | 0.0005 | 0.7 | 0 | 1.05 |

chosen from {0.0001, 0.01, 0.03, 0.06, 0.1, 0.2, 0.3, 0.4, 0.5, 0.6, 0.7, 0.8, 0.9, 0.99, 0.999, 1.001, 1.01, 1.1, 1.2, 1.3, 1.4, 1.5, 1.6, 1.7, 1.8, 1.9, 2.0, 2.1, 2.2, 2.3, 2.4, 2.5, 2.6, 2.7, 2.8, 2.9, 3}. Due to the fact that training cannot guarantee convergence exactly to a strict local minimum, negative eigenvalues are inevitable, which can cause numerical pathologies for the Rényi entropy as $\alpha \to 1$. Therefore, when assessing how $\alpha$ affects the correlation between Rényi entropy and generalization, we restrict $\alpha$ to $(0, 0.9)$ and $(1.2, 3.0]$. Within these ranges, computing the Rényi entropy is stable and free of anomalies. During our analysis of the sharpness–generalization correlation, we vary $\alpha$ and plot the sharpness that attains the highest correlation coefficient.

Table 7: Hyper-parameters of Sharpness-aware Minimization on TinyImageNet

| Algorithm | Model | Momen -tum | LR | SGD Epochs | SAM Epochs | Batch Size | Weight Decay | $\rho$ | $\eta$ | $\alpha$ |
|---|---|---|---|---|---|---|---|---|---|---|
| SGD | ResNet50 | 0.9 | 0.2 | 100 | 0 | 128 | 0.0001 | 0 | 0 | 0 |
| SAM | ResNet50 | 0.9 | 0.2 | 0 | 100 | 128 | 0.0001 | 0.05 | 0 | 0 |
| ASAM | ResNet50 | 0.9 | 0.2 | 0 | 100 | 128 | 0.0001 | 1.0 | 0.01 | 0 |
| FSAM | ResNet50 | 0.9 | 0.2 | 0 | 100 | 128 | 0.0001 | 0.5 | 0.1 | 0 |
| RSAM | ResNet50 | 0.9 | 0.2 | 20 | 80 | 128 | 0.0001 | 1.25 | 0 | 1.1 |

*Note.* In practice, we train with SGD until the validation Top-1 accuracy exceeds 30%, then switch to RSAM; this typically occurs around epoch 20.

Table 8: Hyper-parameters of Sharpness-aware Minimization on ViT-B/16 Finetuning

| Algorithm | Dataset | Momen -tum | LR | SGD Epochs | SAM Epochs | Batch Size | Weight Decay | $\rho$ | $\eta$ | $\alpha$ |
|---|---|---|---|---|---|---|---|---|---|---|
| SGD | CIFAR10 | 0.9 | 0.01 | 20 | 0 | 128 | 0.0005 | 0 | 0 | 0 |
| | CIFAR100 | 0.9 | 0.01 | 20 | 0 | 128 | 0.0005 | 0 | 0 | 0 |
| SAM | CIFAR10 | 0.9 | 0.01 | 0 | 20 | 128 | 0.0005 | 0.05 | 0 | 0 |
| | CIFAR100 | 0.9 | 0.01 | 0 | 20 | 128 | 0.0005 | 0.1 | 0 | 0 |
| ASAM | CIFAR10 | 0.9 | 0.01 | 0 | 20 | 128 | 0.0005 | 0.5 | 0.01 | 0 |
| | CIFAR100 | 0.9 | 0.01 | 0 | 20 | 128 | 0.0005 | 1.0 | 0.01 | 0 |
| FSAM | CIFAR10 | 0.9 | 0.01 | 0 | 20 | 128 | 0.0005 | 0.1 | 1.0 | 0 |
| | CIFAR100 | 0.9 | 0.01 | 0 | 20 | 128 | 0.0005 | 0.1 | 1.0 | 0 |
| RSAM | CIFAR10 | 0.9 | 0.01 | 2 | 18 | 128 | 0.0005 | 0.8 | 0 | 1.3 |
| | CIFAR100 | 0.9 | 0.01 | 2 | 18 | 128 | 0.0005 | 0.6 | 0 | 1.1 |

## J.5 COMPUTATION OF OTHER SHARPNESS MEASURES

We detail how the remaining sharpness measures are computed. Following the public implementation of Dziugaite et al. (2020), we compute the PAC-Bayes–based measure and estimate the Hessian trace via Hutchinson's trick (Eq. 9). The Fisher–Rao norm is computed as in Petzka et al. (2021). For SAM and ASAM, we sweep $\rho \in$

$$\{10^{-6}, 3 \times 10^{-6}, 10^{-5}, 3 \times 10^{-5}, 10^{-4}, 3 \times 10^{-4}, 10^{-3}, 3 \times 10^{-3}, 10^{-2}, 3 \times 10^{-2}, 10^{-1}, 0.3, 1\}$$

and report the sharpness at the value of $\rho$ that yields the highest correlation with generalization. Because SAM/ASAM sharpness is defined with respect to the entire dataset, we evaluate it on a subsample of 1,000 training examples using a single batch of size 1,000 (rather than mini-batches). Data augmentation is disabled during these computations. We evaluate sharpness only for perturbations that do not induce a large increase in the loss. Once the loss rise becomes substantial, the perturbed point should no longer be regarded as residing in the neighborhood of the minimum. For instance, when the unperturbed loss is approximately 0.001 (accuracy approximately 100%) but rises to 5.2 after perturbation (accuracy dropping to 20% or lower), the perturbation has evidently moved the parameters outside the minimum's basin. Notably, the formulations of SAM and ASAM presuppose that weight perturbations remain within the local neighborhood of the minimum.

# K FULL RESULTS

In this section, we report all the results of the tasks in the main body.

## K.1 HESSIAN SPECTRUM

In this section, we provide some spectra of the trained models in the correlation validation experiments, including ResNet18 and ResNet34 on CIFAR10 and ResNet18 and ResNet34 on CIFAR100.

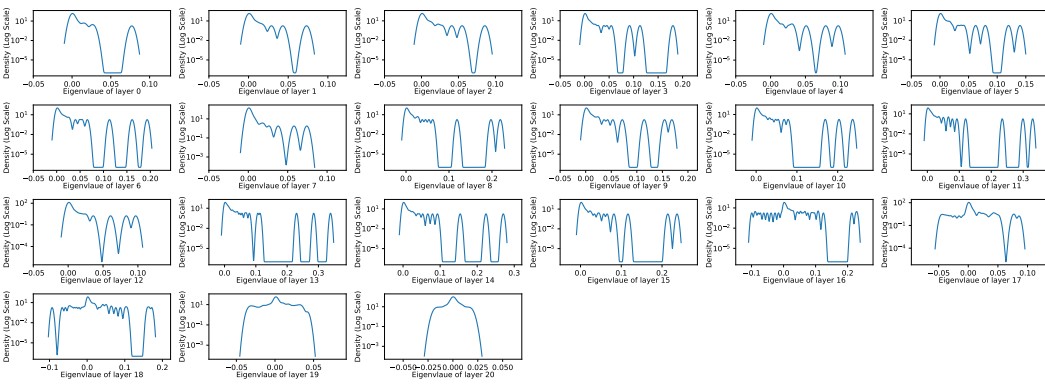

Figure 4: Spectrum of ResNet18 on CIFAR10.

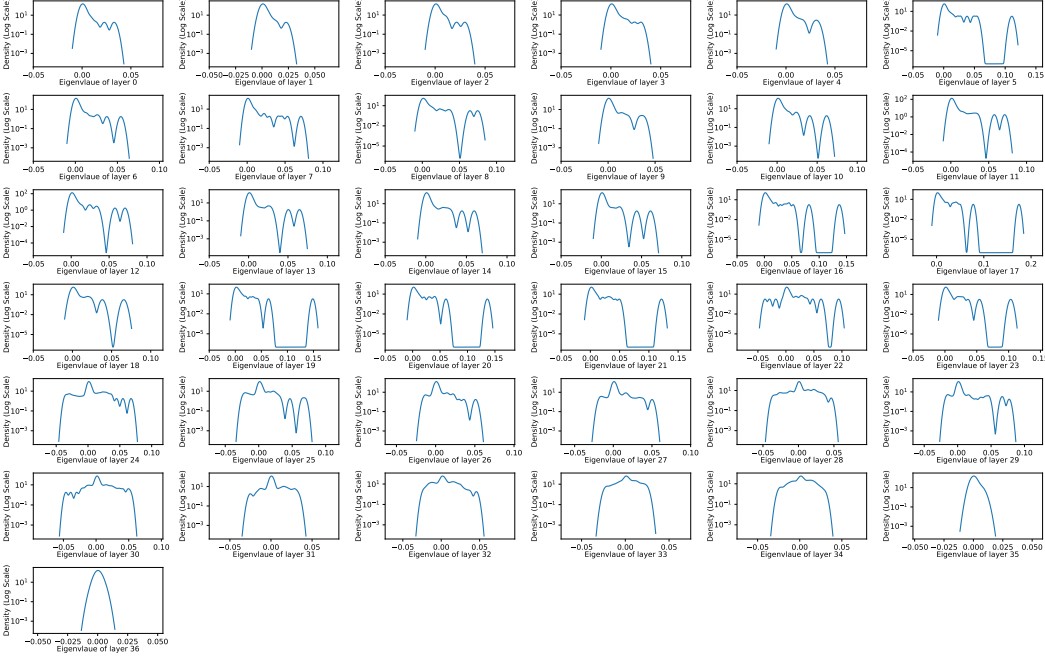

Figure 5: Spectrum of ResNet34 on CIFAR10.

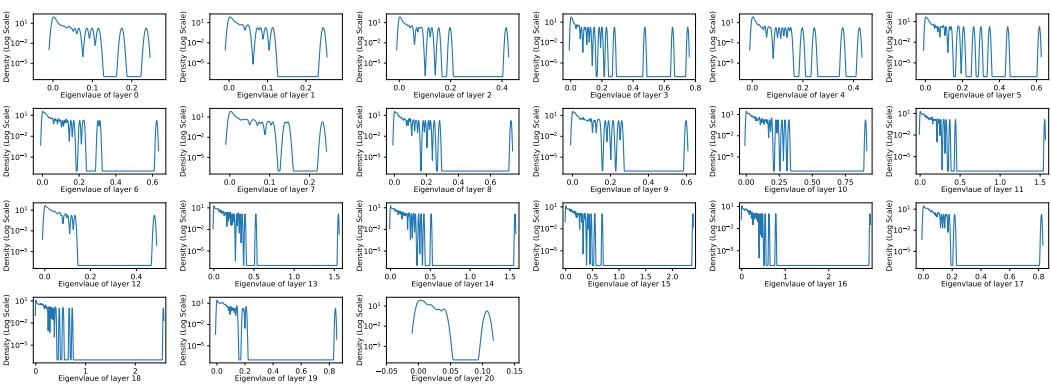

Figure 6: Spectrum of ResNet18 on CIFAR100.

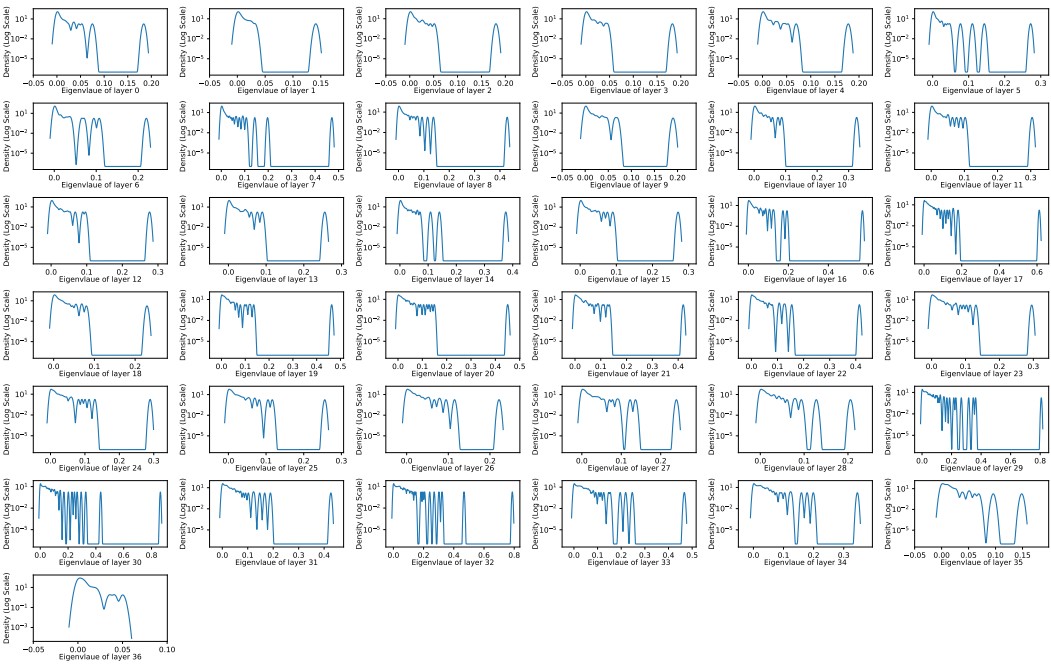

Figure 7: Spectrum of ResNet34 on CIFAR100.

## K.2 Correlation Between Rényi Sharpness and Generalization

In this section, we provide the figures about the correlation between generalization and multiple sharpness measures. We can find that Rényi sharpness is strongly correlated with generalization than the other measures.

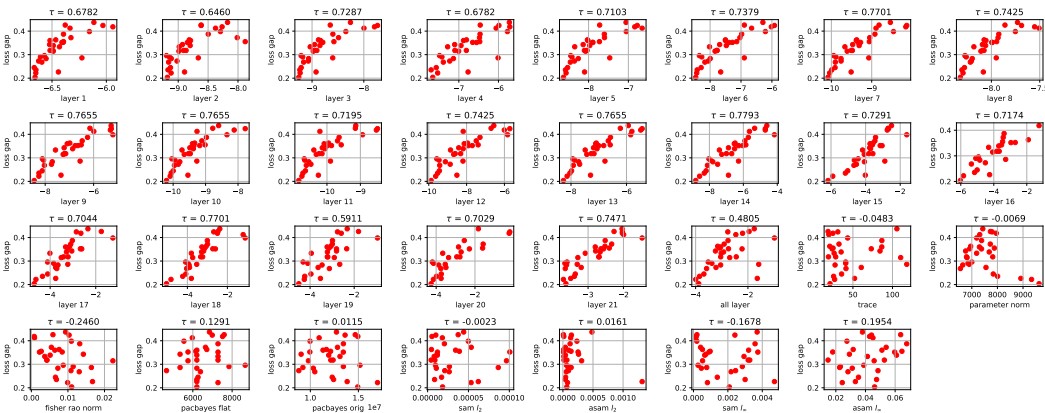

Figure 8: ResNet18 on CIFAR10, The layer 1 to all layer subplots correspond to the Rényi sharpness measure.

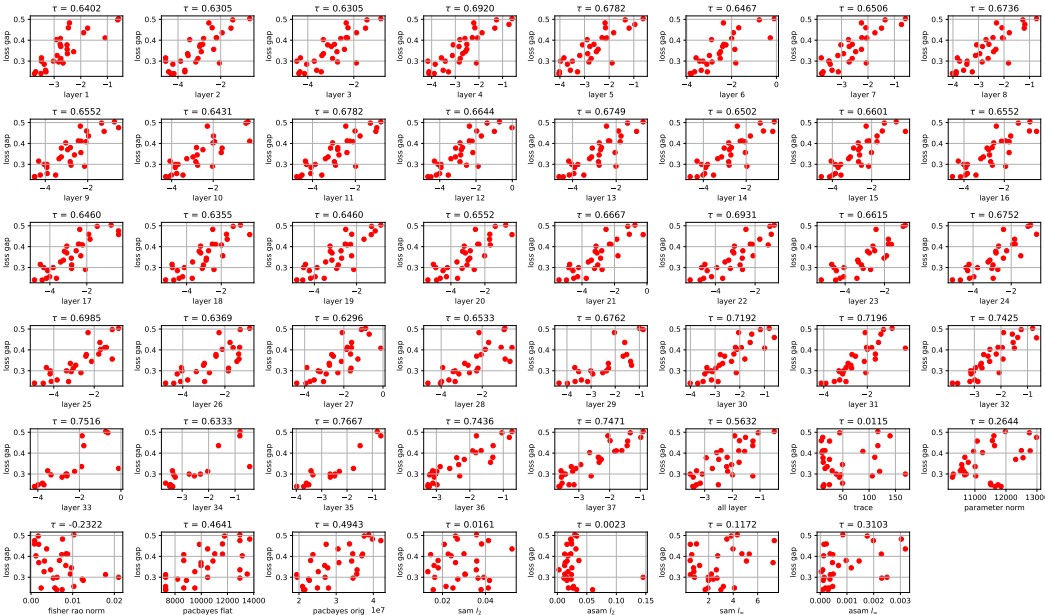

Figure 9: ResNet34 on CIFAR10, The layer 1 to all layer subplots correspond to the Rényi sharpness measure.

| Task | CIFAR10/ResNet18 | CIFAR10/ResNet34 | CIFAR10/ViT | CIFAR100/ResNet18 | CIFAR100/ResNet34 | TinyImageNet/ResNet18 |
|---|---|---|---|---|---|---|
| Correlation coefficient | -0.2092 | -0.2966 | -0.1954 | -0.3149 | -0.5310 | -0.6063 |

Table 9: Correlation coefficient between $\log \det \mathbf{H}$ and generalization gap across tasks. $\mathbf{H}$ is the Hessian matrix of the training loss with respect to the whole weights in the model.

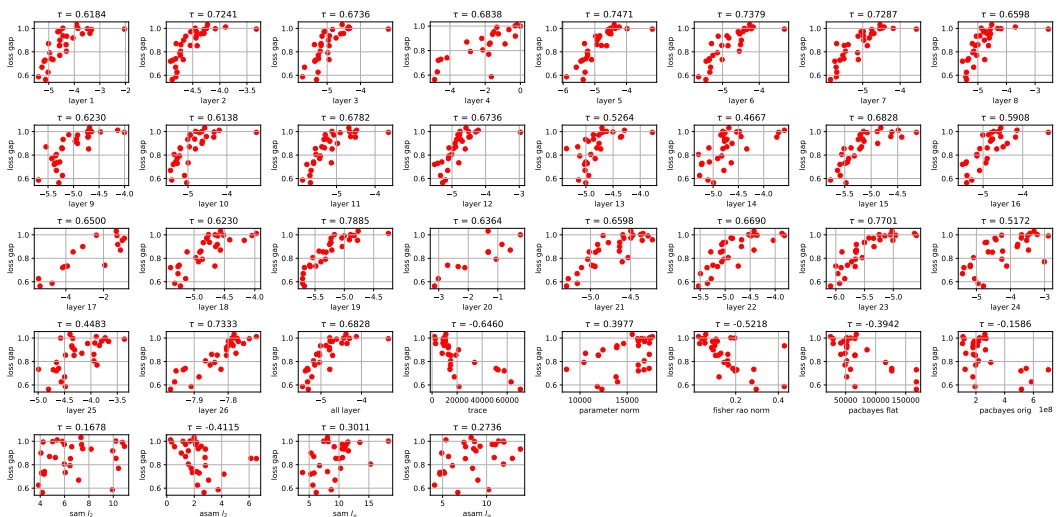

Figure 10: ViT on CIFAR10, The layer 1 to all layer subplots correspond to the Rényi sharpness measure.

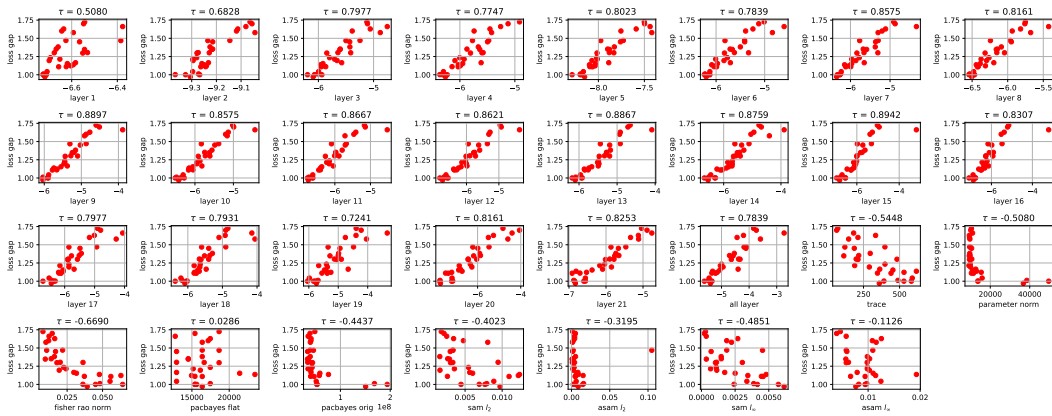

Figure 11: ResNet18 on CIFAR100, The layer 1 to all layer subplots correspond to the Rényi sharpness measure.

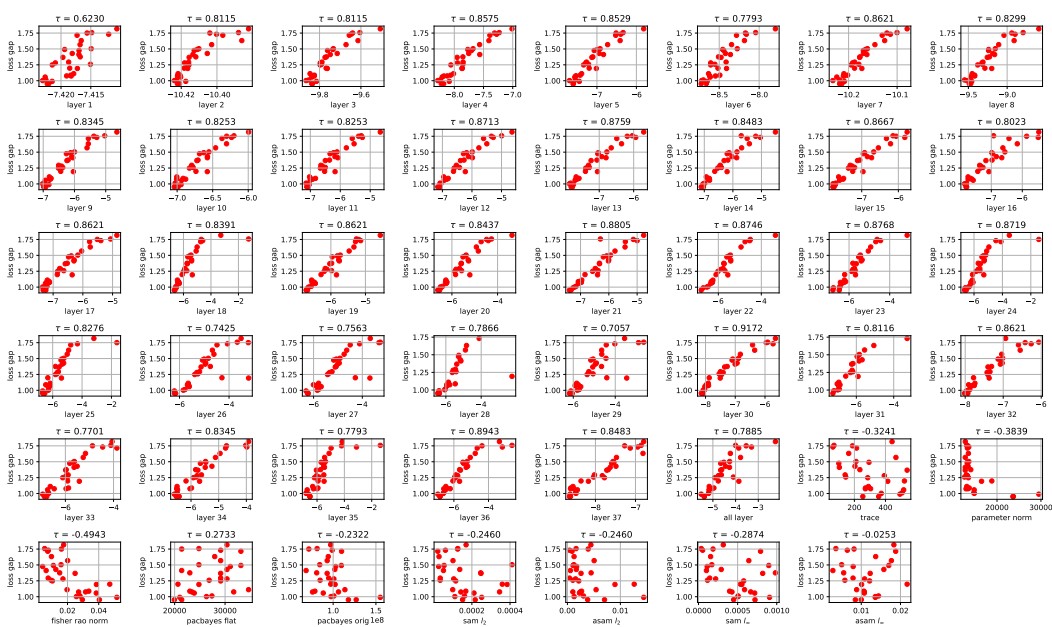

Figure 12: ResNet34 on CIFAR100, The layer 1 to all layer subplots correspond to the Rényi sharpness measure.

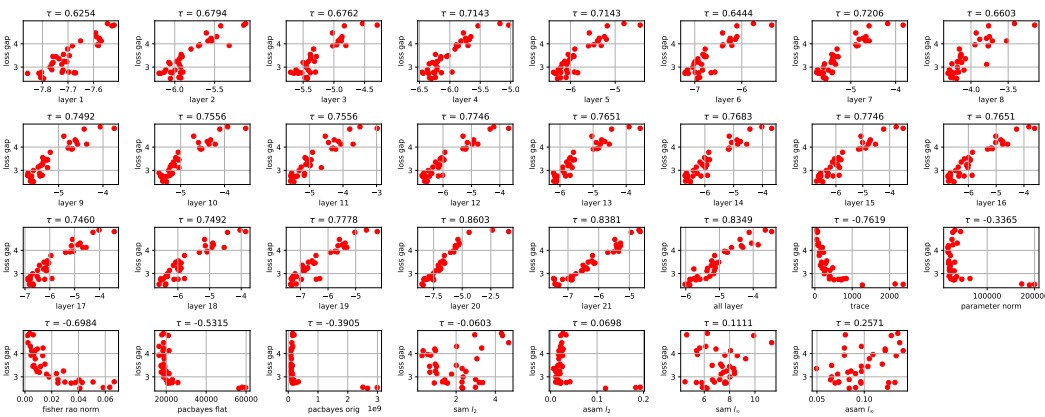

Figure 13: ResNet18 on TinyImageNet, The layer 1 to all layer subplots correspond to the Rényi sharpness measure.

### K.3 MORE RESULTS FOLLOWING ANDRIUSHCHENKO ET AL. (2023)

#### K.3.1 MORE TRAINING RECIPE FOR RESNET18 ON CIFAR10

In previous work, Andriushchenko et al. (2023) showed that their sharpness measure correlates strongly with generalization only within certain hyperparameter subsets or sub-groups. To perform a similar test, we extend our standard ResNet-18/CIFAR-10 setup by introducing two additional hyperparameter dimensions: with/without mixup ($\alpha = 0.5$) (Zhang et al., 2017) and with/without standard augmentations combined with RandAugment (Cubuk et al., 2020). We then compute the Rényi sharpness of the last two layers and compare it with other sharpness measures. The results in Fig. 14 indicate that, even under these richer hyperparameter combinations, Rényi sharpness still exhibits a strong and consistent correlation with generalization.

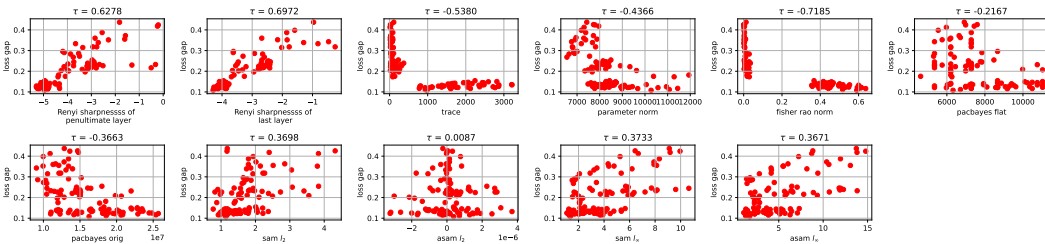

Figure 14: ResNet18 on CIFAR10 with more training configurations. The learning rate, batch size, optimizer, and weight decay are varied following standard ResNet-18-on-CIFAR-10 setups, and we further introduce variants with mixup ($\alpha = 0.5$) (Zhang et al., 2017) and standard augmentations combined with RandAugment (Cubuk et al., 2020), resulting in four times as many models as in the standard setting.

#### K.3.2 PRETRAINING VIT-B/16 ON IMAGENET-1K

Following Andriushchenko et al. (2023), we evaluate ViT models from Steiner et al. (2021), using ViT-B/16-224 weights. Those were trained from scratch on ImageNet-1k for 300 epochs with different hyperparameter settings, and subsequently fine tuned on the same dataset for 20.000 steps with 2 different learning rates. The different hyperparameters include augmentations, weight decay, and stochastic depth / dropout, leading to a rich pool of 56 models. As shown in Figure 15, Rényi sharpness still exhibits a strong and consistent correlation with generalization, while others not.

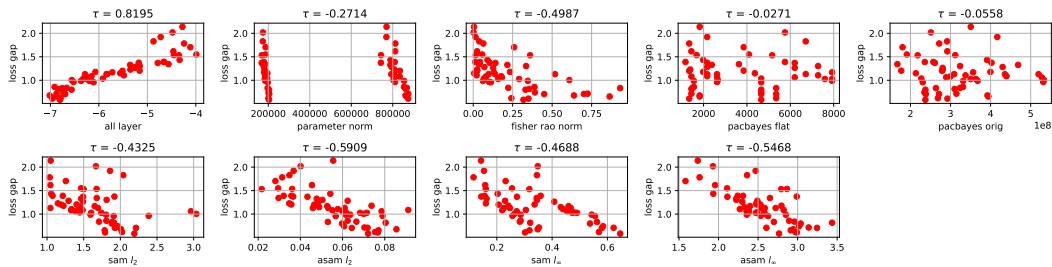

Figure 15: ViT-B/16 trained from scratch on ImageNet-1k. We show for 56 models from Steiner et al. (2021) the generalization gap vs. various sharpness measures. Overall, Rényi sharpness is still strongly correlated with generalization than the other measures.

#### K.3.3 FINE-TUNING ON IMAGENET-1K FROM CLIP

We also follow the experiments that investigate fine-tuning from CLIP Radford et al. (2021). We study the pool of classifiers obtained by Wortsman et al. (2022), who fine-tuned a CLIP ViT-B/32 model on ImageNet multiple times by randomly selecting training hyperparameters, including learning rate, number of epochs, weight decay, label smoothing, and augmentations. We compute the Rényi sharpness of the last layer within the ViT-B/32 model, and compare it with other sharpness

measures. One can confirm from Fig. 16 that Rényi sharpness still exhibits a strong and consistent correlation with generalization, compared to the other measures.

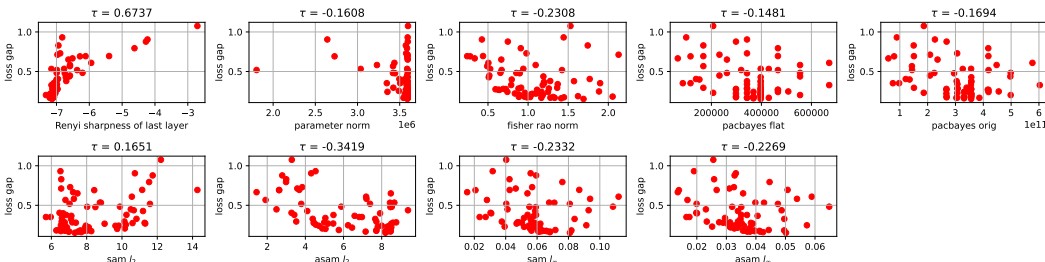

Figure 16: Fine-tuning CLIP ViT-B/32 on ImageNet-1k. We show for 72 models from Wortsman et al. (2022) the generalization gap on ImageNet vs multiple sharpness measures.

## K.4 CORRELATION COEFFICIENT AND RÉNYI ORDER $\alpha$

In this section, we report statistics of Kendall's $\tau$ under different Rényi orders. The order $\alpha$ is varied following the guidelines in Section 4.1. We compute Kendall's $\tau$ for each layer and report the average correlation of all layers. The heatmap in Fig. 17 shows that $\alpha = 0.5$ for $0 < \alpha < 1$ and $\alpha = 1.5$ for $\alpha > 1$ are consistently robust across tasks.

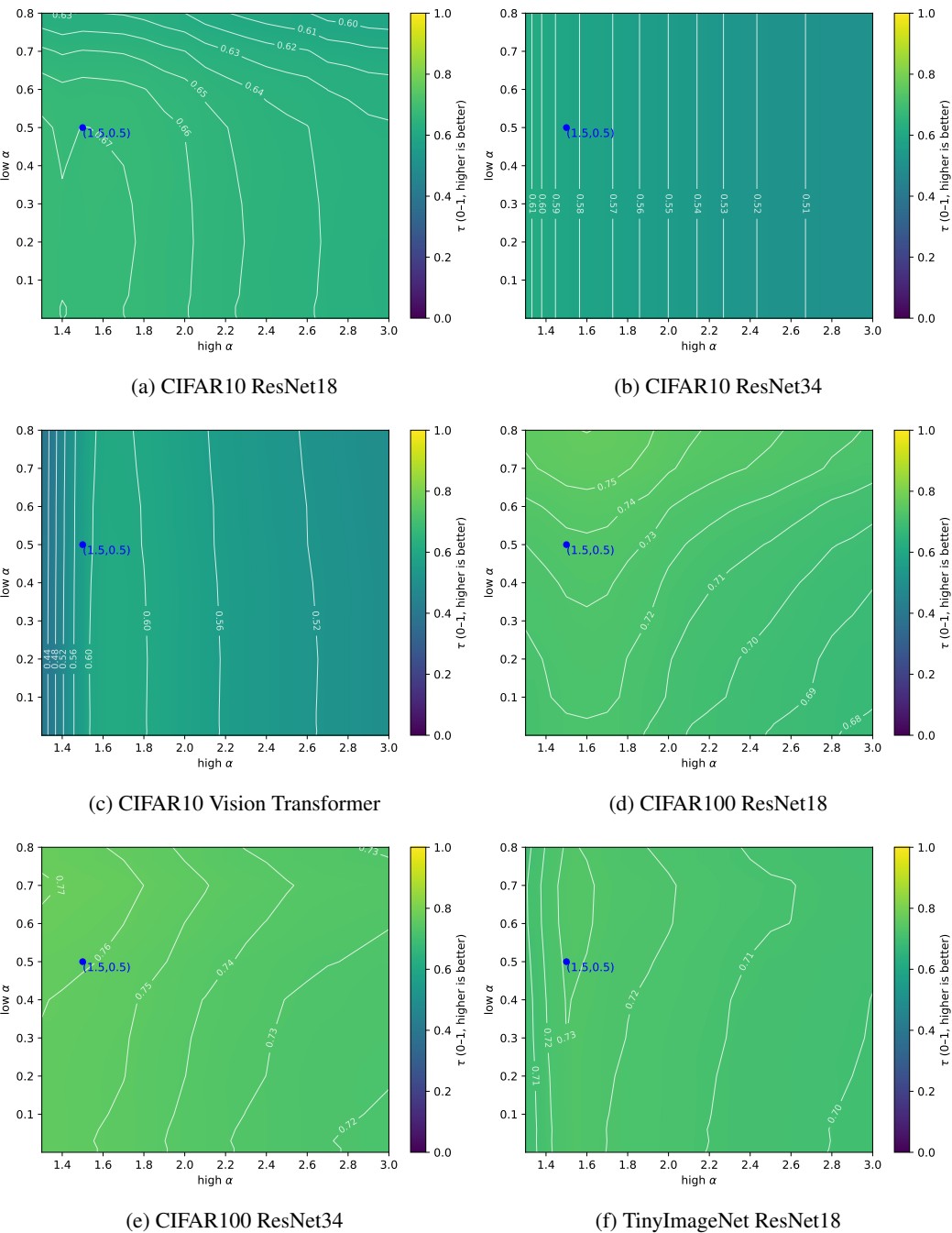

Figure 17: Correlation Coefficient and Rényi Order $\alpha$

### K.5 SCATTER PLOT OF CORRELATION COEFFICIENT AND RÉNYI ORDER $\alpha$

In this section, we provide all the correlation coefficient $\tau$ under different $\alpha$ across multiple tasks:

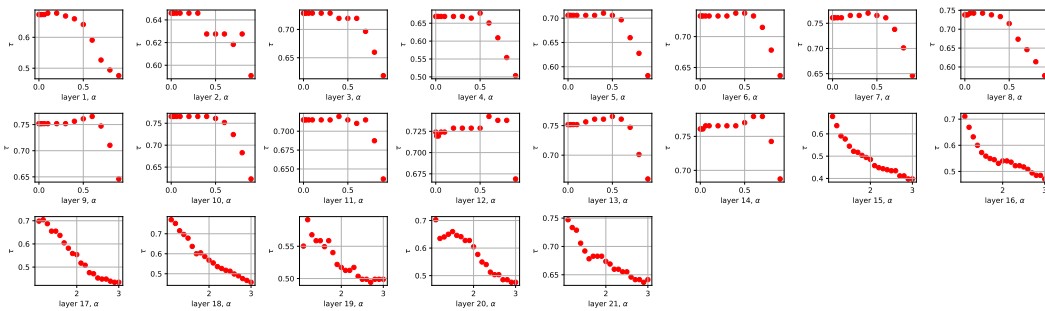

Figure 18: ResNet18 on CIFAR10, we plot the correlation coefficient $\tau$ vs Rényi order $\alpha$.

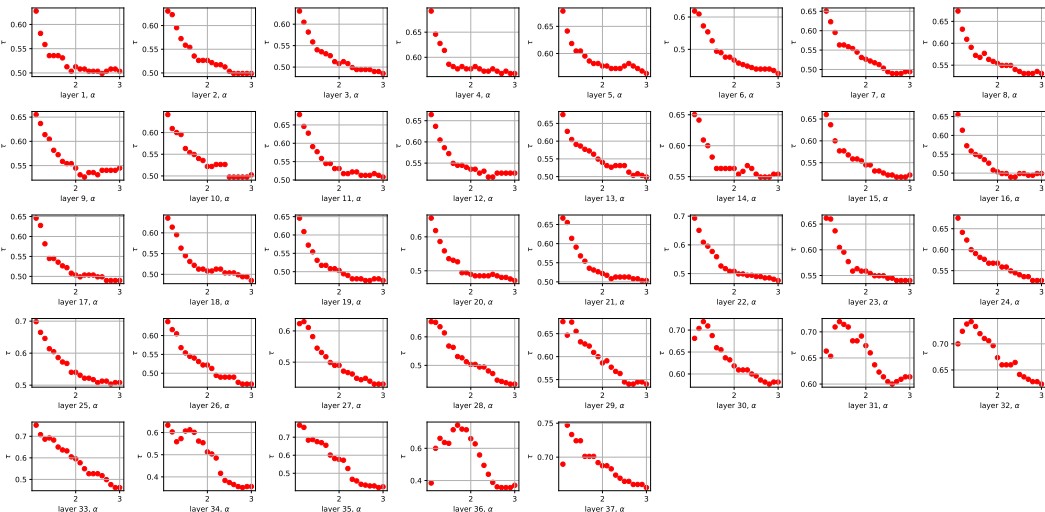

Figure 19: ResNet34 on CIFAR10, we plot the correlation coefficient $\tau$ vs Rényi order $\alpha$.

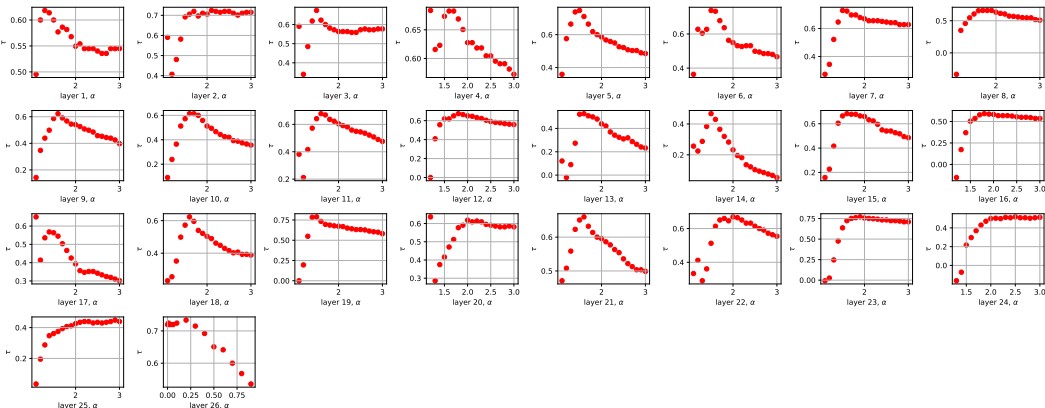

Figure 20: ViT on CIFAR10, we plot the correlation coefficient $\tau$ vs Rényi order $\alpha$.

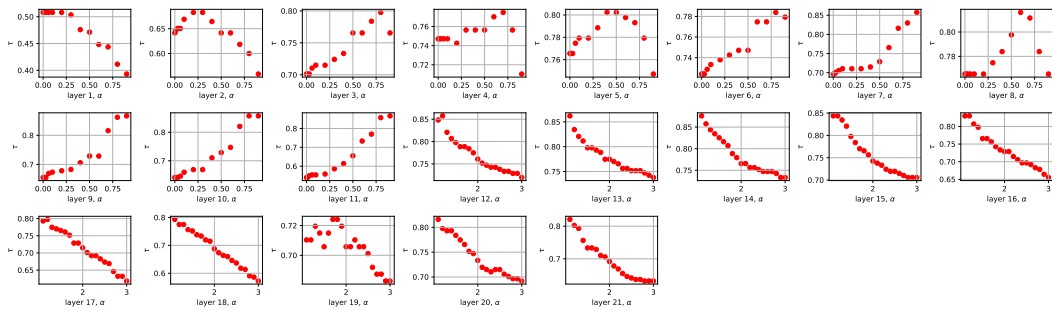

Figure 21: ResNet18 on CIFAR100, we plot the correlation coefficient $\tau$ vs Rényi order $\alpha$.

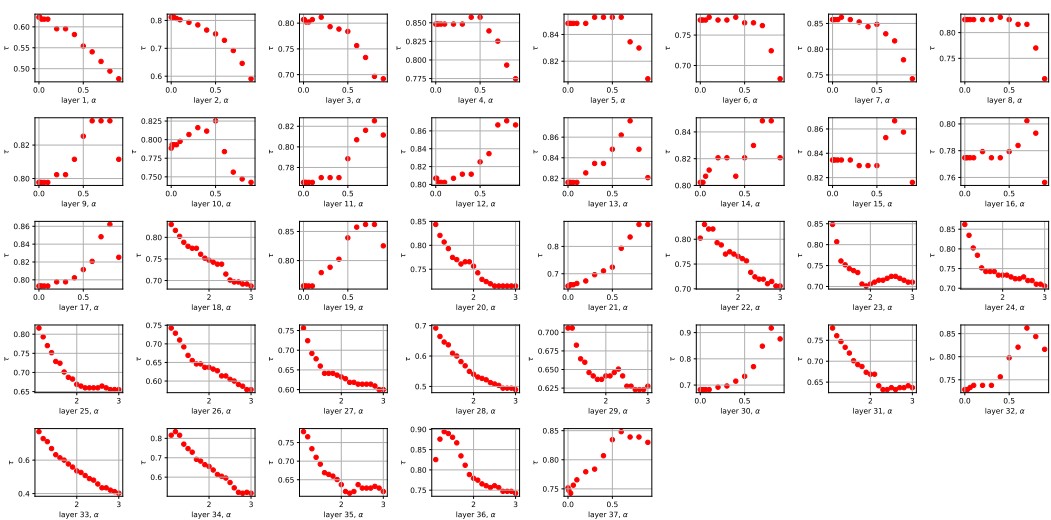

Figure 22: ResNet34 on CIFAR100, we plot the correlation coefficient $\tau$ vs Rényi order $\alpha$.

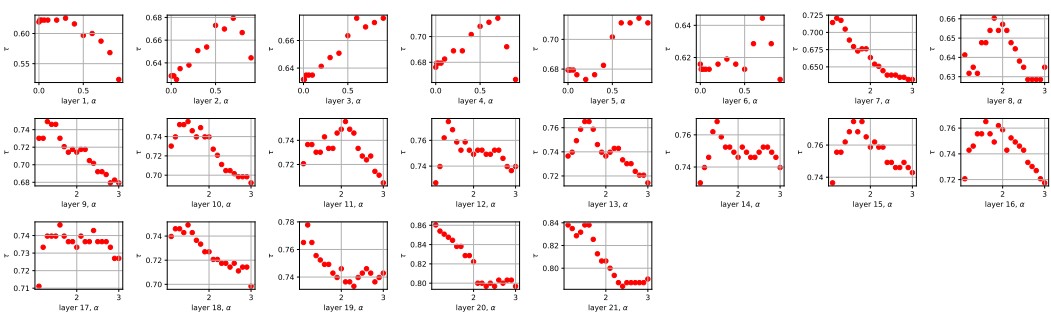

Figure 23: ResNet18 on TinyImageNet, we plot the correlation coefficient $\tau$ vs Rényi order $\alpha$.

## L    LIMITATION

- The generalization bounds in our work relies on homogeneity of the activation function, which holds for ReLU networks and approximately holds for GELU networks. Extending the analysis for other activations is a both interesting and important direction.

- Our proposed RSAM algorithm uses an approximation to Rényi sharpness for simplicity, a tighter approximation or surrogate may further improve generalization.

## M    BROADER IMPACTS

Our work aims to advance the theoretical understanding of network generalization, with the anticipation that theoretical insights can guide future designs of network optimization methods. There are no ethically related issues or negative societal consequences in our work.

