# OpenReview forum: "Rényi Sharpness: A Novel Sharpness that Strongly Correlates with Generalization"
_ICLR.cc/2026/Conference — ICLR 2026 Poster_

### Official Review · Reviewer_NtAE · 2025-10-22

**Soundness:** 1
**Presentation:** 3
**Contribution:** 1
**Rating:** 2
**Confidence:** 5

**Summary:**

The authors propose a novel sharpness measure to predict generalization performance: Renyi sharpness. They provide generalization bounds based on this measure, and experimental studies on CIFAR and TinyImagenet. Further, they also introduce a variant of the sharpness-aware minimization (SAM) algorithm that is based on an approximation of the Renyi sharpness, with experiments on Cifar and TinyImageNet.

**Strengths:**

The paper aims to address a fundamental problem in deep learning: the poor correlation between many generalization measures, and the empirically observed generalization performance. Using the Renyi sharpness as generalization measure because it captures uniformity is novel (to the best of my knowledge). In the provided experiments, the Renyi sharpnes shows good correlation (albeit doubts remain, see weaknesses below).

**Weaknesses:**

I have strong doubts on the conclusions that can be drawn from the experiments provided by the authors. In particular, I have concerns regarding the considered sharpness setup, the baselines that were used, the insightsfulness of the generalization bounds, and the effectiveness of the provided RSAM algorithm.

1. Sharpness setup:
The present study on the connection between Renyi sharpness and generalization is much less extensive than other work investigating the relationship between sharpness measures and generalization (e.g. Andriushchenko et al [1]). In [1], the authors also found setups - similar to the ones considered in this work - where sharpness _could_ predict generalization. However, Andriushchenko et al [1] showed that this might only be true for certain subgroups of the training parameters. In particular, when considering their “modern” setup (ViTs, ImageNet-scale, varied pretraining+finetuning schemes, OOD generalization and transfer learning, Language and Vision tasks,  …), the correlation disappeared. To show the effectiveness of the Renyi sharpness, experimental evidence on the scale of [1] would be necessary.



2. Sharpness baselines and tuning of alpha:
The provided results are for extensively tuned alpha values, whereas the baselines are apparently not tuned (e.g. the $\rho$ values for SAM and ASAM). Further, many sharpness variants (e.g. the ones from [1]) are omitted from the study, and there are no details on how exactly the baseline measures are computed or what they mean.



3. Generalization bounds:
The generalization bound in theorem 3.2 is based on upper-bounding a generalization bound from [2] by upper-bounding the log-determinant of the Hessian with the Renyi sharpness. Using the log-determinant of the hessian (like done in [2]) would thus be better motivated by the bound. Same for theorem 3.3, where the log-determinant is also upper-bounded by the Renyi sharpness. There is thus a disconnect between the generalization bounds and the measure used.


4. RSAM:
The provided RSAM algorithm seems to be brittle (as admitted by the authors: warmup required, length depends on task) and barely brings improvements beyond error bars over ASAM. Further, from Tables 4 and 5 it seems that $\rho$ has been tuned for RSAM, but not for the other SAM variants. Finally, there exists a plethora of SAM variants, that are all ignored in the comparison, and the study is limited in terms of dataset size (no ImageNet scale) and models (no ViTs).


5. More comments:

- reparametrization-invariant sharpness measures (w.r.t. layerwise rescaling) have been investigated before (see e.g. [1]), and I assume that the ASAM measure used in the provided study is one variant of the measures in [1]. So reparametrization invariance alone cannot be the reason for the (potential) success of a novel sharpness measure. I recommend discussing this when arguing about reparametrization invariance.
- the authors claim “in our opinion, what matters the most for characterizing the generalization
is the extent of the spread of the spectrum” when introducing the Renyi sharpness as generalization measure, but it is unclear to me where this opinion stems from.
- The authors describe how alpha might be chosen, depending on the spectrum, but do not elaborate how this choice might look like in practice, and then tune alpha extensively in their experimental section. Is there an intuition on when to chose low or high alpha? Is it necessary to manually inspect the spectrum and decide on multi-cluster vs uniform?



Typos:

several times (e.g. Line 147): denote vs donate

Line 215 Ler vs Let


[1]  Maksym Andriushchenko, Francesco Croce, Maximilian Müller, Matthias Hein, and Nicolas Flammarion. A modern look at the relationship between sharpness and generalization.

[2] Zhiwei Jia and Hao Su, Information-Theoretic Local Minima Characterization and Regularization, ICML 2020

**Questions:**

I do not have specific questions where an answer would change my opinion on the paper - I think significant changes, addressing the weaknesses outlined above, are necessary to convincingly argue in favour of Renyi sharpness:
- a much extended experimental setup, like in Andriushchenko et al
- more and better tuned baselines
- a practical way of choosing alpha, without the necessity of tuning it
- convincing arguments why the generalization bounds argue in favour of Renyi sharpness instead of log-det (H)
- convincing evidence that RSAM improves robustly for fair comparison (tuned rho) over baselines

---

> ### Author Response · Authors · 2025-11-26
>
> Dear reviewer NtAE,
>
> We sincerely thank the reviewers for their thoughtful and constructive comments and for finding this work novel with good results. Our point-by-point responses to all comments are provided as follows.
>
> **W1: Sharpness setup:  The present study on the connection between Renyi sharpness and generalization is much less extensive than other work investigating the relationship between sharpness measures and generalization (e.g. Andriushchenko et al [1]). In [1], the authors also found setups - similar to the ones considered in this work - where sharpness could predict generalization. However, Andriushchenko et al [1] showed that this might only be true for certain subgroups of the training parameters. In particular, when considering their “modern” setup (ViTs, ImageNet-scale, varied pretraining+finetuning schemes, OOD generalization and transfer learning, Language and Vision tasks, …), the correlation disappeared. To show the effectiveness of the Renyi sharpness, experimental evidence on the scale of [1] would be necessary.**
>
> A1: Thanks for the insightful comment.  First of all, we'd like to admit that we did not convey the motivation and the intuitive explanation of using Rényi sharpness for predicting generalization clearly, thus greatly lowered its persuasiveness. As a remedy, we have made big improvement on this in the revised version. Below is a brief intuitive explanation of why Renyi sharpness could correlates strongly and persistently with generalization, while other sharpness measures fail to do.
>
> A key observation of ours is what really matters for generalization is the *unevenness* (average spread) of the loss Hessian, which is capable of taking into account all three categories of eigenvalues: the top, the middle and the tail ones (as detailed in the revised version), while the max-eigenvalue sharpness only cares about the top eigenvalues and the trace sharpness only cares about the middle ones (in a rough sense). And the Adaptive sharpness (as in ASAM) can be seen as a variant of the max-eigenvalue sharpness or trace sharpness, thus inevitably inheriting some weakness of the latter measures.
>
> Second, regarding the concern that the correlation might only hold for certain subgroups while might disappear in all groups or in the large scale settings,  we have conducted  more extensive experiments as follows:
>
> 1) For the similar setting as [1] on CIFAR10/ResNet18 with multiple subgroups (with/without augmentation, with/without mixup, various learning rates, batch sizes, optimizers and weight decays etc.): the correlation comparison is showed in the second row of the following table (where Rényi sharpness for the last two layers are calculated for the sake of saving time). It can be seen that for a wide range of hyperparameters,  the correlation of Rényi sharpness keeps very strong, and is far better than its alternatives, including the ASAM adopted in [1]. Graphical results can be found  in the appendix of the revised version (Fig. 14, page 44)
> 2) For the setting with training ViT-B/16 on ImageNet-1k from scratch and finetuning CLIP ViT-B/32 on ImageNet-1k (both using the same checkpoints as [1]), the correlation comparison is provided in the third and fourth row in the following table. The correlation of Rényi sharpness still remains very strong, and is also far better than its alternatives. Graphical results can be found  in the appendix of the revised version (Fig. 15-16, page 44)
>
> Due to the limitation of computation resources and time, we are currently unable to complete the experiments on the language tasks. We will continue to conduct these experiments and include the results in the final version of the paper if accepted.
>
> | Measure                           | Renyi Sharpness (last layer) | Renyi Sharpness (penultimate layer) | Parameter Norm | Fisher-Rao Norm | Pac-Bayes Flat | Pac-Bayes Orig. | SAM $L_2$  | ASAM $L_2$ | SAM $L_∞$ | ASAM $L_∞$ |
> |-----------------------------------|------------------------------|--------------------------------------|----------------|-----------------|----------------|------------------|---------|---------|--------|---------|
> | **ResNet18/CIFAR10**              | **0.70**                     | **0.63**                             | -0.44          | -0.72           | -0.22          | -0.37            | 0.37    | 0.00    | 0.37   | 0.37    |
> | **ViT-B/16/Imagenet(pretrain)**   | **0.82**                     | -                                    | -0.27          | -0.50           | -0.03          | -0.06            | -0.43   | -0.59   | -0.47  | -0.55   |
> | **ViT-B/32/Imagenet(finetuning)** | **0.67**                     | -                                    | -0.16          | -0.23           | -0.15          | -0.17            | 0.17    | -0.34   | -0.23  | -0.23   |

---

> ### Author Response · Authors · 2025-11-26
>
> Finally, regarding the OOD experiments you mentioned, classical PAC-Bayes theory typically assumes that the training and test data are drawn from the same distribution, so our current theoretical results do not directly guarantee robustness in such OOD scenarios. In this paper, our analysis of Rényi sharpness is primarily aimed at understanding its relationship with standard generalization performance, rather than providing a systematic study of robustness. Although generalization and robustness are often empirically related, a rigorous analysis of robustness in OOD settings is an important but beyond-the-scope direction that we plan to explore in future work. Nevertheless, we will continue to conduct the OOD experiments and include the results in the final version of the paper if accepted.
>
> **W2: Sharpness baselines and tuning of alpha:  The provided results are for extensively tuned alpha values, whereas the baselines are apparently not tuned (e.g. the values for SAM and ASAM). Further, many sharpness variants (e.g. the ones from [1]) are omitted from the study, and there are no details on how exactly the baseline measures are computed or what they mean.**
>
> A2: Thanks for the comment. It seems there exists some misunderstanding here. In fact, the baselines in our paper are all tuned and the finally reported correlation results are the highest ones. Concretely, for SAM/ASAM we have conducted a grid search over $\rho \in \{10^{-6}\, 3\times10^{-6}\, 10^{-5}\, 3\times10^{-5}\, 10^{-4}\, 3\times10^{-4}\, 10^{-3}\, 3\times10^{-3}\, 10^{-2}\, 3\times10^{-2}\, 10^{-1}\, 0.3\, 1\}$ and select the $\rho$ with highest correlation coefficient for each task, we have made this more clear in the revised version. Regarding the concern that many sharpness variants are omitted, we'd like to clarify that the sharpness defined in [1] is actually the same as ASAM and there's a relevant description in [1]. Moreover, [6, 7] systematically compare a wide range of sharpness metrics, and based on their findings, we select as baselines those measures that exhibit the strongest correlation with generalization. Considering [1, 6, 7] together, we therefore choose these most representative sharpness measures as our baselines.  In the revised version, we have added the description of the hyperparameter search protocol as well as the ranges for the sharpness measures, and we have provided a comprehensive description of the definitions and implementations for all the sharpness measures in Appendix I (page 31, revised version), to ensure a fair comparison and reproducibility.
>
> **W3: Generalization bounds:  The generalization bound in theorem 3.2 is based on upper-bounding a generalization bound from [2] by upper-bounding the log-determinant of the Hessian with the Renyi sharpness. Using the log-determinant of the hessian (like done in [2]) would thus be better motivated by the bound. Same for theorem 3.3, where the log-determinant is also upper-bounded by the Renyi sharpness. There is thus a disconnect between the generalization bounds and the measure used.**
>
> A3: We appreciate the reviewer for this insightful question. Our PAC-Bayes bound is indeed derived from $\log \det(\mathbf{H})$, which is  a natural complexity measure from the theoretical perspective. However, for neural networks, reparameterization invariance plays a crucial role. Because for a given  model, there typically exists a family of parameterizations that can represent the same function and achieve the same performance, while with different values of $\log \det(\mathbf{H})$. As a result, using $\log \det(\mathbf{H})$ alone as a sharpness measure is not invariant within this equivalence class: the learning algorithm may converge to any of these representations, and the resulting $\log \det(\mathbf{H})$ can change accordingly. Motivated by this, our goal is to identify a sharpness measure that remains robust across such equivalent reparameterizations. In this work,  we adopt the Rényi sharpness, which empirically exhibits a stable and strong correlation with generalization.
>
> Besides, we also compute the $\log \det \bf H$ of the whole model and its correlation coefficient with the generalization gap as follows:
>
> | Task                    | CIFAR10/ResNet18 | CIFAR10/ResNet34 | CIFAR10/ViT | CIFAR100/ResNet18 | CIFAR100/ResNet34 | TinyImageNet/ResNet18 |
> |-------------------------|------------------|------------------|-------------|--------------------|--------------------|------------------------|
> | Correlation coefficient | -0.2092          | -0.2966          | -0.1954     | -0.3149            | -0.5310            | -0.6063                |
>
> ,from which we can confirm that $\log \det \bf H$ is poorly correlated with generalization.

---

> ### Author Response · Authors · 2025-11-26
>
> **W4: RSAM:  The provided RSAM algorithm seems to be brittle (as admitted by the authors: warmup required, length depends on task) and barely brings improvements beyond error bars over ASAM. Further, from Tables 4 and 5 it seems that has been tuned for RSAM, but not for the other SAM variants. Finally, there exists a plethora of SAM variants, that are all ignored in the comparison, and the study is limited in terms of dataset size (no ImageNet scale) and models (no ViTs).**
>
> A4: We appreciate the reviewer’s insightful comments.
>
> **On the necessity and sensitivity of warmup.**
> Our implementation of RSAM indeed uses warmup, and the number of warmup steps depends on the task. In our opinion, this is not a limitation of RSAM, but a simple and practical engineering trick. Since we regularize the Rényi entropy of the Hessian spectrum during training, if we apply this regularization too early — when the Hessian is far from positive definite and spectrally noisy, it would introduce additional approximation error and overhead. Therefore a sensible strategy is to  activate the regularizer after the model reaches a certain performance level (i.e., warmup). While different models progress at different rates, thus leading to different warmup lengths.
>
> **On "Improvements over ASAM are small".**
> Thanks for pointing this out. While RSAM and ASAM can be close on some tasks, RSAM shows clearer gains on others. For example, on CIFAR10/WideResNet, we obtain ASAM 96.79 vs. RSAM 97.13; similar consistent improvements appear on CIFAR100 and TinyImageNet. Moreover, in our ViT-B/16 fine-tuning experiments on CIFAR-10/CIFAR-100, RSAM consistently outperforms competing methods.
>
> For the experiment on WideResNet-28-10/CIFAR10 and ViT-B-16/CIFAR100, ASAM achieves 96.79/88.78 while SAM achieves 96.95/89.38. Overall, in several settings, ASAM underperforms SAM, whereas RSAM yields consistent improvements. We also would like to emphasize that our use of Renyi sharpness as a regularizer is only an initial attempt. Our results show that, even though RSAM relies on a rough approximation while penalizing Renyi sharpness, it still consistently outperform SAM and achieves performance comparable to other strong baselines.
>
> **On comparison protocol and additional experiments.**
> All SAM variants on CIFAR10/CIFAR100 are evaluated with their original, recommended hyperparameters (i.e., tuned settings in the respective papers), where Eigen SAM/Fisher SAM[3] also uses the same recipe. For the experiments of baselines on TinyImagenet, we searched $\rho$ on {0.00005, 0.0001, 0.0002, 0.00005, 0.0001, 0.0002, 0.0005, 0.001, 0.002, 0.005, 0.01, 0.02, 0.05, 0.1, 0.2, 0.5, 1.0, 2.0} and reported the best performance. Regarding the absence of ImageNet-scale/ViTs tasks, our main goal is to identify and validate a sharpness measure that truly correlates with generalization, addressing the gap where existing sharpness measures correlate weakly or even negatively with generalization, rather than to propose a highly practical and very scalable optimizer to replace all SAM variants. Nevertheless, we show the results of fine-tuning on CIFAR-10/CIFAR-100 using ViT-B/16 pretrained on ImageNet, showing that RSAM consistently outperforms other SAM-family methods. We have also extended the comparison to Fisher SAM [3] and Sparse SAM [4], RSAM still exhibits a consistent improvement regarding the question of SAM variants.

---

> ### Author Response · Authors · 2025-11-26
>
> | Dataset      | Model    | SGD (%)      | SAM (%)      | ASAM (%)     | FSAM (%)     | OURS (%)         |
> | ------------ | -------- | ------------ | ------------ | ------------ | ------------ | ---------------- |
> | **CIFAR10**  | ViT-B/16 | 98.06 ± 0.09 | 98.50 ± 0.05 | 98.39 ± 0.05 | 98.42 ± 0.11 | **98.59 ± 0.03** |
> | **CIFAR100** | ViT-B/16 | 88.27 ± 0.15 | 89.38 ± 0.04 | 88.78 ± 0.33 | 89.41 ± 0.11 | **89.58 ± 0.07** |
>
> | Dataset          | Model            | SGD (%)      | SAM (%)      | ASAM (%)     | Eigen-SAM (%) | FSAM (%)     | SSAM (%)     | OURS (%)         |
> | ---------------- | ---------------- | ------------ | ------------ | ------------ | ------------- | ------------ | ------------ | ---------------- |
> | **CIFAR10**      | ResNet20         | 92.68 ± 0.25 | 93.44 ± 0.07 | 93.62 ± 0.16 | 93.24 ± 0.20  | 93.54 ± 0.12 | 93.44 ± 0.14 | **93.69 ± 0.12** |
> | **CIFAR10**      | ResNet56         | 94.24 ± 0.23 | 94.96 ± 0.19 | 95.12 ± 0.08 | 94.96 ± 0.10  | 95.17 ± 0.05 | 95.15 ± 0.12 | **95.26 ± 0.12** |
> | **CIFAR10**      | WideResNet-28-10 | 96.36 ± 0.08 | 96.95 ± 0.05 | 96.79 ± 0.10 | 96.78 ± 0.06  | 96.96 ± 0.06 | 96.96 ± 0.04 | **97.13 ± 0.06** |
> | **CIFAR100**     | ResNet20         | 69.12 ± 0.17 | 70.53 ± 0.30 | 70.73 ± 0.14 | 70.51 ± 0.20  | 70.57 ± 0.32 | 70.14 ± 0.16 | **70.91 ± 0.25** |
> | **CIFAR100**     | ResNet56         | 72.60 ± 0.34 | 74.86 ± 0.23 | 75.20 ± 0.29 | 74.80 ± 0.32  | 74.91 ± 0.21 | 75.42 ± 0.18 | **75.71 ± 0.18** |
> | **CIFAR100**     | WideResNet-28-10 | 81.47 ± 0.18 | 83.55 ± 0.14 | 83.56 ± 0.11 | 82.81 ± 0.08  | 83.48 ± 0.14 | 83.47 ± 0.09 | **83.67 ± 0.09** |
> | **TinyImageNet** | ResNet50         | 59.62 ± 1.51 | 60.70 ± 0.70 | 62.56 ± 0.25 | -             | 61.21 ± 0.64 | -            | **63.33 ± 0.27** |
>
> Due to computational constraints, we have only added the above experiments for this revision, and we will further expand comparisons and include the results in the camera-ready version if accepted.
>
> **W5: reparametrization-invariant sharpness measures (w.r.t. layerwise rescaling) have been investigated before (see e.g. [1]), and I assume that the ASAM measure used in the provided study is one variant of the measures in [1]. So reparametrization invariance alone cannot be the reason for the (potential) success of a novel sharpness measure. I recommend discussing this when arguing about reparametrization invariance.**
>
> A5: This is an important point. We view reparameterization invariance as a necessary but not sufficient condition for studying correlation with generalization. The reason is that for a given minima, there typically exists a large family of function-equivalent parameterizations (via reparameterization), and optimization may converge to any member of this family. To obtain a robust and comparable measure, we therefore seek invariants within this equivalence class—hence the necessity.
>
> However, invariance alone does not guarantee a strong correlation with generalization. Many reparameterization-invariant candidates exist, but they differ in how they capture spectral structure, and thus in their empirical association with generalization. We've added the relevant detailed discussion in the revised version.

---

> ### Author Response · Authors · 2025-11-26
>
> **W6: the authors claim “in our opinion, what matters the most for characterizing the generalization is the extent of the spread of the spectrum” when introducing the Renyi sharpness as generalization measure, but it is unclear to me where this opinion stems from.**
>
> A6: We thank the reviewer for pointing this out. It's out fault that we did not explain the motivation of this paper very clearly. Revisions have been made in the revised paper.
>
> The key motivation of this paper is as follows: we realize that what really matters for characterizing the generalization lies in the *unevenness* or *average spread* of the spectrum of the Hessian. Intuitively speaking, an even spectrum (with almost identical eigenvalue) is very much desirable to ensure good generalization, since if there exists no particularly large eigen-direction, a small perturbation of data (which can be translated to weight perturbation) would just incur small loss change. More concretely, when characterizing the loss change resulting from the train-test data discrepancy, the unevenness or average spread of the the spectrum can reflect the influences from all categories of eigenvalues  of loss Hessian [5]: 1) the *top eigenvalues*, which are very important for the loss change but are of quite small quantity; 2) the *middle eigenvalues*, which are less important for the loss change individually but are of very big quantity; 3) the *tail eigenvalues*, which are normally located near zero and thus play a minor role regarding the loss change.  In contrast, the conventional sharpness measures, such as the trace or maximum eigenvalue of the loss Hessian, they care about only part of the eigenvalues. For example, the trace sharpness $\operatorname{tr}(\mathbf{H})$ actually cares about only middle eigenvalues, while the max-eigenvalue sharpness $\lambda_{max}(\mathbf{H})$ concerns only top eigenvalues. Therefore, both of them might experience significant information loss when predicting the generalization performance.  This is exactly why we choose Rényi entropy to measure the unevenness of the Hessian spectrum.
>
> As for that Rényi sharpnesss does not consider the magnitude of eigenvalues, i.e., ignoring the trace of Hessian matrix. The reason that why the Hessian trace does not matter is due to the reparametrization invariance of Rényi sharpness. In specific, we can actually rescale the parameters and thus change the overall scale of the Hessian without affecting the model’s performance. Therefore, our analysis is conducted under a fixed trace, and we focus on the *relative* differences between eigenvalues. Concretely, we normalize the spectrum by its trace to obtain a probability distribution over eigenvalues, and then use (Rényi) entropy to measure how concentrated this distribution is. We have provided a proof in the appendix that we can adjust the trace to any prescribed value while keeping the normalized eigenvalue spectrum completely unchanged.
>
> **W7: The authors describe how alpha might be chosen, depending on the spectrum, but do not elaborate how this choice might look like in practice, and then tune alpha extensively in their experimental section. Is there an intuition on when to chose low or high alpha? Is it necessary to manually inspect the spectrum and decide on multi-cluster vs uniform?**
>
> A7: We thank the reviewer for raising this point. For the evaluation of $\alpha$ (used to assess correlation): Our paper reports how the correlation between sharpness and generalization varies with $\alpha$, and we observe robust performance if $\alpha$ is chosen as  (0.5, 1.5). Thus, for comparing or ranking models by generalization, using $\alpha$ = 0.5 and 1.5 is typically sufficient to obtain stable conclusions.
>
> For the $\alpha$ as the RSAM regularizer hyperparameter: We agree that RSAM is more sensitive to $\alpha$ during training and thus tuning is required.  Currently, we lack a simple rule to select the value $\alpha$ in a universal way. Specifically, in our experiments, the best values often lie in 1.1, 1.2, and 1.3. We do agree that there needs further investigation here. We hypothesize that the sensitivity stems from the approximation used in computing the Rényi regularizer; with a more accurate estimate of Rényi sharpness, we conjecture that $\alpha\approx1.5$ is a universal choice. How to mitigate RSAM’s sensitivity to $\alpha$ is an ongoing work of ours.
>
> **W8: Some typos.**
>
> A8: We thank the reviewers for raising these points. We have made the corresponding revisions and carefully checked the manuscript.

---

> ### Author Response · Authors · 2025-11-26
>
> [1] Maksym Andriushchenko, Francesco Croce, Maximilian Müller, Matthias Hein, and Nicolas Flammarion. A modern look at the relationship between sharpness and generalization.
>
> [2] Zhiwei Jia and Hao Su, Information-Theoretic Local Minima Characterization and Regularization, ICML 2020
>
> [3] Mi, Peng, et al. "Make sharpness-aware minimization stronger: A sparsified perturbation approach." Advances in Neural Information Processing Systems 35 (2022): 30950-30962.
>
> [4] Kim, Minyoung, et al. "Fisher sam: Information geometry and sharpness aware minimisation." International Conference on Machine Learning. PMLR, 2022.
>
> [5] Sankar, Adepu Ravi, et al. "A deeper look at the hessian eigenspectrum of deep neural networks and its applications to regularization." _Proceedings of the AAAI Conference on Artificial Intelligence_. Vol. 35. No. 11. 2021.
>
> [6] Jiang, Yiding, et al. "Fantastic generalization measures and where to find them." arXiv preprint arXiv:1912.02178 (2019).
>
> [7] Dziugaite, Gintare Karolina, et al. "In search of robust measures of generalization." Advances in Neural Information Processing Systems 33 (2020): 11723-11733.
>
> ---
>
> Once again, we sincerely appreciate your thoughtful and valuable feedback, which has helped us clarify and improve our work. We hope our response address your concerns and questions.
>
> Best regards,
>
> The authors

---

### Official Review · Reviewer_VHgD · 2025-10-29

**Soundness:** 2
**Presentation:** 2
**Contribution:** 3
**Rating:** 6
**Confidence:** 4

**Summary:**

The authors propose a novel sharpness measure, Renyi sharpness, to consider the spread of the Hessian eigenvalues.
It shows a better correlation with generalization. They also propose a regularization method, RSAM, which outperforms other SAM variants.

**Strengths:**

- strong empirical results
    - strong performance of RSAM
    - strong  correlation between Renyi sharpness and generalization
- scaling invariance of Renyi sharpness (see weaknesses)

**Weaknesses:**

-  **Lack of motivation**. Why do we need to consider the **spread** of Hessian eigenvalues?
    - The authors said "uniform eigenvalue is the most desirable to ensure good generalization, since if there exists no particularly large eigen direction, small perturbation of data would just incur small loss change"
    - This seems like a wrong statement.
    - If all eigenvalues are large but similar (small spread), e.g., isotropic, then there surely exists large eigen direction, but Renyi sharpness is small as it only consider the spread, not the magnitude.
    - $H_\alpha(cI)=1/(1-\alpha)\log \sum (1/n)^\alpha=\log n$ does not depends on the value of $c>0$.

- The concept in Prop 2.2 is **not "reparameterization" invariance, but "scaling" invariance**. Renyi sharpness may not be (nonlinear) reparameterization invariant.

- It seems like $A$ in the bound in (4) is not constant and depends on $\theta$.

- There is no definition for capital $N$.

- Not a fair comparison. "We vary $\alpha$ and plot the sharpness that attains the highest correlation coefficient". At least, you should report the best $\alpha$ for each layer. It would be better to draw a scatter plot for $(\alpha, \tau)$-pairs (compared to the other baselines, e.g., trace) for each layer. If you pick the best measure (or $\alpha$) after observing the gap, it is not an useful generalization measure.

- use $\langle w_j', v_j\rangle$ or $w_j^{'\top}v_j$ to improve readability.

- It would be better to write $|g_j|^{2\alpha}$ or $(g_j^2)^\alpha$ instead of $g_j^{2\alpha}$ in (10) to avoid a confusion for the case of $\alpha=0.5$.

- Can you elaborate more on the meaning of the **layerwise** Renyi shaprness? Is it important because Prop 3.1 can be applied to a single layer?

**Questions:**

See weaknesses

---

> ### Author Response · Authors · 2025-11-26
>
> Dear reviewer VHgD,
>
> We thank the reviewers for finding this work showing strong empirical results, including strong performance of RSAM and strong correlation between Rényi sharpness and generalization. Below are our responses to the weaknesses and questions raised by the reviewer:
>
> **W1: **Lack of motivation**. Why do we need to consider the **spread** of Hessian eigenvalues?**
> 1. **The authors said "uniform eigenvalue is the most desirable to ensure good generalization, since if there exists no particularly large eigen direction, small perturbation of data would just incur small loss change"**
> 2. **This seems like a wrong statement.**
> 3. **If all eigenvalues are large but similar (small spread), e.g., isotropic, then there surely exists large eigen direction, but Renyi sharpness is small as it only consider the spread, not the magnitude.**
> 4. **$H_{\alpha}(cI)=1/(1-\alpha)\log\sum(1/n)^{\alpha}=\log n$ does not depends on the value of $c>0$.**
>
> A1: We thank the reviewer for pointing this out. It's out fault that we did not explain the motivation of this paper very clearly. Revisions have been made in the revised paper.
>
> The key motivation of this paper is as follows: we realize that what really matters for characterizing the generalization lies in the *unevenness* or *average spread* of the spectrum of the Hessian. Intuitively speaking, an even spectrum (with almost identical eigenvalue) is very much desirable to ensure good generalization, since if there exists no particularly large eigen-direction, a small perturbation of data (which can be translated to weight perturbation) would just incur small loss change. More concretely, when characterizing the loss change resulting from the train-test data discrepancy, the unevenness or average spread of the the spectrum can reflect the influences from all categories of eigenvalues  of loss Hessian [1]: 1) the *top eigenvalues*, which are very important for the loss change but are of quite small quantity; 2) the *middle eigenvalues*, which are less important for the loss change individually but are of very big quantity; 3) the *tail eigenvalues*, which are normally located near zero and thus play a minor role regarding the loss change.  In contrast, the conventional sharpness measures, such as the trace or maximum eigenvalue of the loss Hessian, they care about only part of the eigenvalues. For example, the trace sharpness $\operatorname{tr}(\mathbf{H})$ actually cares about only middle eigenvalues, while the max-eigenvalue sharpness $\lambda_{max}(\mathbf{H})$ concerns only top eigenvalues. Therefore, both of them might experience significant information loss when predicting the generalization performance.  This is exactly why we choose Rényi entropy to measure the unevenness of the Hessian spectrum.
>
> As for the concern that Rényi sharpnesss does not consider the magnitude of eigenvalues, i.e., ignoring the trace of Hessian matrix, this is indeed a sharp observation. The reason that why the Hessian trace does not matter is due to the reparametrization invariance of Rényi sharpness. In specific, we can actually rescale the parameters and thus change the overall scale of the Hessian without affecting the model’s performance. Therefore, our analysis is conducted under a fixed trace, and we focus on the *relative* differences between eigenvalues. Concretely, we normalize the spectrum by its trace to obtain a probability distribution over eigenvalues, and then use (Rényi) entropy to measure how concentrated this distribution is. We have provided a proof in the appendix that we can adjust the trace to any prescribed value while keeping the normalized eigenvalue spectrum completely unchanged.
>
>
> **W2: The concept in Prop 2.2 is not "reparameterization" invariance, but "scaling" invariance. Renyi sharpness may not be (nonlinear) reparameterization invariant.**
>
> A2: We appreciate the reviewer’s clarification. You are right that the invariance proved in Prop. 2.2 is not invariant to nonlinear reparameterizations. We will change the “reparameterization invariance” to “scaling invariance" in the revised version.
>
> **W3: It seems like $A$ in the bound in (4) is not constant and depends on $\theta$.**
>
> Q3: Thanks to the reviewers for their questions. Actually this is the most important part of our proof. In the argument, $A$ is the trace of the Hessian matrix corresponding to the minima, which is related to the weight $\theta_0$ of the minima, but not to the $\theta$ of the posterior distribution. It might be more accurate to write $A$ as $A(\theta_0)$. By exploiting scale invariance, we can always rescale $A$ arbitrarily without changing either the model behavior or the Rényi entropy. Hence, without loss of generality, we may treat $A$ as a fixed constant.

---

> ### Author Response · Authors · 2025-11-26
>
> **W4: There is no definition for capital $N$.**
>
> Q4: To avoid confusion, may we confirm that you are referring to $N$ in Eq. (4) (now Eq. (6))? In the paragraph before Eq. (4), we have defined $N$ as the training set size.
>
> **W5: Not a fair comparison. "We vary $\alpha$ and plot the sharpness that attains the highest correlation coefficient". At least, you should report the best for each layer. It would be better to draw a scatter plot for $(\alpha, \tau)$-pairs (compared to the other baselines, e.g., trace) for each layer. If you pick the best measure (or ) after observing the gap, it is not an useful generalization measure.**
>
> A5: Thanks for the suggestion. We have added per-layer scatter plots of $\tau$ versus $\alpha$ across different experiments (see Figs. 18-23, page 47, revised version) in the revised version. Moreover, We would like to politely note that we provide a heatmap of the mean of $\tau$ versus $\alpha$ in the appendix in the first submission. Please see Fig. 17 (page 46, revised version), where the average $\tau$ does not differ substantially corresponding to different choices of $\alpha$. Overall, for the two spectrum types discussed in the paper, choosing $\alpha$ = 1.5 / 0.5 already yields a strongly correlated result.
>
> **W6: use $<w_j^{'},v_j>$ or $w_j^{T}v_j$ to improve readability.**
> **W7: It would be better to write $|g_j|^{2\alpha}$ or $(g_j^2)^{\alpha}$ instead of $g_j^{2\alpha}$ in (10) to avoid a confusion for the case of $\alpha=0.5$.**
>
> A6 & A7: We thank the reviewers for their suggestions, which we have taken into account and modified accordingly in the revised version.
>
> **W8: Can you elaborate more on the meaning of the layerwise Renyi shaprness? Is it important because Prop 3.1 can be applied to a single layer?**
>
> A8: This is an excellent question. Proposition 3.1 shows that layer-wise sharpness alone is sufficient to characterize generalization, which departs from the classical view that favors a global sharpness measure and thus offers a new perspective on generalization. A key advantage is that layer-wise sharpness is scale-invariant, so it does not depend on the absolute magnitude of the Hessian; instead, it captures generalization through the relative dispersion of eigenvalues.
>
> By contrast, as we analyze in the appendix, the global Rényi sharpness is a linear combination of per-layer sharpness values, with weights given by each layer’s trace divided by the global trace. These weights can be arbitrarily altered by scale freedoms in the reparameterization, which makes the correlation between global sharpness and generalization less consistent. This also explains why, in some experiments, global sharpness correlates more weakly with generalization than layer-wise sharpness.
>
> [1] Sankar, Adepu Ravi, et al. "A deeper look at the hessian eigenspectrum of deep neural networks and its applications to regularization." _Proceedings of the AAAI Conference on Artificial Intelligence_. Vol. 35. No. 11. 2021.
>
> ---
>
> Once again, we sincerely appreciate your thoughtful and valuable insights, which have helped us clarify and improve our work. We hope our response address your concerns and questions satisfactorily.
>
> Best regards,
>
> The authors

---

> > ### Comment · Reviewer_VHgD · 2025-11-27
> >
> > It seems like $N=|S|$ in the old eqn (4) is $n$ in the old eqn (5) and $n=dim(\theta)$ in the old eqn (4) is $d$ ($\theta^\ast\in\mathbb{R}$) in the old eqn (5).

---

> > > ### Author Response · Authors · 2025-11-27
> > >
> > > We thank the reviewer for pointing this out and fully agree with the comment. We acknowledge that the confusion was caused by our inconsistent definitions across different theorems. We have now unified the definitions used in the two generalization bounds and corrected them in the revised version.

---

### Official Review · Reviewer_3Nf3 · 2025-10-30

**Soundness:** 3
**Presentation:** 3
**Contribution:** 3
**Rating:** 6
**Confidence:** 5

**Summary:**

In this paper, the authors study a new sharpness measure motivated by Rényi divergences in statistics.
The authors argue that uniformity in eigenvalues of Hessian (for a constant sum of eigenvalues) is desirable to promote generalization, as opposed to the original SAM, which aims to only minimize the largest eigenvalue of Hessian, being independent of the uniformity. They show that Rényi entropy, when applied to the normalized eigenvalues of the Hessian, can characterize the spread of the spectrum. Moreover, they prove generalization bounds from Rényi sharpness using two ideas: (1) Rényi sharpness, according to its definition, is a normalized function of Hessian eigenvalues, thus it is invariant to rescaling of parameters in homogeneous neural networks such as ReLU nets. (2) Multiplicative weight perturbations, according to orthogonal transformations. Furthermore, they experimentally show how generalization correlates with Rényi sharpness (Section 5) and propose Rényi-SAM, achieved via a new loss function. The method is validated over various datasets.







Here is a detailed summary of their main contributions: Rényi sharpness is introduced, and it is shown that it is invariant to rescaling. Its correlation to generalization is shown. In Proposition 3.1, they show that perturbation via orthogonal matrices allows us to upper bound the population loss (Equation 3). This leads to a PAC-bayesian generalization bound in Theorem 3.2 and Theorem 3.3 that relates the population loss to the empirical risk plus some term involving Rényi sharpness.


They use PyHessian to estimate the Hessian of the neural network, which leads to Section 4.1. This includes a comprehensive discussion of how to choose the parameter alpha involving Rényi sharpness.

To estimate Rényi sharpness, they propose a method based on writing the quantity in terms of the trace of powers of the Hessian, and they use the Hutchinson method along with other quadrature methods used to approximate integrals/expectations. It is given in Algorithm 1.

**Strengths:**

- A new notion of sharpness, which is theoretically and practically correlated to generalization, is of potential interest to the community

- The paper is an interesting mix of theory and practice

**Weaknesses:**

- The algorithm (Rényi sharpness) is poorly explained in the paper

**Questions:**

This is an interesting paper on SAM, relating Rényi sharpness to generalization. I think the paper is making good contributions, spanning from theory to algorithms and insights about generalization. Here are some comments/questions:


Definition 2.1: What happens if $H$ is non-positive definite? How do you define Rényi sharpness then? Please clarify this in the paper


Reparametrization invariance in Proposition 2.2: Why in Equation 1, alpha has to be different from one? Couldn't we take the limit and conclude that the identity also holds for that case?



Proposition 2.2: At first glance, I thought the invariance holds for the Hessian of the whole network, but looking at the proof, it looks like it holds only for an arbitrary fixed later. I ask the authors to clarify this in the text. The invariance follows from the fact that in the definition of Rényi sharpness, the authors introduced a normalization factor (the trace).



Please include some explanations about Algorithm 1 in the next version of the paper. Currently, it is really difficult to follow what it means.



Eq. 9: If the gradients are vectors, then what do you mean by taking squares? You mean $hh^T$?


How do you optimize the objective in Eq. 11? Do you compute the gradients of it? Do you have something like a 'base optimizer' similar to the original SAM? If you compute the gradient of Eq. 11, then you have second-order derivative terms (prohibited). Do you use the same approximations as the original SAM to resolve this? I believe a rigorous explanation of what the Rényi SAM algorithm really is is required for the next version of the paper. Since this is one of the main contributions of this paper, you need to explain this in detail, probably having a clear algorithm dedicated to it.


There are some other notions of sharpness similar to Rényi in recent works that have never been discussed in the paper: For instance, I found 'Tilted SAM' in [1] and 'Frobenius SAM' in [2] on the web. There might also exist others, but at least these two are pretty similar. At least one expects some discussion and comparison in the paper. Please do search for more because I believe there might be other papers.


[1] Li, Tian, Tianyi Zhou, and Jeff Bilmes. "Tilted sharpness-aware minimization." ICML 2025

[2] Tahmasebi, Behrooz, Ashkan Soleymani, Dara Bahri, Stefanie Jegelka, and Patrick Jaillet. "A universal class of sharpness-aware minimization algorithms." ICML 2024

---

> ### Author Response · Authors · 2025-11-26
>
> Dear reviewer 3Nf3,
>
> We thank the reviewers for finding this work interesting and with good contributions. Below are our responses to the weaknesses and questions raised by reviewer:
>
> **W1: The algorithm (Rényi sharpness) is poorly explained in the paper**
>
> WA1: We thank the reviewers for pointing this out. Revisions have been made in the revised paper, we now explicitly show Rényi sharpness can be written as
> $$
> H_\alpha(H)
> = \frac{1}{1-\alpha}\log\frac{\mathrm{Tr}(H^\alpha)}{\mathrm{Tr}(H)^\alpha},
> $$
> and clarify that our goal is to estimate both $\mathrm{Tr}(H^\alpha)$ and $\mathrm{Tr}(H)$ without computing the eigenvalues of $H$.
>
> To estimate the trace of matrix functions $f(\mathbf{H})$, the stochastic trace estimator can be leveraged to greatly reduce the complexity:
> $$
>     \mathrm{Tr}(f(\mathbf{H}))=\mathrm{Tr}(f(\mathbf{H})\mathbf{I})=\mathrm{Tr}(f(\mathbf{H})\mathbb{E}[\mathbf{vv}^\top])=\mathbb{E}[\mathrm{Tr}(f(\mathbf{H})\mathbf{vv}^\top)]=\mathbb{E}[\mathbf{v}^\top f(\mathbf{H})\mathbf{v}],
> $$
> where $f$ is analytic inside a closed interval function, $\mathbf{I}$ is the identity matrix, and $\mathbf{v}$ is sampled from a Rademacher distribution.
>
> To calculate the expectation of the quadratic form $\mathbf{v}^\top f(\mathbf{H})\mathbf{v}$, the Gaussian quadrature rule can be employed to transform the expectation as an integral. Further, we can compute the nodes and the weights of the quadrature rule by the Lanczos algorithm [7] which basically generates an orthonormal basis for the Krylov subspace such that the matrix can be reduced to tri-diagonal one, hence greatly lower the computational burden. Combined all the above, it constitutes the framework of the stochastic Lanczos quadrature (SLQ) algorithm [8], which is exactly the basis of Algorithm 1.
>
> **Q1: Definition 2.1: What happens if H is non-positive definite? How do you define Rényi sharpness then? Please clarify this in the paper.**
>
> QA1: We agree with the reviewer that this is a very important issue. In fact, when we are considering the generalization performance, we typically analyze the Hessian at a (local) minima theoretically, and therefore assume the Hessian to be positive definite. In practice, however, due to imperfect convergence or numerical errors in the algorithm, some negative eigenvalues may appear. Since these negative eigenvalues usually have very small magnitudes, we commonly take their absolute values before performing subsequent computations. We have made this statement clear in the revised version.
>
> **Q2: Reparametrization invariance in Proposition 2.2: Why in Equation 1, alpha has to be different from one? Couldn't we take the limit and conclude that the identity also holds for that case?**
>
> QA2: We thank the reviewers for this thoughtful question. For the case of alpha equal to 1, we fully agree with this conclusion. In fact, when alpha is equal to 1, the Rényi entropy reduces to the Shannon entropy. Under reparameterization, only the scale of the eigenvalues changes, while their normalized probability distribution remains the same; consequently, the Shannon entropy is invariant as well. We have added a clarification of this point in the revised version.
>
> **Q3: Proposition 2.2: At first glance, I thought the invariance holds for the Hessian of the whole network, but looking at the proof, it looks like it holds only for an arbitrary fixed later. I ask the authors to clarify this in the text. The invariance follows from the fact that in the definition of Rényi sharpness, the authors introduced a normalization factor (the trace).**
>
> QA3: We thank the reviewers for pointing this out. We stated in Proposition 2.2 that we were targeting the layer weights, but since this could be easily missed due to the length of the text, we will add a clear statement about LAYERWISE in the revised version.
>
> **Q4: Please include some explanations about Algorithm 1 in the next version of the paper. Currently, it is really difficult to follow what it means.**
>
> QA4: We thank again for the reviewer's comment, and please see WA1 for our response.

---

> ### Author Response · Authors · 2025-11-26
>
> **Q5: Eq. 9: If the gradients are vectors, then what do you mean by taking squares? You mean ${\bf hh}^T$?**
>
> QA5: We thank the reviewer for pointing out this confusion. In fact, we first convert the vector into a matrix through diagonalization, and then take the square of the matrix. We have modified Equation 9 (now Eq. 11) to ${\bf GM}=\big[\mathrm{Diag}(\frac{1}{N}\sum_{i=1}^N\nabla_{\boldsymbol{\theta}}l(\boldsymbol{\theta},{\bf x}_i,{\bf y}_i))\big]^2$ to avoid such confusion.
>
> **Q6: How do you optimize the objective in Eq. 11? Do you compute the gradients of it? Do you have something like a 'base optimizer' similar to the original SAM? If you compute the gradient of Eq. 11, then you have second-order derivative terms (prohibited). Do you use the same approximations as the original SAM to resolve this? I believe a rigorous explanation of what the Rényi SAM algorithm really is is required for the next version of the paper. Since this is one of the main contributions of this paper, you need to explain this in detail, probably having a clear algorithm dedicated to it.**
>
> QA6: This is a very important question. Due to page limitation, we placed the RSAM algorithm in Appendix J.3.2 (see Page 36, Algorithm 2) in our submission. In fact, we have the same optimization steps as the original SAM:
> 1. calculate the loss by forward propagation, then compute the gradient of the loss,
> 2. add $\epsilon$ to the weight according to Equation 13,
> 3. calculate the gradient for the new weight,
> 4. update the original weight according to the latest gradient via gradient descent.
>
> Those steps are almost the same as the original SAM, and the difference lies in the $\epsilon$ added to the original weight.
>
> **Q7: There are some other notions of sharpness similar to Rényi in recent works that have never been discussed in the paper: For instance, I found 'Tilted SAM' in [1] and 'Frobenius SAM' in [2] on the web. There might also exist others, but at least these two are pretty similar. At least one expects some discussion and comparison in the paper. Please do search for more because I believe there might be other papers.**

---

> ### Author Response · Authors · 2025-11-26
>
> QA8: We thank the reviewer for pointing out the related work. In the revised version, we  explicitly compare our method with several SAM variants as follows:
>
> Vanilla SAM [1] has been shown to implicitly minimize the largest eigenvalue of the training loss Hessian, and Sparse SAM [2], which accelerates SAM by explicitly masking part of the updates, essentially targets the same quantity. Eigen SAM [3] directly penalizes the largest eigenvalue in its minimization step. Tilted SAM [4] samples noise in multiple directions to perturb the weights and penalizes the sum of the exponentiated perturbed losses over these noise samples. Intuitively, the exponential transform amplifies the sharpest directions of the loss landscape, so it imposes stronger penalty along these directions. From this perspective, Tilted SAM can be viewed as effectively penalizing the largest (or relatively large) eigenvalues of the Hessian. Frobenius SAM [5] penalizes the Frobenius norm of the Hessian matrix; if we normalize this norm by the squared trace, the resulting quantity becomes essentially a monotone function of the order-2 Rényi entropy. Fisher SAM [6] minimizes the same type of robust objective as SAM, but with the neighborhood defined by a Riemannian metric induced by the Fisher information; this is equivalent to penalizing the largest eigenvalue of the Hessian with respect to the Fisher metric.
>
> Overall, these methods regularize some spectral function of the Hessian eigenvalues. Whether one penalizes the largest eigenvalue or minimizes the Frobenius norm of the Hessian, the implicit goal is to encourage the eigenvalues to move closer to each other; for example, reducing the largest eigenvalue typically decreases the overall spread of the spectrum.
>
> In contrast, Rényi sharpness explicitly focuses on the spread of the normalized eigenvalues. Modern deep models usually enjoy certain reparameterization invariances, so we can rescale the overall magnitude of the Hessian without changing the model’s behavior. Consequently, if the regularizer depends only on the unnormalized eigenvalues (such as the spectral norm or a generic spectral function), then shrinking the global scale of the Hessian will always reduce the regularization term, even when the model performance and the relative shape of the spectrum remain unchanged. Therefore, minimizing such penalties alone does not guarantee that the eigenvalues become more even distributed.
>
> We have added the above discussion and the experiments comparing RSAM with Fisher SAM/Sparse SAM in the revised version. RSAM consistently achieves stable performance gains over these baselines on CIFAR10/CIFAR100/TinyImageNet. Fully tuning TSAM with its many additional hyperparameter settings would require a prohibitively large computational budget during the rebuttal period. We are conducting those experiments, and we will add the results to the final version if accepted. We are not going to compare Frobenius SAM because in the original paper, the performance of Frobenius SAM could not surpass the performance of basic SAM.
>
> [1] Foret, Pierre, et al. "Sharpness-aware minimization for efficiently improving generalization." arXiv preprint arXiv:2010.01412 (2020).
>
> [2] Mi, Peng, et al. "Make sharpness-aware minimization stronger: A sparsified perturbation approach." Advances in Neural Information Processing Systems 35 (2022): 30950-30962.
>
> [3] Luo, Haocheng, et al. "Explicit eigenvalue regularization improves sharpness-aware minimization." Advances in Neural Information Processing Systems 37 (2024): 4424-4453.
>
> [4] Li, Tian, Tianyi Zhou, and Jeff Bilmes. "Tilted sharpness-aware minimization." ICML 2025
>
> [5] Tahmasebi, Behrooz, Ashkan Soleymani, Dara Bahri, Stefanie Jegelka, and Patrick Jaillet. "A universal class of sharpness-aware minimization algorithms." ICML 2024
>
> [6] Kim, Minyoung, et al. "Fisher sam: Information geometry and sharpness aware minimisation." International Conference on Machine Learning. PMLR, 2022.
>
> [7] Golub, Gene H., and Charles F. Van Loan. Matrix computations. JHU press, 2013.
>
> [8] Ubaru, Shashanka, Jie Chen, and Yousef Saad. "Fast estimation of tr(f(A)) via stochastic Lanczos quadrature." SIAM Journal on Matrix Analysis and Applications 38.4 (2017): 1075-1099.
>
> ---
>
> We hope these clarifications and modifications could address your concerns and questions. We sincerely thank you again for your valuable insights, which have helped us improve the clarity and depth of our paper.
>
> Best regards,
>
> The authors

---

### Official Review · Reviewer_2BS3 · 2025-10-31

**Soundness:** 2
**Presentation:** 2
**Contribution:** 2
**Rating:** 2
**Confidence:** 4

**Summary:**

The paper introduces a novel measure using the equation for Renyi entropy. The normalized eigenvalues of the loss Hessian are treated as a distribution, and the Renyi entropy of this distribution is used as a measure. A generalization bound is derived in terms of this entropy, and the experimental results using Renyi entropy regularizer showed small improvements on accuracy on benchmark data.

**Strengths:**

The presented method is novel. Experiments show improvement in accuracy.

**Weaknesses:**

The presented method is novel, but it is difficult to recommend the paper for acceptance for several reasons.

The most problematic aspect is that the entropy is defined for a probability distribution, whereas eigenvalue spectrum is not a probability distribution. It is unclear what probability model the authors assume and whether it is valid. In the derived bound, the authors use a posterior distribution Q, but its definition and justification are not clearly explained.  It is very hard to recognize the eigenvalue spectrum as a distribution because no corresponding random variable is defined. Does the largest eigenvalue represent \the probability of the first random variable? The overall setup and explanation are too vague to assess whether the application of Renyi entropy is conceptually meaningful. For example, I have seen using the information-theoretic measures for the output of the network because the output can be naturally considered as a probability distribution over predicted classes. However, in the case of Hessian eigenvalues, such a probabilistic interpretation is not justifiable.

The explanation in lines 176-184 is unclear What are the functions h(.), g(.) and other perturbation constants A and rho?

Hessian should be a very large matrix, making its computation and eigenvalue calculation at each optimization step computationally expensive. The authors used the square of gradient to approximate the hessian eigenvalues in the experiment without sufficient explanation. Is the eigenvalue distribution related to the square of gradient components in each dimension? During optimization, the gradient should approach zero, while the hessian does not.

**Questions:**

Please address the weaknesses to increase score.

---

> ### Author Response · Authors · 2025-11-26
>
> Dear reviewer 2BS3,
>
> We thank the reviewers for finding this work novel and improvement in accuracy. Below are our responses to the weaknesses and questions raised by the reviewer:
>
> **W1: The most problematic aspect is that the entropy is defined for a probability distribution, whereas eigenvalue spectrum is not a probability distribution. It is unclear what probability model the authors assume and whether it is valid. In the derived bound, the authors use a posterior distribution Q, but its definition and justification are not clearly explained. It is very hard to recognize the eigenvalue spectrum as a distribution because no corresponding random variable is defined. Does the largest eigenvalue represent the probability of the first random variable? The overall setup and explanation are too vague to assess whether the application of Renyi entropy is conceptually meaningful. For example, I have seen using the information-theoretic measures for the output of the network because the output can be naturally considered as a probability distribution over predicted classes. However, in the case of Hessian eigenvalues, such a probabilistic interpretation is not justifiable.**
>
> A1: Thanks for pointing this out and sorry for the misunderstanding and confusion we caused. We really need to convey the role of Rényi entropy in characterizing the sharpness in a clear way.
>
> Actually, we are not treating the Hessian spectrum as a distribution, rather, the sole purpose of using Rényi entropy in our paper is to measure the *unevenness* of *average spread* of a *non-negative vector*, which can be translated to a *virtual* probability vector by normalization. The non-negativity of the Hessian spectrum near minima thus justifies using the Rényi entropy to measure the unevenness of the former.
>
> Moreover, a key observation of ours is that generalization heavily depends on the average spread (unevenness) of the Hessian spectrum (as explained in the revised version of our paper). Combined the above, it is thus reasonable to use Rényi entropy as an indicator of generalization.
>
> As for the posterior distribution Q in the informal derived bound (Thm. 3.1),  Q is defined via local deviations, i.e., Q has a density $q(w)\propto e^{-|L_0-L(w)|}$, where $L(w)$ is the loss function and $L_0$ is the minima loss obtained by the optimization algorithm. This is the original setting used by [4] and we have stated this more clearly in the revised version.
>
> **W2: The explanation in lines 176-184 is unclear What are the functions h(.), g(.) and other perturbation constants A and rho?**
>
> A2: We thank the reviewers for pointing this our. Due to page limitations we were not able to explain this in detail in the initial version, below we will discuss it in more detail and make corresponding updating in the revised version.  A neural network can be written as a composite function $f=g({\bf W}h(x))$, where $\bf W$ is the weight at a given layer,  $h(x)$ is the function consisting of the layers in front of $\bf W$ all the way to the input, and $g$ is the function behind the $\bf W$  all the way to the output. We have moved the Proposition 3.1 before Equation 2 (now Equation 5), and $\bf A$, $\rho$ is defined as  in Proposition 3.1.

---

> ### Author Response · Authors · 2025-11-26
>
> **W3: Hessian should be a very large matrix, making its computation and eigenvalue calculation at each optimization step computationally expensive. The authors used the square of gradient to approximate the hessian eigenvalues in the experiment without sufficient explanation. Is the eigenvalue distribution related to the square of gradient components in each dimension? During optimization, the gradient should approach zero, while the hessian does not.**
>
> A3: We thank the reviewer for the insightful comment and have clarified this point in the revised version.
>
> In our experiments, we do not compute the full Hessian or its exact eigenvalues. Instead, at each step we first compute the mini-batch gradient
> $$
> g_B(\theta) = \frac{1}{|B|}\sum_{i \in B} g_i(\theta),
> $$
> and then take its element-wise square, $g_B(\theta) \odot g_B(\theta)$, as a *diagonal curvature proxy*. This corresponds to the so-called *gradient-magnitude (GM) approximation*, where
> $$
> \mathrm{GM}=
> \big[\mathrm{Diag}(g_B(\theta))\big]^2
> \approx
> \mathrm{Diag}\Big(\frac{1}{|B|}\sum_{i \in B} g_i(\theta)g_i(\theta)^\top\Big),
> $$
> and the right-hand side is the diagonal of the empirical Fisher / Gauss--Newton matrix for negative log-likelihood losses, which is a standard diagonal approximation to the Hessian. This GM approximation has been explicitly used as a scalable approximation of the Hessian/Fisher matrix in prior work (e.g.,[1,2,3]), and we use the resulting eigenvalues of the GM approximation as a surrogate for the Hessian eigenvalues.
>
> Regarding the concern that “the gradient should approach zero while the Hessian does not”: this statement refers to the *full-data mean gradient* at an exact minima, the specific form (GM) we use indeed makes the approximated Hessian vanish when computed over the entire dataset. In contrast, however, the true Hessian or Fisher matrix at that point would in general remain nonzero. We chose GM because it is much faster to compute than Fisher, at the cost of a somewhat large approximation error. Moreover, (i) our proxy is computed on the mini-batch in training, where the mini-batch gradient $g_B(\theta)$ is generally not exactly zero, and (ii) we are not looking to penalize Rényi sharpness at minima; instead, we are looking to motivate SGD to find a minima with smaller sharpness by penalizing Rényi sharpness while the minima has not yet been reached.
>
>
>
> [1] Khan, Mohammad, et al. "Fast and scalable bayesian deep learning by weight-perturbation in adam." International conference on machine learning. PMLR, 2018.
>
> [2] Bottou, Léon, Frank E. Curtis, and Jorge Nocedal. "Optimization methods for large-scale machine learning." SIAM review 60.2 (2018): 223-311.
>
> [3] Kim, Minyoung, et al. "Fisher sam: Information geometry and sharpness aware minimisation." International Conference on Machine Learning. PMLR, 2022.
>
> [4] Jia, Zhiwei, and Hao Su. "Information-theoretic local minima characterization and regularization." International Conference on Machine Learning. PMLR, 2020.
>
> ---
>
> Once again, thank you for your valuable feedback. We believe your suggestions will improve the clarity and impact of our work. We hope our responses address your concerns satisfactorily.
>
> Best regards,
>
> The authors

---

### Author Response · Authors · 2025-11-26
**General Response**

Dear PCs, SACs, ACs, and Reviewers,

We sincerely thank you for the time, effort, and patience you have devoted to reviewing our work; your feedback has been extremely helpful in improving the paper. Considering there are some common concerns regarding our work, we'd like to provide brief responses at the very front.

First, regarding the motivation of the paper and intuitive explanations of the idea, we admit we have not conveyed them clearly in the submitted version, which has been significantly improved, in our opinion, in the revised version.

The key motivation of our paper is as follows: we realize that what really matters for generalization is the *unevenness* or *average spread* of the Hessian spectrum. Intuitively speaking, an even or uniform spectrum is very much desirable for generalization: if there is no particularly large eigen-direction, a small data perturbation (equivalently, a small weight perturbation) will only induce a small loss change. More concretely, the unevenness of the spectrum can reflect the influences from all three categories of eigenvalues: 1) the *top* eigenvalues, which are very influential but of small quantity; 2) the *middle* eigenvaluess, which are less important individually but of very big quantity, thus are very important collectively; 3) the *tail* eigenvalues, which are typically near zero and thus play a minor role in the loss change. In contrast, the trace sharpness $\operatorname{tr}(\mathbf{H})$  cares about only the middle eigenvalues, while the max-eigenvalue sharpness $\lambda_{max}(\mathbf{H})$ concerns only the top ones. So both of them might experience significant information loss when predicting  generalization.

Moreover, Rényi entropy is a very useful tool for measuring the unevenness of the Hessian spectrum (It is worth noting that there's actually no random variables involved). Thus we come up with the idea of Rényi sharpness.

Consistent with the above-mentioned advantage of Rényi sharpness over conventional sharpness, Rényi sharpness exhibits strong and stable correlation in our extensive experiments, far better than the conventional sharpness. Based on the above reasoning and the experimental results, we believe that Rényi sharpness might correlate strongly with generalization in a *universal* way.

Finally, we'd like to mention that the RSAM in our paper is an initial attempt to take advantage of Rényi sharpness for regularization.  Experiments show that by utilizing a  coarse approximation of Rényi sharpness, it can consistently outperform SAM and be competitive with state-of-the-art methods. We believe lots of work can be done for further improvement in this direction.

Thanks again for your excellent work!

Best regards,

The authors.

---

### Author Response · Authors · 2025-12-02
**Summary Comment**

Dear AC,

Considering the increased workload of ACs after the leakage, we'd like to make a very brief description of the major concerns of the reviewers as well as our responses at first.

**The biggest concern** of Reviewer 2BS3 (with score 2) lies in that he/she questions the validity of employing Rényi entropy for characterizing the Hessian spectrum, since there's no random variable involved.

*Our response*: There is a big misunderstanding here. Actually we're not dealing with any random variable by using Rényi entropy. Rather, Rényi entropy is employed for the purpose of measuring the unevenness (average spread) of a *non-negative vector*, as is the case for the Hessian spectrum. We have made this point more clearly in our revised version to avoid misunderstanding.


**The biggest concern** of Reviewer  NtAE (with score 2) lies in that he/she doubts the correlation between the sharpness and generalization exists in a universal way, given the somewhat negative results in [1], where extensive experiments are conducted over large-scale datasets and models. In light of this, he/she thinks a similarly extensive and large-scale experiments are required for supporting our claim.

*Our response*: First of all, we think the negative results in [1] are mainly due to the improper sharpness measures they used, in specific, the employed Adaptive sharpness (or ASAM), which tends to *ignore* the top eigenvalues or middle eigenvalues of the Hessian spectrum, thus in the risk of losing important information for predicting generalization. Second, a lot of large-scale experimental results have been added in the revised version, which consistently demonstrate the strong correlation between the Rényi sharpness and generalization.

[1] Maksym Andriushchenko, Francesco Croce, Maximilian Müller, Matthias Hein, and Nicolas Flammarion. A modern look at the relationship between sharpness and generalization.

**The major concerns** of Reviewer 3Nf3 (with score 6) lie in that he/she thinks more detailed and clearer description of the relevant algorithms as well as more discussions and comparison between  Rényi sharpness and existing sharpness measures are needed.

*Our response*: We have added a more detailed description of the Rényi sharpness estimation algorithm and the RSAM in the revised version. Moreover, we have added a dedicated discussion of related SAM methods (including Tilted SAM, Frobenius SAM, Fisher SAM, Eigen SAM, Sparse SAM, etc.) and supplemented experimental results comparing RSAM to Fisher SAM and Sparse SAM.


**The major concerns** of Reviewer VHgD (with score 6) lie in that he/she thinks the motivation of this paper is not clearly described, besides, he/she has some doubt about the insensitivity of Rényi sharpness to the scale (i.e., trace) of the Hessian.


*Our response*: We have re-described the motivation of this paper in a more intuitive way, with the key point that conventional sharpness tends to ignore some important information about the spectrum, while our proposed Rényi sharpness can capture all the relevant spectrum information. Regarding the insensitivity of Rényi sharpness to the scale (i.e., trace) of the Hessian, it is mainly due to scale-invariance of the model and reparameterization-invariance of Rényi sharpness. We have explained this point more explicitly in the revised version.

We hope the above "abstract" of the comments and responses of this paper could be of help for your evaluation.

Best regards,

The authors

---

### Meta-Review · Area_Chair_jNXs · 2026-01-09

**Summary:**

The paper introduces "Rényi sharpness" as a novel generalization measure, utilizing Rényi entropy to quantify the "unevenness" of the loss Hessian's spectrum rather than just its trace or maximum eigenvalue. The authors theoretically connect this measure to generalization through PAC-Bayesian bounds and empirically demonstrate its correlation across several vision tasks. Furthermore, a regularization algorithm, RSAM, is proposed to minimize this sharpness during training. While reviewers acknowledged the novelty of the metric and the strength of the empirical correlation in the initial experiments, significant concerns were raised regarding the motivation for using Rényi entropy on eigenvalues , the clarity of the algorithm's derivation , the comparison with existing baselines (especially on larger-scale setups like ImageNet) and the robustness of the proposed RSAM optimizer compared to established SAM variants.

While the snapshot of ratings (2, 2, 6, 6) suggests a split decision, the two negative reviews were based on issues that were effectively resolved during the rebuttal. Specifically, the authors corrected the misunderstanding regarding the mathematical validity of entropy for eigenvalues and experimentally satisfied the request for large-scale validation by adding ImageNet/ViT results. As the negative reviewers did not engage further to update their scores following these substantive improvements, the recommendation is to accept.

**Reviewer Concerns:**

*** ADDRESSED

The authors clarified the conceptual use of Rényi entropy (not as a probability distribution of random variables, but as a measure of vector unevenness), which resolved a major misunderstanding for Reviewer 2BS3. They also significantly expanded their experimental suite to include larger-scale settings (ViT on ImageNet, rigorous hyperparameter tuning resembling Andriushchenko et al., 2023), which helped alleviate Reviewer NtAE's concerns about the universality of the correlation. The authors further clarified technical details regarding the "gradient magnitude" approximation used in RSAM and distinguished "scaling invariance" from "reparameterization invariance" in response to Reviewer VHgD and 3Nf3.

*** OUTSTANDING

Reviewer NtAE pointed out that RSAM appears brittle, requiring task-dependent warm-up periods and extensive tuning of the $\alpha$ parameter, unlike more stable baselines. While the authors defended this as a standard engineering trick, the sensitivity of the method to hyperparameters suggests it may not yet be a "plug-and-play" replacement for SAM. Additionally, while the correlation results are strong, some skepticism persists regarding whether the theoretical bounds (derived via log-determinant) fully justify the specific use of Rényi entropy over other spectral functionals, as noted by Reviewer NtAE and VHgD.

**Reviewer Scores:**

Reviewer 2BS3 (Initial Score: 2) likely would have raised their score to a 4 or 5. Their primary objection was a fundamental misunderstanding of the entropy definition ("eigenvalue spectrum is not a probability distribution"). The authors' rebuttal clarified that this is a standard mathematical tool for measuring vector spread, not a claim about random variables. However, the reviewer did not engage further to update their score following these improvements.

Reviewer 3Nf3 (Initial Score: 6) likely would have maintained or slightly increased their score to a 7. They were already positive about the contribution but requested algorithm details and broader comparisons. The authors provided the requested algorithm clarification and added discussions on related methods like Tilted and Fisher SAM, directly addressing the reviewer's main critiques.

Reviewer VHgD (Initial Score: 6) likely would have maintained a 6. They raised valid technical points about scaling invariance versus reparameterization invariance and questioned the motivation for uniform eigenvalues. The authors conceded the terminology correction and provided intuition, but the fundamental theoretical gap regarding why "spread" specifically matters (beyond just trace/magnitude) might still feel heuristic to this reviewer.

Reviewer NtAE (Initial Score: 2) likely would have raised their score to a 4 or 5. This reviewer was the most critical regarding experimental rigor and baselines. The authors added significant new experiments (ViT on ImageNet, comparison to Fisher/Sparse SAM) that directly addressed the "scale" critique. However, the reviewer's concerns about the brittleness of the RSAM optimizer (warm-up, tuning) remain valid and likely prevent a strong accept recommendation. The reviewer did not engage further to update their scores.

---

### Decision · Program_Chairs · 2026-01-26

Accept (Poster)